# Learning Dynamics of Logits Debiasing for Long-Tailed Semi-Supervised Learning

**Yue Cheng**[1,2,*]  **Jiajun Zhang**[1,*]  **Xiaohui Gao**[3]  **Weiwei Xing**[1,✉]  **Zhanxing Zhu**[4,✉]

[1]Beijing Jiaotong University    [2]AntGroup
[3]Northwestern Polytechnical University    [4]University of Southampton
{yuecheng,jiajunzhang}@bjtu.edu.cn   gaitxh@foxmail.com
wwxing@bjtu.edu.cn   z.zhu@soton.ac.uk

## Abstract

Long-tailed distributions are prevalent in real-world semi-supervised learning (SSL), where pseudo-labels tend to favor majority classes, leading to degraded generalization. While many long-tailed semi-supervised learning (LTSSL) methods have been proposed, the mechanisms by which they implicitly debias logits remain poorly understood. In this work, we revisit LTSSL through the lens of learning dynamics and provide a theoretical characterization of logits debiasing. Specifically, we derive a step-wise decomposition of the logits updates, showing that predictions are dominated by class-imbalance bias that reliably reflects label priors. To expose this effect, we use the logits of a task-irrelevant baseline image as an indicator of accumulated bias and prove that they converge to the class prior. This provides a unified view where LTSSL remedies such as logit adjustment, reweighting, and resampling correspond to reshaping gradient dynamics. Based on this insight, we propose DyTrim, a principle-based dynamic pruning framework that reallocates gradient budget through class-aware pruning on labeled data and confidence-based soft pruning on unlabeled data. We provide theoretical guarantees that DyTrim reduces class bias and improves generalization. Extensive experiments on standard LTSSL benchmarks show consistent gains across architectures and methods. Code available at: https://jiajun0425.github.io/DyTrim

## 1 Introduction

Semi-supervised learning (SSL), exemplified by FixMatch (Sohn et al., 2020) and ReMixMatch (Berthelot et al., 2019), has been proven to demonstrate significant generalization advantages over supervised learning, particularly in deep neural networks (Li et al., 2025). However, many existing SSL variants, *e.g.* FlexMatch (Zhang et al., 2021), FreeMatch (Wang et al., 2023b) implicitly assume that both labeled and unlabeled data are drawn from a balanced class distribution, *i.e.*, class imbalance. In practice, real-world datasets commonly exhibit a long-tailed label distribution, leading to *biased pseudo-label* toward majority classes. This discrepancy poses significant challenges to the effectiveness of SSL algorithms on real-world datasets.

Recent studies on long-tailed semi-supervised learning (LTSSL) have emerged to mitigate the bias introduced by class imbalance in both labeled and unlabeled data. These methods range from distribution alignment (Wei et al., 2021; Kim et al., 2020), data rebalancing (Fan et al., 2022; Lee et al., 2021), logit adjustment variants (Wei & Gan, 2023; Zhou et al., 2024), to foundation model-based methods (*e.g.*, LADaS; Zheng et al., 2025). In particular, the approach employs a baseline image introduced by Lee & Kim, 2024 as a simple yet effective tool for quantifying classifier bias, which has garnered significant attention in the community (Xing et al., 2025; Yi et al., 2025). Despite these advancements, the underlying mechanisms of how class bias emerges and why existing approaches can mitigate it remain largely unexplored and poorly understood. That also prevents us from exploring a principle-based method to improve performance.

---

*Equal contribution

In this paper, we analyze the underlying mechanisms of class debiasing through the lens of learning dynamics in long-tailed semi-supervised learning (LTSSL), investigating how inputs, the classifier, and pseudo-labels interact and recursively shape one another during training. Specifically, we derive a stepwise decomposition of logit updates in SSL, showing that class imbalance dominates the predictions and prevents the model from leveraging inter-sample similarity, thereby impairing generalization. We further point out that in the learning dynamics of LTSSL, the logits of the baseline image serve as an indicator of the accumulated influence of the network's bias. Building on this framework, we offer a unified view of existing debiasing methods, including logit adjustment (LA) (Menon et al., 2021), reweighting (Wang et al., 2017), and resampling (JAPKOWICZ, 2000), which can all be understood through the lens of learning dynamics.

As a side product of this analysis, we propose a pruning-based debiasing framework for long-tailed remedies, named DyTrim. For labeled data, we compute class-wise pruning ratios to rebalance samples. For unlabeled data, we apply a label-agnostic criterion that prunes low-confidence, inconsistent samples. Beyond empirical improvements, we provide theoretical guarantees demonstrating how our method alleviates class bias and improves generalization. Extensive experiments demonstrate that our method consistently improves LTSSL performance across standard benchmarks and various backbone architectures.

## 2 PRELIMINARIES

**Notions.** We consider a labeled dataset $\mathcal{X} = \{(x^n, y^n)\}_{n=1}^N$ with $N$ samples and an unlabeled dataset $\mathcal{U} = \{u^m\}_{m=1}^M$ with $M$ samples, where $x^n \in \mathbb{R}^d$ is the $n$-th labeled sample with label $y^n \in [C] = \{1, \ldots, C\}$, and $u^m \in \mathbb{R}^d$ is the $m$-th unlabeled sample. Let $N_c$ and $M_c$ denote the number of labeled and unlabeled samples in class $c$, such that $\sum_{c=1}^C N_c = N$ and $\sum_{c=1}^C M_c = M$. If classes are sorted by size, we have $N_1 \geq N_2 \geq \cdots \geq N_C$, and define the imbalance ratios as $\gamma_l = {}^{N_1}/_{N_c} \geq 1$ and $\gamma_u = {}^{\max\{M_i\}}/_{\min\{M_i\}} \geq 1$, respectively. We denote the classifier by $f_\theta : \mathbb{R}^d \mapsto 1, \ldots, C$ with parameters $\theta$, and its logits by $g_\theta(x) \in \mathbb{R}^C$, where $f_\theta(x) = \arg\max_c g_\theta(x)_c$ and $(\cdot)_c$ denotes the $c$-th component. For each iteration of training, we sample minibatches $\mathcal{MX} = \{(x_b^n, y_b^n) : b \in (1, \ldots, B)\} \subset \mathcal{X}$ and $\mathcal{MU} = \{(u_b^m) : b \in (1, \ldots, \mu B)\} \subset \mathcal{U}$ from the training set, where $B$ denotes the minibatch size and $\mu$ denotes the relative size of $\mathcal{MU}$ to $\mathcal{MX}$. For brevity, when clear from context we drop the superscript on $u_b^m$ ($x_b^m$) and simply write $u_b$ ($x_b$).

**Base SSL algorithms.** We use FixMatch (Sohn et al., 2020) as the base SSL algorithm, following other LTSSL studies. Specifically, FixMatch first predicts the class probability of a weakly augmented unlabeled data point $\alpha(u_b)$ as $q_b = \pi_\theta(y|\alpha(u_b))$ and then generates hard pseudo-label $\hat{q}_b = \arg\max_c(q_{b,c})$, where $\pi_\theta(y|\cdot) = \texttt{Softmax}(g_\theta(\cdot))$. For consistency regularization, FixMatch uses a hard pseudo-label $\hat{q}_b$ only when $\max_c(q_{b,c}) \geq \tau$, where $\tau$ denotes a predefined confidence threshold, to improve the quality of the pseudo-labels used for training. We express the training losses of FixMatch $\mathcal{L}$ as:

$$\mathcal{L}(x_b, u_b, \hat{q}, \tau; \theta) = \mathcal{L}_{sup}(\alpha(x_b); \theta) + \mathcal{L}_{con}(\mathcal{A}(u_b), \hat{q}_b, \tau; \theta), \tag{1}$$

where $x_b$ ($u_b$) denotes the $b$-th labeled (unlabeled) samples in a minibatch $\mathcal{MX}$ ($\mathcal{MU}$). $\mathcal{A}(u_b)$ denotes the strongly augmented of $u_b$. The losses and other SSL algorithms, *i.e.* FlexMatch (Zhang et al., 2021) and FreeMatch (Wang et al., 2023b), are detailed in Appendix B.1 to B.3.

**Learning dynamics and its per-step decomposition.** Inspired by Ren & Sutherland (2025), we study how a single gradient update changes the model's confidence on an observation $x_o$. With $\pi_\theta(y \mid x)$ denoting the predicted class probability distribution, the learning dynamics become,

$$\Delta\theta \triangleq \theta^{t+1} - \theta^t = -\eta \cdot \nabla\mathcal{L}(f_\theta(x_b), y_b); \quad \Delta\log\pi^t(y|x_o) \triangleq \log\pi_{\theta^{t+1}}(y|x_o) - \log\pi_{\theta^t}(y|x_o). \tag{2}$$

where the update of $\theta$ during step $t \to t+1$ is given by one gradient update on the sample pair $(x_b, y_b)$ with learning rate $\eta$. $\mathcal{L}$ is the loss function, we use the cross-entropy loss $\mathbf{H}$ in our setting.

**Proposition 1** (Per-step decomposition of learning dynamics; Ren & Sutherland 2025)**.** *Let $\pi = \texttt{Softmax}(\mathbf{z})$ with $\mathbf{z} = g_\theta(x)$. Then the one-step learning dynamics decompose as*

$$\Delta\log\pi_\theta^t(y \mid x_o) = -\eta\mathcal{T}^t(x_o)\mathcal{K}^t(x_o, x_b)\mathcal{G}^t(x_b, y_b) + \mathcal{O}\left(\eta^2\|\nabla_\theta\mathbf{z}(x_b)\|_{op}^2\right), \tag{3}$$

*where $\mathcal{T}^t(x_o) = \nabla_z\log\pi_{\theta^t}(x_o) = I - \mathbf{1}\pi_{\theta^t}^\top(x_o)$ only depends on the model's current predicted probability, $\mathcal{K}^t(x_o, x_b) = (\nabla_\theta z(x_o)|_{\theta^t})(\nabla_\theta z(x_b)|_{\theta^t})^\top$ is the empirical neural tangent kernel*

*(eNTK, Jacot et al. 2018) of the model, the product of the model's gradients with respect to $x_o$ and $x_b$. $\mathcal{G}^t(x_b, y_b) = \nabla_{\mathbf{z}}\mathcal{L}(x_b, y_b)|_{\mathbf{z}^t}$ is the loss gradient. $\|\cdot\|^2_{op}$ denotes the spectral norm, which bounds the second-order remainder term.*

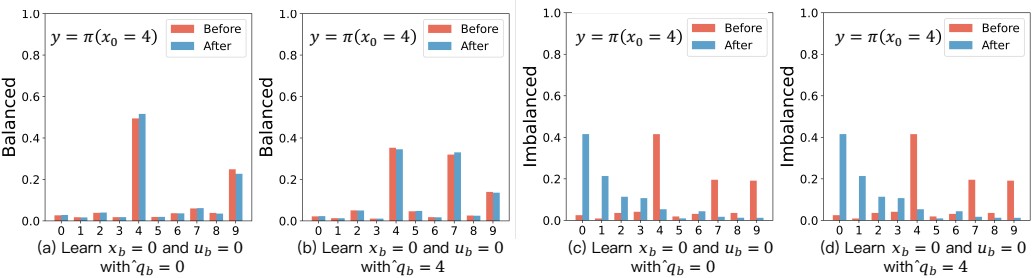

Figure 1: Accumulated influence in the MNIST experiment using a labeled sample $x_b = 0$ and an unlabeled sample $u_b = 0$ for training, with $x_o = 4$ for testing. (a) and (b) shows results from the Balanced experiment (MNIST), (c) and (d) from the Imbalanced experiment (MNIST-LT). (a) and (c) show the influence with accurate pseudo-labels, (b) and (d) with inaccurate pseudo-labels. In (a) and (b), the cumulative influence of pseudo-label authenticity is evident, with the false pseudo-label affecting predictions for similar samples (*e.g.*, probability of 9, 7 and 4). In (c) and (d), the class imbalance masks the influence of false pseudo-label authenticity due to class bias.

This decomposition characterizes how each update at $(x_b, y_b)$ influences predictions at $x_o$, forming the basis for our SSL analysis under class imbalance.

## 3 LEARNING DYNAMICS OF LONG-TAILED SEMI-SUPERVISED DEBIASING

### 3.1 LEARNING DYNAMICS OF SEMI-SUPERVISED LEARNING

In this section, we characterize the learning dynamics of the semi-supervised version of gradient descent (GD) for the FixMatch algorithm Eq. (1),

$$\Delta\theta \triangleq \theta^{t+1} - \theta^t = -\eta \cdot \left(\nabla\mathcal{L}_{sup}(f_\theta(\alpha(x_b)), y_b) + \nabla\mathcal{L}_{con}(f_\theta(\alpha(u_b)), f_\theta(\mathcal{A}(u_b)))\right);$$
$$\Delta f(x_o) \triangleq f_{\theta^{t+1}}(x_o) - f_{\theta^t}(x_o). \tag{4}$$

where $x_o$ denotes the observation data point, the update of $\theta$ during step $t \to t+1$ is given by one gradient update on the labeled sample pair $(x_b, y_b)$ and unlabeled sample $(u_b)$ with learning rate $\eta$. Previous work (Ren & Sutherland, 2025) showed how a single gradient update influences model predictions in supervised learning. We now examine whether such characterization extends to the semi-supervised setting. Since FixMatch (Sohn et al., 2020) update naturally consists of a supervised part $\mathcal{L}_{sup}$ and a consistency part $\mathcal{L}_{con}$, the gradient update can be decomposed accordingly. For an unlabeled sample $u_b$ with target $\hat{q}^t_b = \arg\max_c q^t_{b,c}$, where $q^t_b = \pi_{\theta^t}(\cdot \mid \alpha(u_b))$. The per-step learning dynamics of semi-supervised learning become

$$\Delta\log\pi^t(y|x_o) \triangleq \Delta\log\pi^{t,\mathrm{sup}}_\theta(y \mid x_o; x_b) + \Delta\log\pi^{t,\mathrm{con}}_\theta(y \mid x_o; u_b) \tag{5}$$

where $\Delta\pi^{t,\mathrm{sup}}_\theta$ denotes the influence caused by $x_b$ and $\Delta\pi^{t,\mathrm{con}}_\theta$ denotes the influence caused by $u_b$, respectively. Inspired by Definition 1, we now state the decomposition of the per-step influence in semi-supervised learning below:

**Proposition 2.** *For an labeled (unlabeled) sample $x_b$ ($u_b$) with target $y_b$ ($\hat{q}^t_b$). The one-step learning dynamics of SSL decompose as*

$$\Delta\log\pi^{t,\mathrm{sup}}_\theta(y \mid x_o; x_b) = -\eta\mathcal{T}^t(x_o)\mathcal{K}^t(x_o, \alpha(x_b))\mathcal{G}^t_{\mathrm{sup}}(\alpha(x_b), y_b) + \mathcal{O}\left(\eta^2\|\nabla_\theta\mathbf{z}(\alpha(x_b))\|^2_{op}\right)$$
$$\Delta\log\pi^{t,\mathrm{con}}_\theta(y \mid x_o; u_b) = -\eta\mathcal{T}^t(x_o)\mathcal{K}^t(x_o, \mathcal{A}(u_b))\mathcal{G}^t_{\mathrm{con}}(\mathcal{A}(u_b), \hat{q}^t_b) + \mathcal{O}\left(\eta^2\|\nabla_\theta\mathbf{z}(\mathcal{A}(u_b))\|^2_{op}\right) \tag{6}$$

*where $\mathcal{K}^t(x_o, \alpha(x_b))$ and $\mathcal{K}^t(x_o, \mathcal{A}(u_b))$ are eNTK evaluations of the logit network $\mathbf{z}(\cdot) = g_\theta(\cdot)$, with different inputs. $\mathcal{G}^t_{\mathrm{sup}}(\alpha(x_b), y_b) = \nabla_{\mathbf{z}}\mathcal{L}_{\mathrm{sup}}(\alpha(x_b), y_b)|_{\mathbf{z}^t}$ and $\mathcal{G}^t_{\mathrm{con}}(\hat{q}_b, \mathcal{A}(u_b)) = \nabla_{\mathbf{z}}\mathcal{L}_{\mathrm{con}}(\hat{q}_b, \mathcal{A}(u_b))|_{\mathbf{z}^t}$, respectively.*

As shown in Proposition 2, each update of $\theta$ in FixMatch decomposes into a supervised part driven by $(x_b, y_b)$ and a consistency part driven by $(u_b, \hat{q}_b^t)$. While this decomposition captures the per-step influence on $\pi_\theta(y \mid x_o)$, in practice training consists of many such steps, and the accumulated effect is governed by the iterative interaction between labeled and unlabeled updates. The detailed technical proofs are deferred to Appendix C.1.

**Accumulated influence and a demonstration on MNIST.** To demonstrate this, we train a WRN-28-2 on MNIST and visualize the accumulated influence in Figure 1. In Figure 1(a), when $\hat{q}_b$ is correct, the consistency term reinforces the supervised signal, gradually pulling the prediction of $x_o$ toward the correct class, *i.e.*, $q_{b,4_\uparrow}$ and $q_{b,9_\downarrow}$, consistent with the constructive dynamics implied by Eq. (6). In contrast, when $\hat{q}_b$ is incorrect (Figure 1(b)), the consistency update exerts the opposite effect, *i.e.*, $q_{b,4_\downarrow}, q_{b,7_\uparrow}$ and $q_{b,9_\downarrow}$, systematically reducing the correct probability of $x_o$. This illustrates how pseudo-label errors, even if small at each step, can accumulate across iterations into a negative loop. The Figure 1(c) and (d) show that under class imbalance, such accumulated influence can drive the classifier to consistently predict the majority class (here $q_{b,0} > q_{b,4}$), regardless of the true label. This confirms the implication of our dynamics analysis: in SSL, the imbalance influence of labeled data is passed to the pseudo-labels through the classifier, so imbalance bias can be amplified rather than averaged out, leading to catastrophic bias.

## 3.2 Learning dynamics analysis of accumulated bias under class imbalance

The aforementioned phenomenon, together with the learning dynamics of the semi-supervised framework, illustrates how class imbalance accumulates into systematic bias. While per-update dynamics capture the influence of individual samples on predictions, they fall short of reflecting the global effect of imbalance. This motivates the search for an indicator that bridges class-imbalance bias with the underlying learning dynamics. Replacing the inputs $x_o$ with a task irrelevant baseline image $\mathcal{I}$, we can regard the Eq. (6) as such an attributing indicator (Sundararajan et al., 2017). To justify this choice, we analyze its theoretical properties in both linear and deep settings, and then incorporate it into the per-step influence decomposition.

**Baseline image and its invariance property.** For simplicity, we first consider a two-layer MLP with no bias in the first layer and a bias vector $\boldsymbol{b} \in \mathbb{R}^C$ in the output layer $h(x) = h^{(2)} \circ h^{(1)}(x)$, where $h^{(1)}(x) = \sigma(\boldsymbol{W}_1 x)$ and $h^{(2)} = \boldsymbol{W}_2 x + \boldsymbol{b}$. This setting allows us to isolate and examine the predicted class probability $\pi_\theta(\mathcal{I})$ of a baseline image. For a baseline image $\mathcal{I} \in \mathbb{R}^d$, we have

$$h(\mathcal{I}) = \boldsymbol{W}_2 h^{(1)}(\mathcal{I}) + \boldsymbol{b}. \tag{7}$$

In modern neural networks, the explicit bias term $\boldsymbol{b}$ is often absorbed into the normalization layer, *e.g.*, BatchNorm, LayerNorm, with other layers typically set without bias. Without loss of generality, we take BatchNorm as an example for analysis. Since the BatchNorm transformation can be equivalently viewed as an affine linear layer with learnable parameters, we may replace $h^{(2)}$ with a `BatchNorm(·)` layer, *i.e.*,

$$h(\mathcal{I}) = \mathtt{BatchNorm}\big(h^{(1)}(\mathcal{I})\big) = \frac{h^{(1)}(\mathcal{I}) - \mathbb{E}[h^{(1)}(\mathcal{I})]}{\sqrt{\mathrm{Var}[h^{(1)}(\mathcal{I})] + \epsilon}} \cdot \boldsymbol{W}_2 + \boldsymbol{b}. \tag{8}$$

where $\epsilon$ is a small positive constant that ensures numerical stability. The baseline image is typically a solid color image, which inherently lacks task-related patterns, see Appendix D.1 for more discussions. This representation shows that, for baseline images, the dependence of $h(\mathcal{I})$ on the input is effectively controlled only through the affine parameters $(\boldsymbol{W}_2, \boldsymbol{b})$ of the normalization layer. We now state the main results regarding the prediction $\pi_\theta(\mathcal{I})$ for such baseline images:

**Proposition 3** (Invariance of baseline image under affine normalization). *Let $\mathcal{I} = k \cdot \mathbf{1}_d$ be a solid color image, where $k \in \{0, 1, \ldots, 255\}$ and $\mathbf{1}_d \in \mathbb{R}^d$ is an all-one vector. Suppose the output of the first hidden transformation is normalized by a normalization layer (e.g., BatchNorm, InstanceNorm, or GroupNorm) with affine parameters $(\boldsymbol{W}_2, \boldsymbol{b})$. Then the logits $h(\mathcal{I})$ are independent of $k$ and reduce to*

$$h(\mathcal{I}) = \boldsymbol{b}, \quad \pi_\theta(\mathcal{I}) = \mathtt{Softmax}(\boldsymbol{b}). \tag{9}$$

One can immediately notice that $\pi_\theta(\mathcal{I})$ in Eq. (9) does not contain any term related to the pixel value $k$ of $\mathcal{I}$. This observation implies that the representation $\pi_\theta(\mathcal{I})$ of a baseline image is entirely

determined by the BatchNorm bias term $\boldsymbol{b}$, and is invariant to the actual pixel value $k$. The detailed technical proofs are deferred to Appendix C.2.

Building upon this invariance, we now establish a direct connection between the baseline image and the underlying class distribution. Specifically, for the classifier formulation in Eq. (8) and Eq. (9), we show that the logits of the baseline image encode the class-imbalance ratio present in the training data, thus providing a direct bridge between $\pi_\theta(\mathcal{I})$ and the class prior induced by the long-tailed distribution in training. We empirically validate this connection on CIFAR10-LT by analyzing the distribution of baseline logits: as shown in Figure 2, the baseline logits closely align with the empirical class prior. When we remove the bias term in our ablation model, this alignment vanishes, indicating that the baseline logits lose their responsiveness to the class prior.

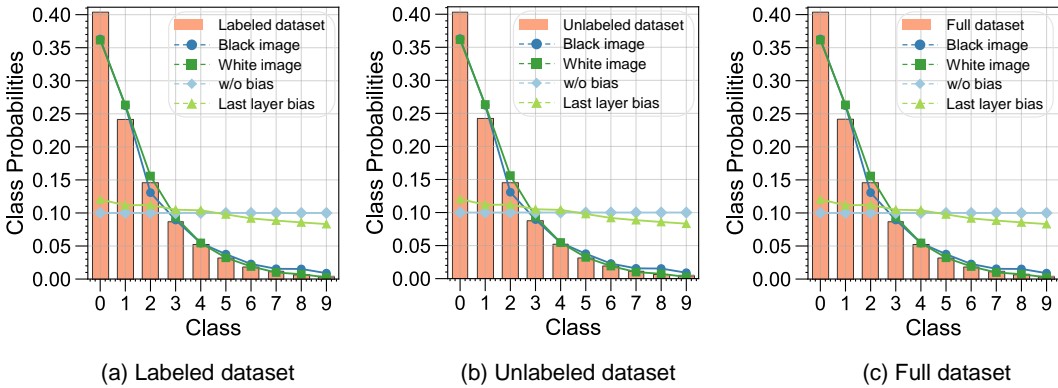

Figure 2: Class distributions and measured biaseddegree under $\gamma_l = 100$ and $\gamma_u = 100$. The bar plots show the class distributions for (a) labeled, (b) unlabeled, and (c) full datasets.

**Theorem 1** (Bias as the conditional distribution prior). *Assume the model $h(x)$ as characterized in Eq. (8) is trained using cross-entropy loss:*

$$\mathcal{L} = \mathbb{E}_{(x,y)}\big[ -y^\top \log \texttt{Softmax}(h(x))\big]. \tag{10}$$

*At a population risk minimizer $(\boldsymbol{W}_2^\star, \boldsymbol{b}^\star)$ we have*

$$\hat{p}^\star(x) = P(y \mid x), \qquad \hat{p}^\star(\mathcal{I}) = \texttt{Softmax}(\boldsymbol{b}^\star) = P\big(y \,\big|\, \tfrac{h^{(1)}(\mathcal{I}) - \mathbb{E}[h^{(1)}(\mathcal{I})]}{\sqrt{\mathrm{Var}[h^{(1)}(\mathcal{I})] + \epsilon}} = \boldsymbol{0}\big). \tag{11}$$

*For the baseline image $\mathcal{I}$ in Proposition 3, the baseline prediction thus coincides with the conditional class distribution at the normalized-zero feature state, capturing the class prior induced by the long-tailed training distribution. See the detailed to Appendix C.3.*

Thus, $\pi_\theta(\mathcal{I})$ serves as a natural proxy for the *accumulated bias* of the model, bridging the class imbalance in the training set to the learning dynamics of the classifier.

**Per-step influence decomposition of the baseline image.** Let $\pi_\theta(y|\cdot)$ denote the estimate of the underlying class prior. Then we can track the change in the model's confidence by observing $\log \pi_\theta(y|\mathcal{I})$. Then the learning dynamics on the baseline image become,

$$\Delta \log \pi^t(y|\mathcal{I}) \triangleq \log \pi_{\theta^{t+1}}(y|\mathcal{I}) - \log \pi_{\theta^t}(y|\mathcal{I}). \tag{12}$$

**Proposition 4.** *Let $\pi = \texttt{Softmax}(\mathbf{z})$ and $\mathbf{z} = g_\theta(x)$. The one-step dynamics on the baseline image decompose as*

$$\Delta \log \pi_\theta^t(y \mid \mathcal{I}; x) = -\eta \mathcal{T}^t(\mathcal{I})\mathcal{K}^t(\mathcal{I}, x)\mathcal{G}^t(x, y) + \mathcal{O}\big(\eta^2 \|\nabla_\theta \mathbf{z}(x)\|_{op}^2\big) \tag{13}$$

*where $\mathcal{T}^t(\mathcal{I}) = \nabla_{\mathbf{z}} \log \pi^t(\mathcal{I}) = I - \mathbf{1}\pi_{\theta^t}^T(\mathcal{I})$, $\mathcal{K}^t(\mathcal{I}, x) = \big(\nabla_\theta \mathbf{z}(\mathcal{I})\big|_{\theta^t}\big)\big(\nabla_\theta \mathbf{z}(x)\big|_{\theta^t}\big)^T$ is the eNTK of the logit network $\mathbf{z}$, $x$ can be $\alpha(x_b)$ or $\mathcal{A}(u_b)$, $y$ can be $y_b$ and $\alpha(u_b)$. See Appendix C.4 for more details.*

Compared with Proposition 2, the main difference is that the $\mathcal{T}^t(\mathcal{I})$ and $\mathcal{K}^t(\mathcal{I}, x)$ term. Since the baseline image $\mathcal{I}$ lies far from the data manifold, the coupling kernel $\mathcal{K}^t(\mathcal{I}, x)$ is typically small.

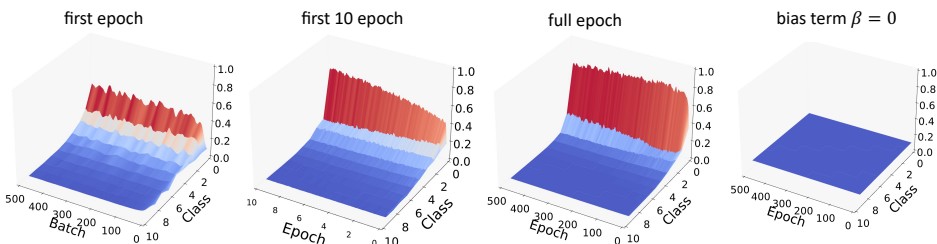

Figure 3: The change of logits's probability distribution $\pi_\theta(\mathcal{I})$ for the baseline image on CIFAR-10-LT. The left three panels depict the dynamics of reference logits under FixMatch: at epoch 1, epochs 1-10, and full epochs. The rightmost panel illustrates the dynamics after removing all bias terms.

Thus, the learning dynamics in Eq. (13) are mainly governed by the output-sensitivity term $\mathcal{T}^t(\mathcal{I})$ and the gradient signal $\mathcal{G}^t$, with the latter providing both the *energy* and *direction* for the model's adaptation. Under this formulation, the baseline image $\mathcal{I}$ serves as an indicator that isolates the model's global bias state. Tracking $\pi_\theta^t(\mathcal{I})$ over training therefore provides a direct and interpretable measurement of how class-level bias accumulates during semi-supervised learning. Therefore, as the number of labeled and unlabeled samples from the majority class increases, the output of $\pi_\theta^t(\mathcal{I})$ will be progressively squeezed into a biased long-tailed distribution. Even with $\mathcal{G}^t$ guiding the adaptation direction, this process can still be steered by the biased state encoded in $\pi_\theta^t(\mathcal{I})$, further amplifying the long-tailed shift, as illustrated in Figure 3.

## 4 DYNAMICS ANALYSIS OF LOGITS DEBIASING IN SEMI-SUPERVISED LEARNING

Analyzing the dynamics of logits debiasing methods in long-tailed semi-supervised learning is challenging because different algorithms such as Logits Adjustment, Reweighting, and Resampling employ distinct formulations. In this section, we propose a unified framework based on the per-step influence decomposition (Proposition 4). This framework enables us to analyze how these methods modify the update gradient flow, thereby influencing the model's bias evolution during training. We also introduce a pruning-based method, DyTrim, as a byproduct of our analysis. It can be integrated in a plug-and-play manner with other logits debiasing methods.

### 4.1 PER-STEP DECOMPOSITION OF LOGITS ADJUSTMENT

The typical logits debias method used during long-tail semi-supervised learning is logits adjustment (LA) (Menon et al., 2021), which introduces a class-dependent shift in the logits, expressed as:

$$\tilde{\pi}_\theta(y|x) = \text{Softmax}(\tilde{\mathbf{z}}(x)), \qquad \tilde{\mathbf{z}}(x) = g_\theta(x) - \lambda\phi, \tag{14}$$

where $\lambda \geq 0$ controls the adjustment strength and $\phi \in \mathbb{R}^C$ is estimates of the class priors. Thanks to the $\tilde{z}$ implemented in CDMAD (Lee & Kim, 2024), the resulting logits adjustment is almost identical to such simple subtraction, *i.e.*, $\tilde{z}(x) = g_\theta(x) - \log\pi$, where $\pi = \pi_\theta(\mathcal{I})$. Thus, the change of the model's prediction on the baseline image $\mathcal{I}$ can be represented as,

$$\Delta\log\tilde{\pi}_\theta^t(y \mid \mathcal{I}; x_b) = -\eta\,\mathcal{T}^t(\mathcal{I})\,\mathcal{K}^t(\mathcal{I}, x_b)\,\tilde{\mathcal{G}}_{LA}^t(x, y) \;+\; \mathcal{O}\big(\eta^2\|\nabla_\theta\tilde{\mathbf{z}}(x_b)\|_{\text{op}}^2\big). \tag{15}$$

where $\tilde{\mathcal{G}}_{LA}(x, y) = \pi_\theta^t(\alpha(u_b)|\mathcal{A}(u_b)) - \pi$ represents the influence of the adjusted logits. Compared with Proposition 4, the main difference is that the gradient term has been modified by class prior $\pi$, which allows us to answer *how does learning with debiasing affect the gradients for unlabeled samples?* When adjusting the model's logits by class prior, the gradient flow will ensure that the model compensates for the class imbalance during training. See more discussions in Appendix C.5. We also conducted experiments on CIFAR10-LT to demonstrate the effectiveness of this debiasing, as illustrated in Figure 3, which illustrates that the bias measured in the baseline image after applying LA to the CDMAD method is alleviated.

## 4.2 PER-STEP DECOMPOSITION OF REWEIGHTING

Reweighting is another prevalent debiasing technique in long-tail semi-supervised learning (Lai et al., 2022), which introduces class-dependent weights in the loss function, expressed as:

$$\mathcal{L}_{sup}^{rw} = \sum_{k=1}^{C} w_k^l \mathcal{L}_{sup}(\alpha(x_b^k); \theta); \quad \mathcal{L}_{con}^{rw} = \sum_{k=1}^{C} w_k^u \mathcal{L}_{con}(\mathcal{A}(u_b^k), \hat{q}_b, \tau; \theta); \tag{16}$$

where $w_k^l$ ($w_k^u$) is the weight of the $k$-th class in labeled (unlabeled) samples. For simplicity, we assume the class weight distributions are consistent between labeled and unlabeled data, *i.e.*, $w_k^l$ and $w_k^u$ follow the same proportional relationship and remain fixed during training. Under this reweighting scheme, the gradient signals for both supervised and consistency terms are scaled by their respective class weights. Hence, we can decompose the learning dynamics for reweighting similarly to Eq. (15),

$$\Delta \log \pi_\theta^{t,rw}(y \mid \mathcal{I}; x) = -\eta \mathcal{T}^t(\mathcal{I}) \tilde{\mathcal{K}}_{rw}^t(\mathcal{I}, x; w^c) \tilde{\mathcal{G}}_{rw}^t(x, y; w^c) + \mathcal{O}\left(\eta^2 |\nabla \theta \mathbf{z}(x)| \text{op}^2\right) \tag{17}$$

where $\tilde{\mathcal{K}}_{rw}^t(\mathcal{I}, x; w^c) = w^c \mathcal{K}_{rw}^t(\mathcal{I}, x)$ and $\tilde{\mathcal{G}}_{rw}^t(x, y; w^c) = w^c \mathcal{G}^t(x, y)$. Thus, reweighting acts by scaling both the similarity kernel and the gradient term with the class weight $w^c$. Intuitively, this modulates the strength of interaction between samples and the magnitude of their gradients in a class-dependent manner: samples from classes with larger $w^c$ exert a stronger influence on the update of $\theta$, while those from classes with smaller $w^c$ contribute less. When $w^c$ is designed as a function of class frequency (*e.g.*, inverse frequency), this mechanism increases the effective contribution of under-represented classes and attenuates that of head classes. See more discussions in Appendix C.5.

## 4.3 DYTRIM: A BASELINE IMAGE GUIDED DATA PRUNING FRAMEWORK FOR LTSSL

Under the per-step influence framework of Proposition 4, logits adjustment and reweighting reshape the gradient flow by modifying the update direction or magnitude, while resampling acts directly on the data distribution by changing the frequency with which different classes enter training. Yet all these methods leave the sample set itself intact at each step and ignore the heterogeneous per-step utility of individual samples, allowing redundant head-class examples to continue dominating the learning dynamics. This motivates debiasing at the data-selection level, where dynamically controlling which samples participate in each update provides a more direct mechanism for mitigating accumulated bias in LTSSL, as illustrated in Figure 4.

**Per-step decomposition of dynamic pruning.** Differs from logits adjustment, reweighting, or resampling, dynamic pruning directly alters the set of samples that participate in each gradient update, instead of modifying the loss or sampling distribution. We define step-dependent scoring functions $\mathcal{H}_t^l(\cdot)$ for labeled samples $\mathcal{X}$ and $\mathcal{H}_t^u(\cdot)$ for unlabeled samples $\mathcal{U}$, which dynamically quantify sample utility at training step $t$. For the dynamic pruning process, samples are discarded by the step-dependent pruning probabilities $\mathcal{P}_t^l$ and $\mathcal{P}_t^u$:

$$\mathcal{P}_t^l(x; \mathcal{H}_t^l) = \mathbb{1}(\mathcal{H}_t^l(x), \bar{\boldsymbol{H}}_t^l); \quad \text{and} \quad \mathcal{P}_t^u(u; \mathcal{H}_t^u) = \mathbb{1}(\mathcal{H}_t^u(u), \bar{\boldsymbol{H}}_t^u), \tag{18}$$

where $\bar{\boldsymbol{H}}_t^l$ and $\bar{\boldsymbol{H}}_t^u$ are adaptive thresholds, $\mathbb{1}(\cdot, \cdot)$ is the indicator function. Under this dynamic pruning mechanism, the one-step decomposition of dynamic pruning decomposes as

$$\Delta \log \pi_\theta^{t,\text{prune}}(y \mid \mathcal{I}; x) = -\eta \mathcal{T}^t(\mathcal{I}) \mathcal{K}^t(\mathcal{I}, x) \tilde{\mathcal{G}}_{dytr}^t(x, y) + \mathcal{O}\left(\eta^2 |\nabla \theta \mathbf{z}(x)| \text{op}^2\right)$$
$$\tilde{\mathcal{G}}_{dytr}^t(x, y) = \mathcal{P}_t(x) \mathcal{G}^t(x, y) \tag{19}$$

where

$$\mathcal{P}_t(x) = \left\{ \begin{array}{ll} \mathcal{P}_t^l(x; \mathcal{H}_t^l) & x \in \mathcal{X}, \\ \mathcal{P}_t^u(u; \mathcal{H}_t^u) & x \in \mathcal{U}, \end{array} \right. \tag{20}$$

This decomposition shows that dynamic pruning reshapes the update dynamics by gating sample participation through $\mathcal{P}_t^l$ and $\mathcal{P}_t^u$, effectively zeroing out the kernel–gradient interactions $\mathcal{K}^t(\mathcal{I}, x) \mathcal{G}^t(x, y)$ of low-utility samples. In contrast to logits adjustment and reweighting, which only alter gradient signals, or resampling, which changes the sampling measure, pruning directly removes redundant head-class examples and underlearned unlabeled ones from the optimization

path, thereby reallocating the model's effective update budget toward samples that meaningfully influence bias correction. Although the kernel $\mathcal{K}^t(\mathcal{I}, x)$ itself remains unchanged, its operational contribution becomes $\mathbb{E}_{x \sim p}[\mathcal{P}_t(x)\mathcal{K}^t(\mathcal{I}, x)]$, selectively amplifying informative interactions while suppressing those that drive long-tailed drift. This sample-level intervention yields a more direct and fine-grained control of the learning dynamics than existing debiasing strategies.

Building on this perspective, we now instantiate how dynamic pruning is implemented in practice. We introduce DyTrim, a baseline-guided dynamic pruning framework designed to accommodate the distributional mismatch that real-world LTSSL typically exhibits between labeled and unlabeled data. Since such mismatch renders a single participation rule inadequate, DyTrim employs two complementary pruning mechanisms, one tailored to the long-tailed labeled set and the other to the noisy and imbalance-unknown unlabeled set. See more details about Appendix C.6.

**Dynamic pruning for labeled data.** Since the labeled data follow a long-tailed class distribution, we design a class-aware pruning policy $\mathcal{P}_t^l$ guided by $\pi_\theta(\mathcal{I})$. Critically, the classifier's pseudo-labels are primarily influenced by the labeled samples, which introduce bias toward majority classes. Since Proposition 3 shows that the baseline image has invariance to solid-color intensity, from first principles, we leverage the logits from a **black image** $\mathcal{I}$ to calibrate pruning probabilities. Given the labeled dataset $\mathcal{X}$ in the $t$-th epoch, a class-aware pruning probability is assigned to each sample based on its score, which is formulated as:

$$\mathcal{P}_t^l(x_b^n) = \begin{cases} 1 & \mathcal{H}_t^l(x_b^n) \in \boldsymbol{H}_{\prec r_c, t}^l, \\ 0 & \mathcal{H}_t^l(x_b^n) \notin \boldsymbol{H}_{\prec r_c, t}^l, \end{cases} \tag{21}$$

where $\boldsymbol{H}_{\prec r_c, t}^l$ denotes the $r_c \times N_c$ smallest scoring values of the class $c$ and $r_c = \pi_\theta(\mathcal{I})_c$ is the class-aware pruning probability. The labeled scoring function $\mathcal{H}_t^l(x_b^n)$ is defined using the supervised loss $\mathcal{L}_{sup}(x_b^n, y_b^n)$ to quantify sample utility. See more details about Appendix E.1.

**Dynamic pruning for unlabeled data.** While the distribution of the label of the unlabeled data and its imbalance ratio $\gamma_u$ are unknown. To address the uncertainty and bias of pseudo-labels, we design a label-insensitive soft pruning policy $\mathcal{P}_t^u$ inspired by (Qin et al., 2024), which introduces randomness and gradient scaling into the pruning process. Specifically, for an unlabeled dataset $\mathcal{U}$ at the $t$-th epoch, a pruning probability is assigned to each sample based on its score, which is formulated as:

$$\mathcal{P}_t^u(u_b^m) = \begin{cases} r & \mathcal{H}_t^u(u_b^m) < \bar{\mathcal{H}}_t^m \text{ and } p^*(u_b^m) \geq \tau, \\ 0 & \mathcal{H}_t^u(u_b^m) \geq \bar{\mathcal{H}}_t^u \text{ or } p^*(u_b^m) < \tau, \end{cases} \tag{22}$$

where $\bar{\mathcal{H}}_t^u$ is the adaptive threshold and $r$ is a randomized pruning rate, $\tau$ is the confidence threshold $\tau$ and $p^*(u_b^m) = \max(\text{softmax}(g_\theta^*(\alpha(u_b^m))))$ denote the debiased pseudo-label confidence. See more details about Appendix E.2.

## 5 EXPERIMENT

In this section, we conducted comprehensive experiments to verify the effectiveness of the proposed DyTrim on CIFAR10-LT, CIFAR100-LT (Cui et al., 2019), STL10-LT (Kim et al., 2020), and ImageNet-127 (Deng et al., 2009; Huh et al., 2016) datasets. Due to limited space, we defer the detailed experimental settings and additional experiments to the Appendix G.

### 5.1 RESULTS ON CIFAR10/100-LT, STL10-LT AND IMAGENET-LT

Under the consistent condition where $\gamma_u$ is known and matched to $\gamma_l$, the results in Table 1 show that CISSL algorithms consistently outperform their vanilla SSL counterparts by mitigating class imbalance while effectively exploiting unlabeled data. Among them, the proposed DyTrim achieves the best performance across all imbalance ratios. Compared with the state-of-the-art CDMAD, DyTrim improves bACC by 1.2% and GM by 1.4% on average, without incurring additional computational overhead. Furthermore, when integrated into FlexMatch and FreeMatch, DyTrim yields substantial improvements, boosting bACC/GM by 2–3% on average. Table 2 evaluates the methods on CIFAR-100-LT, which involves more classes and a stronger imbalance. The results demonstrate that DyTrim consistently outperforms all competing approaches under this more challenging setting.

Table 1: Comparison of bACC/GM on CIFAR-10-LT under different imbalance ratio $\gamma = \gamma_l = \gamma_u$, where $\gamma_u$ is assumed to be known. "*" indicates our own implementation.

| Base SSL Algorithm | Debiasing Strategy | $\gamma = 50$ | | $\gamma = 100$ | | $\gamma = 150$ | |
|---|---|---|---|---|---|---|---|
| | | bACC | GM | bACC | GM | bACC | GM |
| Vanilla | | 65.2 ±0.05 | 61.1 ±0.09 | 58.8 ±0.13 | 58.2 ±0.11 | 55.6 ±0.43 | 44.0 ±0.98 |
| Re-sampling | | 64.3 ±0.48 | 60.6 ±0.67 | 55.8 ±0.47 | 45.1 ±0.30 | 52.2 ±0.05 | 38.2 ±1.49 |
| LDAM-DRW | | 68.9 ±0.07 | 67.0 ±0.08 | 62.8 ±0.17 | 58.9 ±0.60 | 57.9 ±0.20 | 50.4 ±0.30 |
| cRT | | 67.8 ±0.13 | 66.3 ±0.15 | 63.2 ±0.45 | 59.9 ±0.40 | 59.3 ±0.10 | 54.6 ±0.72 |
| FixMatch | FixMatch | 79.2 ±0.33 | 77.8 ±0.36 | 71.5 ±0.72 | 66.8 ±1.51 | 68.4 ±0.15 | 59.9 ±0.43 |
| | DARP+cRT | 85.8 ±0.43 | 85.6 ±0.56 | 82.4 ±0.26 | 81.8 ±0.17 | 79.6 ±0.42 | 78.9 ±0.35 |
| | CReST+LA | 85.6 ±0.36 | 81.9 ±0.45 | 81.2 ±0.70 | 74.5 ±0.99 | 71.9 ±2.24 | 64.4 ±1.75 |
| | ABC | 85.6 ±0.26 | 85.2 ±0.29 | 81.1 ±1.14 | 80.3 ±1.29 | 77.3 ±1.25 | 75.6 ±1.65 |
| | CoSSL | 86.8 ±0.30 | 86.6 ±0.25 | 83.2 ±0.49 | 82.7 ±0.60 | 80.3 ±0.55 | 79.6 ±0.57 |
| | SAW+LA | 86.2 ±0.15 | 83.9 ±0.35 | 80.7 ±0.15 | 77.5 ±0.21 | 73.7 ±0.06 | 71.2 ±0.17 |
| | Adsh | 83.4 ±0.06 | 82.9 ±0.13 | 76.5 ±0.35 | 74.8 ±0.34 | 71.5 ±0.30 | 68.8 ±0.35 |
| | DebiasPL | 85.6 ±0.20 | 85.2 ±0.23 | 80.6 ±0.50 | 79.9 ±0.57 | 76.6 ±0.12 | 75.8 ±0.71 |
| | UDAL | 86.5 ±0.29 | 86.2 ±0.26 | 81.4 ±0.39 | 80.6 ±0.38 | 77.9 ±0.33 | 75.8 ±0.71 |
| | L2AC | 86.6 ±0.31 | 86.7 ±0.30 | 82.1 ±0.57 | 81.5 ±0.64 | 77.6 ±0.53 | 75.8 ±0.71 |
| | CDMAD | 87.3 ±0.12 | 87.0 ±0.15 | 83.6 ±0.46 | 83.1 ±0.57 | 80.8 ±0.86 | 79.9 ±1.07 |
| | **DyTrim** | **88.0** ±0.31 | **87.8** ±0.32 | **84.8** ±0.48 | **84.4** ±0.51 | **82.0** ±0.09 | **81.3** ±0.03 |
| FlexMatch | FlexMatch* | 72.6 ±0.72 | 70.2 ±0.88 | 67.7 ±0.73 | 63.6 ±1.27 | 62.6 ±0.63 | 56.1 ±1.13 |
| | CDMAD* | 74.4 ±0.82 | 73.0 ±1.12 | 68.4 ±0.46 | 66.8 ±0.53 | 67.0 ±0.52 | 63.2 ±0.44 |
| | **DyTrim** | **77.2** ±0.42 | **76.2** ±0.44 | **70.7** ±0.49 | **67.8** ±0.70 | **68.6** ±0.22 | **66.3** ±0.07 |
| FreeMatch | FreeMatch* | 71.9 ±0.24 | 69.4 ±0.61 | 65.7 ±0.18 | 60.9 ±0.69 | 62.5 ±0.12 | 57.3 ±0.53 |
| | CDMAD* | 74.7 ±0.64 | 73.6 ±1.23 | 69.9 ±0.65 | 68.2 ±0.74 | 66.2 ±0.27 | 63.2 ±0.44 |
| | **DyTrim** | **76.9** ±0.45 | **75.9** ±0.52 | **72.3** ±0.12 | **71.4** ±0.57 | **69.4** ±0.35 | **67.5** ±0.63 |

As shown in Table 3, DyTrim consistently outperforms prior techniques such as CDMAD on the large-scale ImageNet-LT benchmark (Liu et al., 2019), demonstrating its complementary benefits rather than merely overlapping with existing rebalancing approaches. See more details about large-resolution (224 × 224) in Appendix H.4. Under the inconsistent condition where $\gamma_u$ was unknown and mismatched to $\gamma_l$, the results in Table 4 show that DyTrim remains the most effective method overall. When the labeled and unlabeled data distributions deviate, DyTrim consistently outperforms CDMAD on both CIFAR-10-LT and STL-10-LT. See more details in Appendix H.2 for the dynamics of baseline image logits during training.

Table 2: Comparison of bACC on CIFAR-100-LT under different imbalance ratio, where $\gamma_u$ is assumed to be known. "*" indicates our own implementation.

| Base SSL Alogrithm | Debiasing Strategy | $\gamma = 20$ | $\gamma = 50$ | $\gamma = 100$ |
|---|---|---|---|---|
| FixMatch | FixMatch | 49.6 ±0.78 | 42.1 ±0.33 | 37.6 ±0.48 |
| | DARP | 50.8 ±0.77 | 43.1 ±0.54 | 38.3 ±0.47 |
| | DARP+cRT | 51.4 ±0.68 | 44.9 ±0.54 | 40.4 ±0.78 |
| | CReST | 51.8 ±0.12 | 44.9 ±0.50 | 40.1 ±0.65 |
| | CReST+LA | 52.9 ±0.07 | 47.3 ±0.17 | 42.7 ±0.70 |
| | ABC | 53.3 ±0.79 | 46.7 ±0.26 | 41.2 ±0.06 |
| | CoSSL | 53.9 ±0.78 | 47.6 ±0.26 | 43.0 ±0.61 |
| | UDAL | 54.1 ±0.23 | 48.0 ±0.56 | 43.7 ±0.41 |
| | CPE | 52.4 ±0.17 | 45.6 ±0.68 | 39.9 ±0.40 |
| | CDMAD | 54.3 ±0.44 | 48.8 ±0.75 | 44.1 ±0.29 |
| | **DyTrim** | **55.5** ±0.53 | **50.8** ±0.80 | **44.8** ±0.27 |
| FlexMatch | FlexMatch* | 36.5 ±0.51 | 29.6 ±0.35 | 25.8 ±0.79 |
| | CDMAD* | 39.2 ±0.47 | 31.9 ±0.46 | 27.0 ±0.66 |
| | **DyTrim** | **40.9** ±0.09 | **33.5** ±0.21 | **29.8** ±0.67 |
| FreeMatch | FreeMatch* | 35.9 ±0.69 | 31.3 ±0.65 | 24.5 ±0.66 |
| | CDMAD* | 36.9 ±0.96 | 32.8 ±0.93 | 28.0 ±0.68 |
| | **DyTrim** | **39.0** ±0.61 | **33.4** ±0.70 | **29.8** ±0.09 |

## 5.2 RESULTS ON VIT BACKBONES

In addition, Table 5 highlights the performance of various algorithms under both consistent and inconsistent imbalance settings with ViT backbones. On CIFAR-10-LT, DyTrim yields the best results, improving bACC 0.6% over CDMAD and nearly 4% over FixMatch when $\gamma_l = \gamma_u = 100$. Under the inconsistent condition, DyTrim maintains a clear margin, surpassing CDMAD almost 2%. On CIFAR-100-LT, although the absolute accuracies are lower due to the increased diffi-

Table 3: Comparison of bACC on ImageNet-LT.

| Algorithm | ImageNet-LT |
|---|---|
| FixMatch* | 20.0 |
| w/+CDMAD* | 35.4 |
| w/+DyTrim | **37.2** |

Table 4: Comparison of bACC/GM on CIFAR-10-LT and STL-10-LT under different imbalance ratio $\gamma_l \neq \gamma_u$, where $\gamma_u$ is assumed to be unknown. "*" indicates our own implementation.

| Base SSL Algorithm | Debiasing Strategy | CIFAR-10-LT ($\gamma_l = 100$, $\gamma_u =$ Unknown) | | | | STL-10-LT ($\gamma_u =$ Unknown) | | | |
| | | $\gamma_u = 50$ | | $\gamma_u = 150$ | | $\gamma_l = 10$ | | $\gamma_l = 20$ | |
| | | bACC | GM | bACC | GM | bACC | GM | bACC | GM |
|---|---|---|---|---|---|---|---|---|---|
| FixMatch | FixMatch | 73.9 ±0.25 | 70.5 ±0.52 | 69.6 ±0.60 | 62.6 ±1.11 | 72.9 ±0.09 | 69.6 ±0.01 | 63.4 ±0.21 | 52.6 ±0.09 |
| | DARP | 77.3 ±0.17 | 75.5 ±0.21 | 72.9 ±0.24 | 69.5 ±0.18 | 77.8 ±0.33 | 76.5 ±0.40 | 69.9 ±1.77 | 65.4 ±3.07 |
| | DARP+LA | 82.3 ±0.32 | 81.5 ±0.29 | 78.9 ±0.23 | 77.7 ±0.06 | 78.6 ±0.30 | 77.4 ±0.40 | 71.9 ±0.49 | 68.7 ±0.51 |
| | DARP+cRT | 82.7 ±0.21 | 82.3 ±0.25 | 80.7 ±0.44 | 80.2 ±0.61 | 79.3 ±0.23 | 78.7 ±0.21 | 74.1 ±0.61 | 73.1 ±1.21 |
| | ABC | 82.7 ±0.64 | 82.0 ±0.76 | 78.4 ±0.87 | 77.2 ±1.07 | 79.1 ±0.46 | 78.1 ±0.57 | 73.8 ±0.15 | 72.1 ±0.15 |
| | SAW | 79.8 ±0.25 | 79.1 ±0.32 | 74.5 ±0.97 | 72.5 ±1.37 | 78.3 ±0.25 | 77.0 ±0.19 | 71.9 ±0.81 | 69.0 ±0.81 |
| | SAW+LA | 82.9 ±0.38 | 82.6 ±0.38 | 79.1 ±0.81 | 78.6 ±0.91 | 79.4 ±0.26 | 78.4 ±0.17 | 73.9 ±0.91 | 71.8 ±0.99 |
| | SAW+cRT | 81.6 ±0.38 | 81.3 ±0.32 | 77.6 ±0.40 | 77.1 ±0.41 | 78.9 ±0.22 | 77.8 ±0.14 | 72.3 ±0.86 | 69.5 ±0.83 |
| | CPE | 86.2 ±0.26 | 85.9 ±0.33 | 82.4 ±0.49 | 82.1 ±0.53 | 79.0 ±0.05 | 78.7 ±0.54 | 77.0 ±0.73 | 76.1 ±0.68 |
| | CDMAD | 85.7 ±0.36 | 85.3 ±0.38 | 82.3 ±0.23 | 81.8 ±0.29 | 79.9 ±0.23 | 78.9 ±0.38 | 75.2 ±0.40 | 73.5 ±0.31 |
| | **DyTrim** | **86.4** ±0.43 | **86.0** ±0.43 | **83.8** ±0.34 | **83.4** ±0.33 | **80.7** ±0.64 | **79.8** ±0.70 | **77.9** ±1.04 | **76.7** ±1.26 |
| FlexMatch | FlexMatch* | 67.7 ±0.67 | 62.8 ±0.65 | 63.0 ±0.77 | 56.3 ±1.70 | 62.1 ±0.29 | 60.8 ±0.43 | 56.9 ±0.90 | 51.4 ±0.81 |
| | CDMAD* | 69.2 ±0.22 | 67.0 ±0.11 | 67.0 ±1.69 | 63.4 ±0.91 | 65.5 ±1.05 | 63.7 ±1.02 | 62.4 ±1.05 | 60.5 ±0.99 |
| | **DyTrim** | **72.5** ±0.39 | **70.7** ±0.45 | **70.3** ±1.01 | **67.4** ±0.21 | **68.0** ±0.94 | **66.4** ±0.85 | **63.9** ±0.16 | **61.7** ±0.28 |
| FreeMatch | FreeMatch* | 69.3 ±0.99 | 65.4 ±1.45 | 63.5 ±0.76 | 55.7 ±0.77 | 63.9 ±0.77 | 62.0 ±0.90 | 59.0 ±1.43 | 57.6 ±0.67 |
| | CDMAD* | 71.0 ±0.98 | 69.0 ±1.05 | 67.1 ±0.96 | 64.3 ±0.99 | 66.1 ±0.32 | 63.8 ±0.97 | 61.5 ±0.47 | 59.5 ±0.63 |
| | **DyTrim** | **72.3** ±0.69 | **71.1** ±1.23 | **69.9** ±0.15 | **67.4** ±0.37 | **68.0** ±0.64 | **66.5** ±1.20 | **64.6** ±0.77 | **62.7** ±1.16 |

culty, DyTrim still matches or slightly improves upon CDMAD, while consistently outperforming FixMatch. Additional experimental results are provided in Appendix H.

Table 5: Comparison of bACC/GM on CIFAR-10-LT and CIFAR-100-LT with TinyViT under different imbalance ratio, where $\gamma_u$ is assumed to be known. "*" indicates our own implementation.

| Base SSL Algorithm | Debiasing Strategy | CIFAR-10-LT ($\gamma_l = 100$) | | | | CIFAR-100-LT ($\gamma_l = 100$) | |
| | | $\gamma_u = 100$ | | $\gamma_u = 150$ | | $\gamma_u = 100$ | |
| | | bACC | GM | bACC | GM | bACC | GM |
|---|---|---|---|---|---|---|---|
| FixMatch | FixMatch* | 45.5 ±0.14 | 30.0 ±0.41 | 45.3 ±0.12 | 28.9 ±0.96 | 23.2 ±0.13 | 5.7 ±0.33 |
| | CDMAD* | 48.7 ±0.49 | **40.5** ±0.26 | 45.4 ±0.13 | **39.9** ±0.10 | 24.0 ±0.15 | **9.0** ±0.77 |
| | **DyTrim** | **49.3** ±0.47 | 40.3 ±0.36 | **47.3** ±0.12 | 39.7 ±0.57 | **24.1** ±0.22 | 8.9 ±0.15 |

## 5.3 Scalability Evaluation of DyTrim

DyTrim exhibited robust extensibility as a universal plug-in component, consistently boosting performance across diverse SSL frameworks (CDMAD/CCL), datasets (CIFAR/STL10-LT), and imbalance ratios ($\gamma = 1 \sim 150$), as shown in Table 6. Notably, it achieved up to +1.4% (CDMAD on CIFAR10-LT) and +2.7% (STL10-LT, $\gamma_l$=20) gains without architecture-specific tuning, validating its versatility in semi-supervised long-tailed scenarios.

Table 6: Comparison of bACC with two SOTA algorithms with and without DyTrim on CIFAR-10, CIFAR-100, and STL-10. ↓ and ↑ indicate improvements or degradations over the baseline.

| Dataset | Imbalance ratio | FixMatch+ | | | FixMatch+ | | |
| | | CDMAD | CDMAD+**DyTrim** | Gain | CCL | CCL+**DyTrim** | Gain |
|---|---|---|---|---|---|---|---|
| CIFAR10-LT | $\gamma_l = \gamma_u = 100$ | 83.6 ±0.46 | **84.8** ±0.48 | ↑1.2 | 86.2 ±0.35 | **86.7** ±0.39 | ↑0.5 |
| | $\gamma_l = \gamma_u = 150$ | 80.8 ±0.86 | **82.0** ±0.09 | ↑1.2 | 84.0 ±0.21 | **84.0** ±0.26 | ↑0.0 |
| | $\gamma_l = 100, \gamma_u = 1$ | 87.5 ±0.46 | **88.9** ±0.88 | ↑1.4 | 93.9 ±0.12 | **94.1** ±0.17 | ↑0.2 |
| CIFAR100-LT | $\gamma_l = \gamma_u = 20$ | 54.3 ±0.44 | **55.5** ±0.53 | ↑1.2 | 57.5 ±0.16 | **58.1** ±0.49 | ↑0.6 |
| STL10-LT | $\gamma_l = 10$ | 79.9 ±0.23 | **80.7** ±0.64 | ↑1.2 | 84.8 ±0.15 | **85.1** ±0.33 | ↑0.3 |
| | $\gamma_l = 20$ | 75.2 ±0.40 | **77.9** ±1.04 | ↑2.7 | 83.1 ±0.18 | **83.3** ±0.40 | ↑0.2 |

## 6 Conclusion

In this work, we provide a theoretical characterization of class bias in LTSSL through an in-depth analysis of the learning dynamics. We derive a step-wise decomposition of logit updates, demonstrating how class imbalance dominates predictions and how debiasing methods, such as logit adjustment, reweighting, and resampling. Our theoretical insights bridge the gap between existing methods and their effect on gradient dynamics, highlighting the critical role of sample-level interventions. Based on this foundation, we introduce DyTrim, a dynamic pruning framework that mitigates class imbalance by reallocating gradient budgets. Empirical results across multiple benchmarks and SSL methods demonstrate that DyTrim consistently improves performance.

## ACKNOWLEDGEMENT

This work was supported by the Beijing Natural Science Foundation under Grant (No.L231005), and by the National Key Research and Development Program of China under Grant (No.2024YFB3312200). Yue Cheng would like to thank Bochen Lyu, Qianying Tang, and Xiang Wei for the useful discussion.

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

APPENDIX

# A RELATED WORK

## A.1 MORE ABOUT MECHANISMS OF LONG-TAILED DEBIASING

This paper considers learning dynamics to study the debiasing mechanisms of SSL algorithms. We briefly introduce differences between the settings considered here and those in previous works. For debiasing on long-tailed learning, Menon et al. (2021) considered a unified framework for debiasing from the perspective of logits adjustment, which requires statistical label frequency. CCL (Zhou et al., 2024) considered debiasing from an information-theoretical lens. LCGC (Xing et al., 2025) used gradient flow to analyze the debiasing process. However, these methods only elucidate the model's behavior from an ad hoc perspective. We aim to develop a more comprehensive framework that enables a principle-based lens of the bias generation mechanisms inherent in long-tailed semi-supervised learning.

## A.2 MORE ABOUT SEMI-SUPERVISED LEARNING

Modern SSL methods typically integrate diverse strategies for exploiting unlabeled data, such as entropy minimization (Zhou et al., 2024), consistency regularization (Sohn et al., 2020), and contrastive learning (Zhou et al., 2024; Lee et al., 2022). Among them, most SSL approaches rely on selecting reliable pseudo-labels during training. FixMatch (Sohn et al., 2020) adopts a fixed confidence threshold of 0.95, whereas FlexMatch (Zhang et al., 2021) adapts thresholds per class based on learning difficulty and training progress. FreeMatch (Wang et al., 2023b) integrates global and local adjustments with a class-fairness regularizer to promote prediction diversity, while Soft-Match (Chen et al., 2023) employs a soft thresholding scheme that reweights samples to balance quantity and quality. In contrast, our method bypasses threshold tuning altogether and directly enforces class-balanced pseudo-labeling through dynamic pruning.

## A.3 MORE ABOUT LONG-TAILED SEMI-SUPERVISED DEBIASING

Existing debiasing methods for LTSSL dominantly rely on consistent distribution assumptions (Guo & Li, 2022; Lee et al., 2021) and logit adjustment strategies (Wei & Gan, 2023). Notable approaches include CReST (Wei et al., 2021), which focuses on minority classes through selective self-training, and CoSSL (Cai et al., 2021), which balances representations using tail-class feature augmentation. Recent advances, like BaCon (Feng et al., 2024), utilize contrastive learning for balanced features, while SMCLP (Du et al., 2024) exploits collaborative label-instance correlations, and CPE (Ma et al., 2024) employs multiple expert classifiers. Innovative methods such as InPL (Yu et al., 2023) and DebiasMatch (Wang et al., 2022) move beyond traditional pseudo-labeling; InPL uses energy scores to detect reliable inliers, whereas DebiasMatch applies adaptive debiasing with a marginal loss to reduce long-tailed pseudo-label bias. Despite these advances, LTSSL techniques often demand intricate mechanisms or additional modules (Lee et al., 2021), posing challenges in minimizing bias while maintaining simplicity.

## A.4 MORE ABOUT DYNAMIC DATASET PRUNING

To reduce training cost on datasets, dynamic dataset pruning methods (Chen et al., 2024; Killamsetty et al., 2021; Sagawa et al., 2019; Schaul et al., 2015; Zhang et al., 2024) aim to reduce the number of training iterations while maintaining performance. Existing methods employ a variety of criteria to guide pruning, among which loss-based (Attendu & Corbeil, 2023; Kawaguchi & Lu, 2020; Thao Nguyen et al., 2023) method is the most popular. UCB (Raju et al., 2021) applies the cross-entropy loss with exponential moving average (EMA) smoothing to mitigate noise. Infobatch (Qin et al., 2024) randomly prunes low-loss samples and amplifies the gradients of retained ones to preserve the expected gradient. SCAN (Guo & Kankanhalli, 2024) categorizes samples as redundant or ill-matched based on their loss and gradually increases the pruning ratio using cosine annealing. While thsese methods effectively accelerate training and can yield nearly unbiased results, none have explored their potential to mitigate class imbalance in SSL by pruning.

## B    MORE BASE SSL ALGORITHMS

### B.1    MORE ABOUT TRAINING LOSSES OF FIXMATCH

Training losses of FixMatch on a minibatch for the labeled set $\mathcal{MX}$ and a minibatch for the unlabeled set $\mathcal{MU}$ can be expressed as follows:

$$\mathcal{L}_{sup}(x_b;\theta) = \frac{1}{B}\sum_{x_b\in\mathcal{MX}}\mathbf{H}\left(\pi_\theta(y|\alpha(x_b)),p_b\right) \tag{23}$$

with

$$\mathcal{L}_{con}(u_b,\hat{q},\tau;\theta) = \frac{1}{\mu B}\sum_{b=1}^{B}\mathbb{1}(\max(\hat{q}_b)\geq\tau)\mathbf{H}(P_\theta(y|\mathcal{A}(u_b),\hat{q}_b), \tag{24}$$

where $\hat{q}$ denote the concatenations of $\hat{q}_b$. $\mathcal{L}_{sup}$ denotes the supervised loss for weakly augmented labeled data points $u_b$. $\mathcal{L}_{con}$ denotes the consistency regularization loss with the confidence threshold $\tau$.

### B.2    MORE ABOUT FLEXMATCH

To overcome the limitation of FixMatch using a fixed threshold $\tau$ across all classes, Flex-Match (Zhang et al., 2021) introduces the *Curriculum Pseudo Labeling (CPL)* strategy. The key idea is to dynamically adjust the confidence threshold according to the learning status of each class. Specifically, FlexMatch first predicts the class probability for a weakly augmented unlabeled sample $u_b$ as $q_b = \pi_\theta(y|\alpha(u_b))$, and then estimates the learning effect of each class $c$ by $\sigma_t(c)$, *i.e.*, the number of samples predicted as class $c$ that exceed the fixed threshold $\tau$. After normalization, a ratio coefficient $\beta_t(c)$ is obtained, which defines the class-adaptive threshold:

$$T_t(c) = \beta_t(c)\cdot\tau. \tag{25}$$

In this way, hard-to-learn classes receive a lower threshold to include more samples in training, while easy-to-learn classes gradually increase their thresholds to ensure pseudo-label quality. The unsupervised loss is defined as:

$$\mathcal{L}_{con}(u_b,\hat{q},T_t;\theta) = \frac{1}{\mu B}\sum_{b=1}^{\mu B}\mathbb{1}(\max(q_b) > T_t(\arg\max(q_b)))\,\mathbf{H}(\hat{q}_b,\pi_\theta(y|\mathcal{A}(u_b))), \tag{26}$$

where $\hat{q}_b = \arg\max_c q_{b,c}$ denotes the hard pseudo-label, and $\mathcal{A}(\cdot)$ is the strong augmentation function. The overall training objective is

$$\mathcal{L}_t = \mathcal{L}_{sup} + \lambda\mathcal{L}_{con}. \tag{27}$$

where $\lambda$ is weighting hyperparameter.

### B.3    MORE ABOUT FREEMATCH

Unlike FixMatch and FlexMatch, which rely on fixed or indirectly adjusted thresholds, FreeMatch (Wang et al., 2023b) proposes *Self-Adaptive Thresholding (SAT)* that dynamically determines thresholds based on the model's prediction confidence. Specifically, FreeMatch first estimates a global threshold $\tau_t$ using an exponential moving average (EMA) of model confidence:

$$\tau_t = \rho\tau_{t-1} + (1-\rho)\frac{1}{\mu B}\sum_{b=1}^{\mu B}\max(q_b), \tag{28}$$

and further refines it with class-specific local statistics $\tilde{p}_t(c)$:

$$\tau_t(c) = \frac{\tilde{p}_t(c)}{\max_{c'}\tilde{p}_t(c')}\cdot\tau_t. \tag{29}$$

At the early stage of training, thresholds are low to encourage more unlabeled data utilization and faster convergence. As the model becomes more confident, thresholds increase to suppress incorrect pseudo-labels and reduce confirmation bias. The unsupervised loss at iteration $t$ is thus:

$$\mathcal{L}_{con}(u_b,\hat{q},\tau_t;\theta) = \frac{1}{\mu B}\sum_{b=1}^{\mu B}\mathbb{1}(\max(q_b) > \tau_t(\arg\max(q_b)))\,\mathbf{H}(\hat{q}_b,\pi_\theta(y|\mathcal{A}(u_b))). \tag{30}$$

In addition, FreeMatch introduces *Self-Adaptive Fairness (SAF)* regularization $\mathcal{L}_f$, which dynamically calibrates the prediction distribution to encourage diverse predictions and prevent class collapse during early training. Concretely, let $h_t \in \mathbb{R}^C$ denotes the normalized class histogram of model predictions at iteration $t$, and let $h^* \in \mathbb{R}^C$ denotes the target distribution (*e.g.*, a uniform distribution). The SAF regularization is defined as

$$\mathcal{L}_f = D_{\text{KL}}(h_t \,\|\, h^*), \tag{31}$$

where $D_{\text{KL}}(\cdot\|\cdot)$ is the Kullback–Leibler divergence. The final training objective is:

$$\mathcal{L} = \mathcal{L}_{sup} + w_u \mathcal{L}_{con} + w_f \mathcal{L}_f, \tag{32}$$

where $w_u$ and $w_f$ are weighting hyperparameters.

## C  PROOF FOR SECTION 3 AND SECTION 4.

### C.1  PROOF OF PROPOSITION 2

**Proposition 1.** *For an labeled (unlabeled) sample $x_b$ ($u_b$) with target $y_b$ ($\hat{q}_b^t = \arg\max_c q_{b,c}^t$), where $q_b^t = \pi_{\theta^t}(y|\alpha(u_b))$. The one-step learning dynamics of SSL decompose as*

$$\begin{aligned}
\Delta \log \pi_\theta^{t,\text{sup}}(y \mid x_o; x_b) &= -\eta \mathcal{T}^t(x_o) \mathcal{K}^t(x_o, \alpha(x_b)) \mathcal{G}_{\text{sup}}^t(\alpha(x_b), y_b) + \mathcal{O}\left(\eta^2 \|\nabla_\theta \mathbf{z}(\alpha(x_b))\|_{op}^2\right) \\
\Delta \log \pi_\theta^{t,\text{con}}(y \mid x_o; u_b) &= -\eta \mathcal{T}^t(x_o) \mathcal{K}^t(x_o, \mathcal{A}(u_b)) \mathcal{G}_{\text{con}}^t(\mathcal{A}(u_b), \hat{q}_b^t) + \mathcal{O}\left(\eta^2 \|\nabla_\theta \mathbf{z}(\mathcal{A}(u_b))\|_{op}^2\right)
\end{aligned} \tag{6}$$

*where $\mathcal{K}^t(x_o, \alpha(x_b))$ and $\mathcal{K}^t(x_o, \mathcal{A}(u_b))$ are eNTK evaluations of the logit network $\mathbf{z}(\cdot) = g_\theta(\cdot)$, with different inputs. $\mathcal{G}_{\text{sup}}^t(\alpha(x_b), y_b) = \nabla_{\mathbf{z}} \mathcal{L}_{\text{sup}}(\alpha(x_b), y_b)|_{\mathbf{z}^t}$ and $\mathcal{G}_{\text{con}}^t(\hat{q}_b, \mathcal{A}(u_b)) = \nabla_{\mathbf{z}} \mathcal{L}_{\text{con}}(\hat{q}_b, \mathcal{A}(u_b))|_{\mathbf{z}^t}$, respectively.*

*Proof.* We aim to derive the one-step learning dynamics of SSL for both supervised and contrastive terms. Suppose that we want to observe the model's prediction on an "observing example" $x_o$. Starting from Eq. (5), we first approximate $\log \pi^{t+1}(y|x_o)$ using first Taylor expansion (with a slight abuse of notation, we write $\pi^t$ for $\pi_\theta^t$):

$$\log \pi^{t+1}(y|x_o) = \log \pi^t(y|x_o) + <\nabla \log \pi^t(y|x_o), \theta^{t+1} - \theta^t> + \mathcal{O}(\|\theta^{t+1} - \theta^t\|^2).$$

Then, assuming the model updates its parameters using SGD calculated by a "labeled updating example" $(x_b, y_b)$ and an "unlabeled updating example" $(\mathcal{A}(u_b), \hat{q}_b^t)$.

Thus, for for **supervised learning dynamics**, we have, we have

$$\begin{aligned}
\Delta \log \pi^{t+1,sup}(y \mid x_o; x_b) &= \log \pi^{t+1,sup}(y \mid x_o; x_b) - \log \pi^{t,sup}(y \mid x_o; x_b) \\
&= \nabla_\theta \log \pi^t(y \mid x_o)\big|_{\theta^t}(\theta^{t+1} - \theta^t) + \mathcal{O}(\|\theta^{t+1} - \theta^t\|^2)
\end{aligned}$$

Assuming this step is driven solely by supervised loss, we plug in the definition of SGD and repeatedly use the chain rule:

$$\begin{aligned}
\nabla_\theta \log \pi_\theta^t(y \mid x_o)\big|_{\theta^t}(\theta^{t+1} - \theta^t) &= \nabla_\theta \log \pi_\theta^t(x_o)\big|_{\theta^t}\left(-\eta \nabla_\theta \mathcal{L}_{\text{sup}}(\alpha(x_b))\big|_{\theta^t}\right)^\top \\
&= \left(\nabla_z \log \pi_\theta^t(x_o)\big|_{z^t} \nabla_\theta z^t(x_o)\big|_{\theta^t}\right)\left(-\eta \nabla_\theta \mathcal{L}_{sup}(\alpha(x_b))\big|_{\theta^t}\right) \\
&= \nabla_z \log \pi_\theta^t(x_o)\big|_{z^t} \nabla_\theta z^t(x_o)\big|_{\theta^t}\left(-\eta \left(\nabla_z \mathcal{L}_{\text{sup}}(\alpha(x_b))\big|_{z^t} \nabla_\theta z^t(\alpha(x_b))\big|_{\theta^t}\right)\right)^\top \\
&= -\eta \nabla_{\mathbf{z}} \log \pi_\theta^t(x_o)\big|_{\mathbf{z}^t}\left[\nabla_\theta \mathbf{z}^t(x_o)\big|_{\theta^t}\left(\nabla_\theta \mathbf{z}^t(\alpha(x_b))\big|_{\theta^t}\right)^\top\right]\left(\nabla_{\mathbf{z}} \mathcal{L}_{\text{sup}}(\alpha(x_b))\big|_{\mathbf{z}^t}\right)^\top \\
&= -\eta \mathcal{T}^t(x_o) \mathcal{K}^t(x_o, \alpha(x_b)) \mathcal{G}^t(\alpha(x_b), y_b).
\end{aligned}$$

Similarly, for **consistency learning dynamics**, the only difference is that the update sample is changed from $\alpha(x_b)$ to $\mathcal{A}(u_b)$, and the loss is changed from $\mathcal{L}_{sup}$ to $\mathcal{L}_{con}(\mathcal{A}(u_b), \hat{q}_b^t)$. Note that $\hat{q}_b^t = \arg\max_c q_{b,c}^t$ is treated as a constant in this small step (stop-grad), so the gradient can still be directly calculated w.r.t. $z$. Thus,

$$\theta^{t+1} = \theta^t - \eta \nabla_\theta \mathcal{L}_{con}(\mathcal{A}(u_b), \hat{q}_b^t)\big|_{\theta^t}.$$

Parallel to the above derivation, we obtain

$$\Delta \log \pi_\theta^{t,con}(y \mid x_o; u_b) = -\eta \mathcal{T}^t(x_o) \underbrace{\nabla_\theta z^t(x_o)\big|_{\theta^t} \left(\nabla_\theta z^t(\mathcal{A}(u_b))\big|_{\theta^t}\right)^\top}_{\mathcal{K}^t(x_o, \mathcal{A}(u_b))} \underbrace{\nabla_z \mathcal{L}_{con}(\hat{q}_b^t, \mathcal{A}(u_b))\big|_{\mathbf{z}^t}}_{\mathcal{G}_{con}^t(\mathcal{A}(u_b), \hat{q}_b^t)}$$
$$+ \mathcal{O}\left(\eta^2 |\nabla_\theta \mathbf{z}(\mathcal{A}(u_b))|_{op}^2\right).$$

$\square$

## C.2 PROOF OF PROPOSITION 3

**Proposition 2.** (Invariance of baseline image under affine normalization) *Let* $\mathcal{I} = k \cdot \mathbf{1}_d$ *be a baseline image, where* $k \in \{0, 1, \ldots, 255\}$ *and* $\mathbf{1}_d \in \mathbb{R}^d$ *is an all-one vector. Suppose the output of the first hidden transformation is normalized by a normalization layer (*e.g.*, BatchNorm, LayerNorm, InstanceNorm, or GroupNorm) with affine parameters* $(\boldsymbol{W}_2, \boldsymbol{b})$. *Then the logits* $h(\mathcal{I})$ *are independent of* $k$ *and reduce to*

$$h(\mathcal{I}) = \boldsymbol{b}, \quad \pi_\theta(\mathcal{I}) = \texttt{Softmax}(\boldsymbol{b}). \tag{9}$$

*Proof.* Consider a neural network with two layers: the first layer is a linear transformation, and the second layer is a normalization layer followed by an affine transformation. For an input $\mathcal{I} \in \mathbb{R}^d$, assume the model has the following structure:

$$h^{(1)}(\mathcal{I}) = \sigma(\boldsymbol{W}_1 \mathcal{I}); \quad h(\mathcal{I}) = \texttt{BatchNorm}(h^{(1)}(\mathcal{I})) = \frac{h^{(1)}(\mathcal{I}) - \mathbb{E}[h^{(1)}(\mathcal{I})]}{\sqrt{\text{Var}[h^{(1)}(\mathcal{I})] + \epsilon}} \cdot \boldsymbol{W}_2 + \boldsymbol{b},$$

Let the baseline image $\mathcal{I} = k \cdot \mathbf{1}_d$, where $\mathbf{1}_d$ is a vector of ones, and $k$ is a scalar. Our goal is to show that the output $h(\mathcal{I})$ for the baseline image is independent of $k$ and depends only on the bias term $\boldsymbol{b}$. For the baseline image $\mathcal{I} = k \cdot \mathbf{1}_d$, the output of this neural network is:

$$h^{(1)}(\mathcal{I}) = \sigma(\boldsymbol{W}_1 \cdot (k \cdot \mathbf{1}_d)) = \sigma(k \cdot \boldsymbol{W}_1 \mathbf{1}_d) = \sigma(k \cdot \boldsymbol{w}).$$

where $\boldsymbol{w} = \boldsymbol{W}_1 \mathbf{1}_d \in \mathbb{R}^m$, which is a constant vector. We see that the output of the first layer depends on $k$ and the constant vector $\boldsymbol{w}$, and it is passed through the activation function $\sigma$. Now, consider the effect of the BatchNorm layer. For the baseline image $\mathcal{I} = k \cdot \mathbf{1}_d$, since $h^{(1)}(\mathcal{I}) = \sigma(k \cdot \boldsymbol{w})$ is a constant vector, the mean $\mathbb{E}[h^{(1)}(\mathcal{I})]$ and variance $\text{Var}[h^{(1)}(\mathcal{I})]$ are constants that depend only on $\boldsymbol{w}$. From first principles, we can set $k = 0$ $\square$

Note that if the input $\mathcal{I}$ is random Gaussian noise or a batch mean, the situation would be different.

- **Gaussian Noise**. Let $\mathcal{I}_n \sim \mathcal{N}(0, \sigma^2) \in \mathbb{R}^d$ be a random Gaussian noise vector. After normalization:

$$h(\mathcal{I}_n) = \frac{h^{(1)}(\mathcal{I}_n) - \mathbb{E}(h^{(1)}(\mathcal{I}_n))}{\sqrt{Var[h^{(1)}(\mathcal{I}_n)] + \epsilon}} \cdot \boldsymbol{W}_2 + \boldsymbol{b}$$

Since the input pixel values are random, the mean and variance of the first-layer output depend on the noise distribution characteristics. These statistics fluctuate with the randomness of the input, in contrast to the baseline image, where the normalized output is solely determined by the bias term $\boldsymbol{b}$.

- **Batch Mean**. Let $\mathcal{I}_\mu = \frac{1}{B} \sum_{i=1}^B x_i \in \mathbb{R}^d$ be the batch mean vector. After normalization, the affine transformation:

$$h(\mathcal{I}_\mu) = \frac{h^{(1)}(\mathcal{I}_\mu) - \mathbb{E}(h^{(1)}(\mathcal{I}_\mu))}{\sqrt{Var[h^{(1)}(\mathcal{I}_\mu)] + \epsilon}} \cdot \boldsymbol{W}_2 + \boldsymbol{b}$$

Unlike Gaussian noise images, the mean input of data within a batch does not contain complete randomness; the mean and variance are relatively stable but still do not solely depend on the $\boldsymbol{b}$.

### C.3   PROOF OF THEOREM 1

**Theorem 1.** (Bias as the conditional distribution prior) *Assume the model $h(x)$ as characterized in Eq. (8) is trained using cross-entropy loss:*

$$\mathcal{L} = \mathbb{E}_{(x,y)}\big[-y^\top \log \texttt{Softmax}(h(x))\big]. \tag{10}$$

*At a population risk minimizer $(W_2^\star, b^\star)$ we have*

$$\hat{p}^\star(x) = P(y \mid x), \qquad \hat{p}^\star(\mathcal{I}) = \texttt{Softmax}(b^\star) = P\big(y \mid \tfrac{h^{(1)}(\mathcal{I})-\mathbb{E}[h^{(1)}(\mathcal{I})]}{\sqrt{\text{Var}[h^{(1)}(\mathcal{I})]+\epsilon}} = 0\big). \tag{11}$$

*For the baseline image $\mathcal{I}$ in Proposition 3, the baseline prediction thus coincides with the conditional class distribution at the normalized-zero feature state, capturing the class prior induced by the long-tailed training distribution.*

*Proof.* Consider the two-layer network $f_\theta(x) = \frac{h^{(1)}(x)-\mathbb{E}[h^{(1)}(x)]}{\sqrt{\text{Var}[h^{(1)}(x)]+\epsilon}} \cdot \gamma + \beta$, where $h^{(1)}(x) = W_1 x$. The cross-entropy loss is given by:

$$\mathcal{L} = \mathbb{E}_{(x,y)}\left[-y^\top \log \texttt{Softmax}(h(x))\right].$$

Minimizing the population risk results in $\hat{p}^\star(x) = \texttt{Softmax}(h(x)) = P(y \mid x)$.

For the baseline image $\mathcal{I}$, we analyze the model's output:

$$\hat{p}^\star(\mathcal{I}) = \texttt{Softmax}(b^\star).$$

Since $\frac{h^{(1)}(\mathcal{I})-\mathbb{E}[h^{(1)}(\mathcal{I})]}{\sqrt{\text{Var}[h^{(1)}(\mathcal{I})]}} \to 0$ for a baseline image with no input signal, the model's output is determined solely by $b^\star$.

Thus, we have:

$$P\left(y \mid \frac{h^{(1)}(\mathcal{I}) - \mathbb{E}[h^{(1)}(\mathcal{I})]}{\sqrt{\text{Var}[h^{(1)}(\mathcal{I})] + \epsilon}} = 0\right) = \texttt{Softmax}(b^\star).$$

Finally, we conclude that the baseline prediction corresponds to the conditional class distribution at the normalized-zero feature state, capturing the class prior induced by the long-tailed distribution. $\qquad\square$

### C.4   PROOF OF PROPOSITION 4

**Proposition 3.** *Let $\pi = \texttt{Softmax}(z)$ and $z = g_\theta(x)$. The one-step dynamics decompose as*

$$\Delta \log \pi^t(y \mid \mathcal{I}) = -\eta \mathcal{T}^t(\mathcal{I})\mathcal{K}^t(\mathcal{I}, x)\mathcal{G}^t(x, y) + \mathcal{O}(\eta^2 \|\nabla_\theta z(x)\|_{\text{op}}^2), \tag{13}$$

*where $\mathcal{T}^t(\mathcal{I}) = \nabla_z \log_{\pi^t}(\mathcal{I}) = I - \mathbf{1}\pi_{\theta^t}^T(\mathcal{I})$, $\mathcal{K}^t(\mathcal{I}, x) = (\nabla_\theta z(\mathcal{I})|_{\theta^t})(\nabla_\theta z(x)|_{\theta^t})^T$ is the empirical neural tangent kernel of the logit network $z$, and $\mathcal{G}^t(x, y) = \nabla_z \mathcal{L}(x, y) \mid_{z^t}$.*

*Proof.* Inspired by the analysis of the learning dynamic of (Ren et al., 2022; Ren & Sutherland, 2025). In this work, we want to observe the classifier's prediction on the baseline image $\mathcal{I}$. Starting from Eq (12), we first approximate $\log \pi^{t+1}(y \mid \mathcal{I})$ using first-order Talyor expansion, with slightly abused symbols, we use $\pi^t$ to represent $\pi_{\theta^{t+1}}$, $x$ to represent labeled sample $x_b^n$ and $u$ to represent unlabeled sample $u_b^m$:

$$\log \pi^{t+1}(y|\mathcal{I}) = \log \pi^t(y|\mathcal{I}) + <\nabla \log \pi^t(y|\mathcal{I}), \theta^{t+1} - \theta^t> + \mathcal{O}(\|\theta^{t+1} - \theta^t\|^2)$$

Then, assuming the model updates its parameters using SGD calculated by an "updating labeled example" $(x, y)$ or an "updating unlabeled example" $u$, we can rearrange the terms in the above equation to get the following expression:

$$\Delta \log \pi^t(y|\mathcal{I}) = \log \pi^{t+1}(y|\mathcal{I}) - \log \pi^{t+1}(y|\mathcal{I}) = \nabla_\theta \log \pi^t(y|\mathcal{I})|_{\theta^t}(\theta^{t+1} - \theta^t) + \mathcal{O}(\|\theta^{t+1} - \theta^t\|^2),$$

To evaluate the leading term, we first take a labeled sample as an example plug in the definition of SGD, and repeatedly use the chain rule:

$$
\begin{aligned}
\nabla_\theta \log \pi^t(y|\mathcal{I})|_{\theta^t}(\theta^{t+1} - \theta^t) &= (\nabla_z \log \pi^t(y|\mathcal{I})|_{z^t})(-\eta \nabla_\theta \mathcal{L}(x)|_{\theta^t})^T \\
&= (\nabla_z \log \pi^t(y|\mathcal{I})|_{z^t})(-\eta \nabla_\theta \mathcal{L}(x)|_{z^t} - \nabla_\theta z^t(x)|_{\theta^t})^T \\
&= -\eta \nabla_z \log \pi^t(\mathcal{I})|_{z_t} [\nabla_\theta z(\mathcal{I})|_{\theta^t} (\nabla_\theta z(x)|_{\theta^t})^T](\nabla_z \mathcal{L}(x)|_{z^t})^T \\
&= -\eta \mathcal{T}^t(\mathcal{I})\mathcal{K}^t(\mathcal{I}, x)\mathcal{G}^t(x, y)
\end{aligned}
\tag{33}
$$

$\square$

### C.5 MORE ABOUT ANALYZING THE DYNAMICS OF THE LOGITS DEBIASING ALGORITHM

#### C.5.1 PER-STEP DECOMPOSITION OF RESAMPLING

Resampling is another widely used strategy for mitigating class imbalance in long-tail semi-supervised learning. Instead of modifying the loss, resampling adjusts the data distribution by altering the frequency with which each class is drawn. Let $\mathbb{P}_{\mathrm{rs}}(x \in c) = r^c$ denote the (possibly normalized) sampling ratio for class $c$, which determines the probability of selecting samples from that class during training. Then the per-step update of the log-posterior under resampling becomes

$$
\Delta \log \pi_\theta^{t,\mathrm{rs}}(y \mid \mathcal{I}; x) = -\eta \, \mathcal{T}^t(\mathcal{I}) \, \tilde{\mathcal{K}}_{rs}^t(\mathcal{I}, x; r^c) \, \tilde{\mathcal{G}}_{rs}^t(x, y; r^c) + \mathcal{O}(\eta^2 \|\nabla_\theta \mathbf{z}(x)\|_{\mathrm{op}}^2),
\tag{34}
$$

where $\tilde{\mathcal{K}}_{rs}^t(\mathcal{I}, x; r^c) = \mathbb{E}_{x \sim r^c}[\mathcal{K}^t(\mathcal{I}, x)]$, $\tilde{\mathcal{G}}_{rs}^t(x, y; r^c) = \mathbb{E}_{x \sim r^c}[\mathcal{G}^t(x, y)]$. This decomposition highlights that resampling influences learning solely through changing the expectation measure. The modified kernel $\tilde{\mathcal{K}}_{rs}^t$ reshapes how training samples transfer influence to the test input, while the modified residual term $\tilde{\mathcal{G}}_{rs}^t$ reweights the magnitude of each update. Increasing the sampling ratio of tail classes therefore amplifies their effective contribution at every step, accelerating their representation and decision boundary updates to match those of head classes, *i.e.* offering a direct dynamical explanation for the effectiveness of resampling in long-tail regimes.

#### C.5.2 PER-STEP DECOMPOSITION OF CDMAD

In this section, we use the loss function of a specific method in logits adjustment, CDMAD (Lee & Kim, 2024), as a case study and integrate it into the learning dynamics framework we propose. The consistency loss of CDMAD as:

$$
\mathcal{L}_{con}(u_b, \hat{q}, \tau; \theta) = \frac{1}{\mu B} \sum_{b=1}^{B} \mathbb{1}(\max(\hat{q}_b) \geq \tau) \mathbf{H}(P_\theta(y|\mathcal{A}(u_b), q_b^*),
\tag{35}
$$

where $\mathbf{H}$ is cross-entropy loss, $q_b^* = \arg\max(\pi_\theta(y|\alpha(u_b)) - \pi_\theta(y|\mathcal{I}))$. Our framework reveals that CDMAD operates through two complementary dynamical mechanisms:

$$
\begin{aligned}
\Delta \log \pi_\theta^t(y \mid \mathcal{I}) = -\eta \mathcal{T}^t(\mathcal{I})(\mathcal{K}^t(\mathcal{I}, \alpha(x_b))\mathcal{G}_{\mathrm{sup}}^t(\alpha(x_b), y_b) + \\
\mathcal{K}^t(\mathcal{I}, \mathcal{A}(u_b))\mathcal{G}_{\mathrm{con}}^t(\mathcal{A}(u_b), \alpha(x_b))) + \mathcal{O}^2
\end{aligned}
\tag{36}
$$

According to the analysis of Xing et al. (2025), $\mathcal{G}^t$ using the baseline image enhances the balance of the base SSL model implicitly utilizing the integrated gradient flow $\nabla_\theta \mathcal{L}_{\mathrm{Con}} = \sum_b \left( \sum_{i=1}^d \mathrm{IntegratedGrads}_i(u_b) \right) \nabla g_b + \sum_b q_{\mathcal{A},b} \frac{\partial q_{\mathcal{A},b}}{\partial \theta}$. We now place $\nabla_\theta \mathcal{L}_{\mathrm{Con}}$ directly into $\mathcal{G}_{\mathrm{con}}^t$ to capture the influence of the consistency loss on the model's update dynamics. The updated $\mathcal{G}_{\mathrm{con}}^t$ is:

$$
\mathcal{G}_{\mathrm{con}}^t(\mathcal{A}(u_b), \alpha(x_b)) = \sum_b \left( \sum_{i=1}^d \mathrm{IntegratedGrads}_i(u_b) \right) \nabla g_b + \sum_b q_{\mathcal{A},b} \frac{\partial q_{\mathcal{A},b}}{\partial \theta}.
\tag{37}
$$

The term $\mathcal{G}_{\mathrm{con}}^t(\mathcal{A}(u_b), \alpha(u_b))$ now explicitly includes the consistency loss gradient $\nabla_\theta \mathcal{L}_{\mathrm{Con}}$, which involves the Integrated Gradients over the perturbations $u_b$ as well as the change in model output probabilities.

Table 7: Comparison of bACC/GM on CIFAR-10-LT under different baseline images.

| FixMatch+DyTrim | CIFAR-10-LT | |
| --- | --- | --- |
| Type of baseline | $\gamma_l = \gamma_u = 100$ | $\gamma_l = 100, \gamma_u = 150$ |
| Noise | 77.7 / 76.8 | 76.7 / 75.8 |
| Batch Means | 78.0 / 76.1 | 76.7 / 74.2 |
| Red | 83.5 / 83.2 | 82.2 / 81.7 |
| Green | 83.7 / 83.3 | 81.5 / 81.0 |
| Blue | 84.5 / 84.2 | 83.1 / 82.6 |
| Gray | 84.1 / 83.7 | 82.3 / 81.9 |
| White | 84.2 / 83.8 | 82.4 / 82.0 |
| Black | **84.8 / 84.4** | **83.8 / 83.4** |

### C.6 EFFECT OF THE BASELINE IMAGE FOR GUIDING DATA PRUNING

The training objective can be interpreted as the minimization of the empirical risk $\mathcal{L}$. Assuming that all labeled samples $x_b^n$ from $\mathcal{X}$ and unlabeled samples $u_b^m$ from $\mathcal{U}$ are drawn from continuous distributions $\rho^l(x_b^n)$ and $\rho^u(u_b^m)$, respectively, the training objective can be formulated as:

$$\arg\min_{\theta \in \Theta} \underset{x_b^n \in \mathcal{X}, u_b^m \in \mathcal{U}}{\mathbb{E}} [\mathcal{L}(x_b^n, u_b^m; \theta)] = \int_{x_b^n} \mathcal{L}_{sup}(x_b^n, \theta)\rho^l(x_b^n)dx_b^n + \int_{u_b^m} \mathcal{L}_{con}(u_b^m, \theta)\rho^l(u_b^m)du_b^m.$$

(38)

After applying a data pruning policy, we sample $x_b^n$ and $u_b^m$ to obtain the labeled pruned subset $\mathcal{S}_t^l$ and the unlabeled pruned subset $\mathcal{S}_t^u$, according to the labeled pruning probabilities $\mathcal{P}_t^l(x_b^n)$ and unlabeled pruning probabilities $\mathcal{P}_t^u(u_b^m)$, respectively. For the labeled samples, we directly optimize over the pruned subset $\mathcal{S}_t^l$ without reweighting the loss terms. Notably, the class-aware pruning probability $r_c = \pi_\theta(\mathcal{I})_c$ inherently adjusts $\mathcal{S}_t^l$ toward an asymptotically balanced class distribution. By retaining more samples from minority classes (lower $r_c$) and pruning more samples from majority classes (higher $r_c$), the pruned subset $\mathcal{S}_t^l$ naturally mitigates class imbalance. As a result, even without explicit rescaling, the empirical risk over $\mathcal{S}_t^l$ approximates:

$$\arg\min_{\theta \in \Theta} \underset{x_b^n \in \mathcal{S}_t^l}{\mathbb{E}} [\mathcal{L}_{sup}(x_b^n, \theta)] \propto \frac{1 - \mathcal{P}_t^l(x_b^n)}{c_t^l} \int_z \mathcal{L}_{sup}(x_b^n, \theta)\rho_l(x_b^n)dx_b^n,$$

(39)

where $c_t^l = \mathbb{E}_{x_b^n \sim \rho_l}[1 - \mathcal{P}_t^l(x_b^n)]$. The term $\frac{1 - \mathcal{P}_t^l(z)}{c_t^l}$ acts as an *implicit reweighting* due to the class-aware pruning policy. For unlabeled samples, pruning with uniform probability $r$ and rescaling losses by $\gamma_t(u) = \frac{1}{1 - \mathcal{P}_t^u(u)}$ yields

$$\arg\min_{\theta \in \Theta} \underset{u_b^m \in \mathcal{S}_t^u}{\mathbb{E}} [\gamma_t(u_b^m)\mathcal{L}_{con}(u_b^m, \theta)] \propto \frac{1}{c_t^u} \int_z \mathcal{L}_{con}(u_b^m, \theta)\rho^l(u_b^m)du_b^m,$$

(40)

where $c_t^u = \mathbb{E}_{u_b^m \sim \rho_u}[1 - \mathcal{P}_t^u(u_b^m)]$. Crucially, even with uniform pruning rates, the interplay of consistency regularization and confidence thresholding ensures $\mathcal{S}_t^u$ to be implicitly balanced, thus training on $\mathcal{S}_t^u$ with rescaled factor $\gamma_t(u_b^m)$ could achieve a better result as training on the $\mathcal{U}$.

## D  MORE ABOUT THE BASELINE IMAGE

### D.1  MORE DETAIL ABOUT THE SELECTION OF BASELINE IMAGE

**Sensitivity of different baseline images $\mathcal{I}$.** We further examined the sensitivity of DyTrim to the choice of baseline image by conducting ablation studies on CIFAR-10-LT with different types of inputs, including noise, dataset means, and solid colors. Table 7 shows that solid-color images consistently outperform noise or mean-based baselines. Among them, white and black images deliver the strongest results.

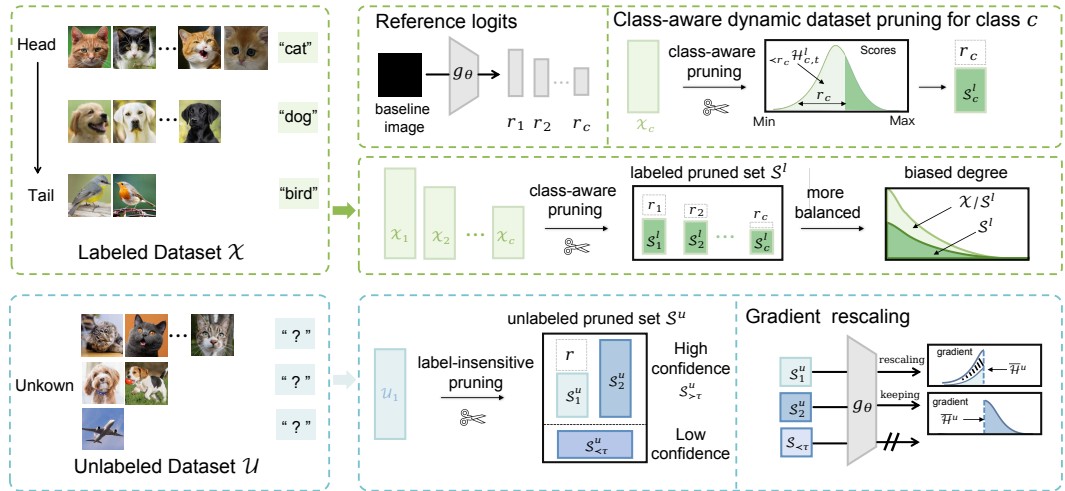

Figure 4: Illustration of the proposed DyTrim framework. DyTrim mainly consists of two operations, named labeled pruning and unlabeled pruning. $\boldsymbol{H}^l_{\prec r_c, t}$ and $\bar{\boldsymbol{H}}^u_t$ denote the adaptive thresholds of scores of labeled samples and unlabeled samples, with slight abuse of symbols. $\mathcal{S}^u_{\prec \tau}$ denote the low confidence unlabeled sample which $p^*(u^m_b) \geq \tau$. Labeled pruning provides a class-aware pruning policy for each sample from class $c$. Unlabeled pruning provides a random pruning policy from the original unlabeled $\mathcal{U}$ and uses a gradient rescaling strategy ($\times 1/(1-r)$ for which samples from $s^u_1$ is selected to pruning) to keep the approximately same gradient expectation.

## D.2 DETAIL OF THE BIAS TERM AND RUNNING STATISTICS

**Effects of bias term.** When the bias term $\beta$ of the BN layer is frozen and equal to 0, $h(\mathcal{I})$ becomes $\gamma * (\langle \mathbf{w}, k \rangle - \mathbb{E}[\langle \mathbf{w}, k \rangle]) / \sqrt{\text{Var}[\langle \mathbf{w}, k \rangle]}$ which is the same as the Eq.(7) except for a bias term. Ignoring the running statistics strategy, the form of $h(\mathcal{I})$ only depends on the $\beta$. As a result, $h(\mathcal{I})$ becomes $h(\mathcal{I}) \to 0$ during training and $h(\mathcal{I}) \to -\gamma * \mathbb{E}_{mom}[\langle \mathbf{w}, x_b \rangle] / \sqrt{\text{Var}_{mom}[\langle \mathbf{w}, x_b \rangle]}$ during testing. This shows that the $g^*_\theta$ operation has no effect in the training phase and only eliminates the impact of the unbalanced running means in the testing phase. This will affect the ability to benefit $h$ from $g^*_\theta$, as shown in Table. 8.

**Effects of running statistics.** When we do not keep running estimates, batch statistics are instead used during evaluation time as well. The form of $h(\mathcal{I})$ becomes $h(\mathcal{I}) \to \beta$ both training and testing. We can rewrite $g^*_\theta(x_t) = \gamma * (\langle \mathbf{w}, x_t \rangle - \mathbb{E}[\langle \mathbf{w}, x_t \rangle]) / \sqrt{\text{Var}[\langle \mathbf{w}, x_t \rangle]}$. On the other hand, as $h(\mathcal{I}) \to 0$, the benefit of $g^*_\theta$ is also vanishes, also shown in Table. 8.

We then extend our results to a non-linear neural network, thus we have the following corollary:

Table 8: Comparison of bACC/GM on CIFAR-10-LT.

| Metric | With original $g^*_\theta$ | $g^*_\theta$ without $\beta$ | $g^*_\theta$ without $\mathbf{x}_{mom}$ | $g^*_\theta$ without $\beta$ & $\mathbf{x}_{mom}$ |
|---|---|---|---|---|
| bACC | 83.6 ± 0.46 | 80.92 ± 0.02↓2.68 | 71.63 ± 0.35↓11.97 | 64.01 ± 0.14↓19.59 |
| GM | 83.1 ± 0.57 | 80.37 ± 0.23↓2.73 | 67.85 ± 0.51↓15.25 | 54.48 ± 0.36↓28.62 |

# E MORE DETAILS ABOUT DYTRIM

## E.1 MORE ABOUT LABELED PRUNING

Specifically, we exploit the pruning policy to prune samples based on their scores. Then, for the pruned labeled samples, their scores remain unmodified as previously. For the remaining samples,

their scores are updated by the losses in the current epoch. To ensure dynamic adaptation:

$$\mathcal{H}^l_{c,t+1}(x^n_b) = \begin{cases} \mathcal{H}^l_{c,t}(x^n_b) & x^n_b \in \mathcal{X}n\mathcal{S}^l, \\ \mathcal{L}_{sup}(x^n_b) & x^n_b \in \mathcal{S}^l. \end{cases} \tag{41}$$

where $\mathcal{S}^l$ denotes the pruned subset formed for labeled datasets.

### E.2 More about unlabeled pruning

For a remaining sample with score $\mathcal{H}^u_t(u^m_b) < \bar{\mathcal{H}}^m_t$, whose corresponding pruning probability is $r$, its gradient is scaled to $1/(1-r)$ times of the original, otherwise the gradient remains unchanged. The score $\mathcal{H}^u_{t+1}(u^m_b)$ is derived from the consistency regularization loss values $\mathcal{L}_{con}(\alpha(u^m_b), \mathcal{A}(u^m_b))$ for unlabeled data points. To enhance pseudo-label reliability, we further apply a confidence threshold $\tau$, where only samples with $p^*(u^m_b) > \tau$ contribute to $\mathcal{L}_{con}$, where $\mathcal{L}_{con} = \frac{1}{B} \sum_{b=1}^{B} \mathbb{I}(p^*(u^m_b) > \tau)\mathbf{H}(P_\theta(y|\mathcal{A}(u^m_b), \hat{q}_b)$. Thus, we formulate the update of $\mathcal{H}^u_{t+1}(u^m_b)$ as:

$$\mathcal{H}^u_{t+1}(u^m_b) = \begin{cases} \mathcal{H}^u_t(u^m_b) & u^m_b \in \mathcal{U}n\mathcal{S}^u, \\ \mathcal{L}_{con}(u^m_b) & u^m_b \in \mathcal{S}^u. \end{cases} \tag{42}$$

where $\mathcal{S}^u$ denotes the pruned subset formed for labeled datasets. **Initialization:** at $t = 0$, scores $\mathcal{H}^u_t$ and $\mathcal{H}^l_t$ are all set to $\{1\}$, as no prior loss is available.

## F   Pseudo code of the proposed algorithm

The pseudo-code that describes the DyTrim is presented in Algorithm 1 and Algorithm 2.

---

**Algorithm 1** DyTrim for Labeled Data Selection

---

**Input:** Labeled set of $N$ samples $\mathcal{X} = \{(x^n, y^n)\}^N_{n=1}$, score set of the samples $\mathcal{V}^l$, number of classes $n_c$, biased degree $b$
**Output:** Labeled pruned set $\mathcal{S}^l$ ($\mathcal{S}^l \subseteq \mathcal{X}, |\mathcal{S}^l| <= |\mathcal{X}|$)

1: $\mathcal{S}^l \leftarrow \emptyset$                                          ▷ Initialize the labeled pruned set
2: **for** $c = 0$ to $n_c - 1$ **do**
3:     $\mathcal{I}_c \leftarrow \{i \mid y_i = c\}$
4:     $\mathcal{V}^l_c \leftarrow \{\mathcal{V}^l_i \mid i \in \mathcal{I}_c\}$                   ▷ Select scores of class $c$ samples
5:     $k_c \leftarrow \lfloor (1 - b_c) \cdot |\mathcal{X}_c| \rfloor$   ▷ Compute target pruned set size of class $c$ based on biased degree
6:     $\mathcal{I}^{top}_c \leftarrow \text{TopK}(\mathcal{I}_c, \mathcal{V}^l_c, k_c)$            ▷ Select indices of top-$k_c$ scored samples
7:     $\mathcal{S}^l \leftarrow \mathcal{S}^l \cup \mathcal{I}^{top}_c$
8: **end for**
9: **return** $\mathcal{S}^l$

---

## G   Experimental Settings

### G.1   Models

Unless otherwise specified, we adopt Wide ResNet (WRN) (Zagoruyko & Komodakis, 2016) as the default backbone following common practice in semi-supervised learning. Additionally, we also evaluate Tiny Vision Transformers (TinyViT) (Wu et al., 2022) on CIFAR-10-LT and CIFAR-100-LT. For ImageNet-127, we employ ResNet-50 (He et al., 2016) as the backbone to ensure scalability on large-scale datasets.

### G.2   Implementation details

All experiments are trained for 500 epochs with 500 steps per epoch, resulting in a total of 250,000 iterations. We use Stochastic Gradient Descent (SGD) (Bottou, 2012) with a fixed learning rate of $\eta = 0.0015$ and a batch size of 32. The pruning ratio of the unlabeled dataset is set to 0.7, and the parameter $\delta$ is aligned with InfoBatch (Qin et al., 2024), fixed at 0.875. For CIFAR-10-LT, the largest

---

**Algorithm 2** DyTrim for Unlabeled Data Selection

---

**Input:** Unlabeled set of $M$ samples $\mathcal{U} = \{(u^m)\}_{m=1}^M$, score set of the samples $\mathcal{V}^u$, pruning ratio $r$, weight of samples $w$
**Output:** Unlabeled pruned set $\mathcal{S}^l$ ($\mathcal{S}^l \subseteq \mathcal{U}$, $|\mathcal{S}^u| <= |\mathcal{U}|$)

1:   $\mathcal{S}^u \leftarrow \emptyset$         ▷ Initialize the unlabeled pruned set
2:   $\mathcal{I}_0 \leftarrow \{i \mid \mathcal{V}_i^u = 0\}$         ▷ Select low confidence samples
3:   $\mathcal{I}_{\neq 0} \leftarrow \{i \mid \mathcal{V}_i^u \neq 0\}$         ▷ Select high confidence samples
4:   $\mathcal{S}^u \leftarrow \mathcal{S}^u \cup \mathcal{I}_0$
5:   $\mu \leftarrow \text{Mean}(\{\mathcal{V}_i^u \mid i \in \mathcal{I}_{\neq 0}\})$
6:   $\mathcal{I}_{\text{well}} \leftarrow \{i \in \mathcal{I}_{\neq 0} \mid \mathcal{V}_i^u < \mu\}$         ▷ Select well-learned samples
7:   $\mathcal{I}_{\text{poor}} \leftarrow \mathcal{I}_{\neq 0} \setminus \mathcal{I}_{\text{well}}$         ▷ Select poorly-learned samples
8:   $\mathcal{S}^u \leftarrow \mathcal{S}^u \cup \mathcal{I}_{\text{poor}}$
9:   $\mathcal{I}_{\text{select}} \leftarrow \text{Randomly select } \lfloor (1-r) \cdot |\mathcal{I}_{\text{well}}| \rfloor \text{ samples from } \mathcal{I}_{\text{well}}$
10: $\mathcal{S}^u \leftarrow \mathcal{S}^u \cup \mathcal{I}_{\text{select}}$
11: $w_i \leftarrow 1, \ \forall i \in \{1, \ldots, M\}$         ▷ Reset weights
12: $w_i \leftarrow \frac{1}{1-r}, \ \forall i \in \mathcal{I}_{\text{select}}$         ▷ Rescaling
13: **return** $\mathcal{S}^u$

---

labeled class contains 1,500 samples, while the largest unlabeled class contains 3,000 samples. For CIFAR-100-LT, the largest labeled and unlabeled classes contain 150 and 300 samples, respectively. For STL-10-LT, the largest labeled class contains 450 samples. To assess classification performance, we adopt balanced accuracy (bACC) (Huang et al., 2016) and geometric mean (GM) (Kubat, 1997) for CIFAR-10-LT and STL-10-LT. For CIFAR-100-LT and ImageNet-127, evaluation is conducted solely using bACC. Each experiment is repeated three times on RTX 4090 GPUs to ensure reproducibility, and we report both the mean and the standard error.

# H   ADDITIONAL EXPERIMENTAL RESULTS

## H.1   BASELINES

The classification performance of the DyTrim was compared with those of the following algorithms: 1. vanilla algorithm - Deep CNN trained with cross-entropy loss, 2. CIL algorithms - Resampling (JAPKOWICZ, 2000), LDAM-DRW (Cao et al., 2019), and cRT (Kang et al., 2020), 3. SSL algorithms - FixMatch (Sohn et al., 2020), and 4. CISSL algorithms - DARP, DARP+LA, DARP+cRT (Kim et al., 2020), CReST, CReST+LA (Wei & Gan, 2023), ABC (Lee et al., 2021), CoSSL (Fan et al., 2022), DASO (Oh et al., 2022), SAW, SAW+LA and SAW+cRT (Lai et al., 2022) combined with FixMatch. Adsh(Guo & Li, 2022), DebiasPL (Wang et al., 2022), UDAL(Lazarow et al., 2023) and L2AC (Wang et al., 2023a) combined with FixMatch. We report the performance of the baseline algorithms reported in Tables of Lai et al. (2022) and Fan et al. (Fan et al., 2022) when it is reproducible; the performance measured using the uploaded code was reported otherwise.

## H.2   ADDITIONAL RESULTS ON CIFAR-10-LT

Following prior works (Xing et al., 2025; Lee & Kim, 2024; Guo et al., 2024), we evaluate under a more challenging scenario where the unlabeled set is imbalanced in the reverse direction of the labeled set (Table 9). Across all settings, DyTrim delivers consistent gains by applying balanced pruning on the labeled data. Notably, when combined with FixMatch, DyTrim surpasses CDMAD by more than 1% in both bACC and GM. Similar benefits are observed for FlexMatch and FreeMatch: DyTrim improves FlexMatch by approximately 1.1–1.3% and FreeMatch by around 0.9–1.5%.

We also compared the classification performance of CDMAD with ACR (Xiang et al., 2020) and BaCon, two recent CISSL algorithms. From Table. 10, we can observe that CDMAD outperforms both ACR and BaCon.

To further validate the balanced classification effect of DyTrim, we visualized the dynamics of baseline image logits during training as shown in Figure. 5 (a), (b) and (c). The results clearly showed that DyTrim significantly reduced classifier bias induced by class imbalance.

Table 9: Comparison of bACC/GM on CIFAR-10-LT($\gamma_l = 100, \gamma_u = 100$(reversed)).

| Algorithm | CIFAR-10-LT, $\gamma_l = 100, \gamma_u = 100$(reversed) | | | | | |
|---|---|---|---|---|---|---|
| | ABC | SAW | SAW+LA | SAW+cRT | CDMAD | DyTrim |
| FixMatch+ | 69.5/66.8 | 72.3/68.7 | 74.1/72.0 | 75.5/73.9 | 77.1/75.4 | **78.2 / 76.7** |
| FlexMatch+ | $-/-$ | $-/-$ | $-/-$ | $-/-$ | 67.2/65.1 | **68.3 / 66.4** |
| FreeMatch+ | $-/-$ | $-/-$ | $-/-$ | $-/-$ | 68.5/66.4 | **69.4 / 67.9** |

Table 10: Comparison of bACC/GM on CIFAR-10-LT

| Algorithm/CIFAR-10-LT | $\gamma_l = \gamma_u = 100$ | $\gamma_l = \gamma_u = 1$ |
|---|---|---|
| FixMatch+ACR | 81.8 / 81.4 | 85.6 / 85.3 |
| FixMatch+BaCon | 84.4 / 84.0 | 82.0 / 81.5 |
| FixMatch+CDMAD | 83.6 / 83.1 | 87.5 / 87.1 |
| FixMatch+DyTrim | **84.8 / 84.4** | **87.9 / 87.5** |

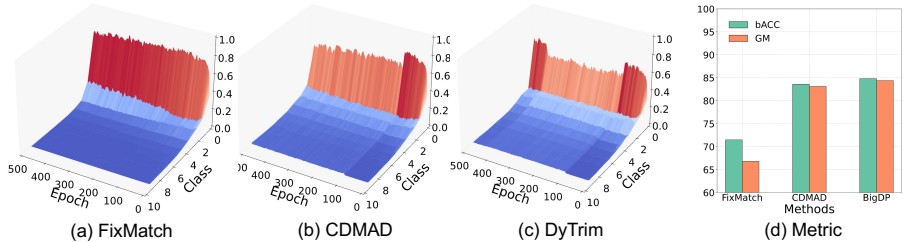

| (a) FixMatch | (b) CDMAD | (c) DyTrim | (d) Metric |
|---|---|---|---|

Figure 5: (a), (b) and (c) present the change of $\pi_\theta(\mathcal{I})$ for the baseline image on CIFAR-10-LT with $\gamma_l = \gamma_u = 100$ across different methods. (d) present the bACC and GM on those methods.

## H.3  RESULTS ON SMALL-IMAGENET-127

ImageNet-127 is a naturally long-tailed dataset, widely used to evaluate class-imbalanced semi-supervised learning (CISSL) algorithms at scale. Following standard protocol, we downsample images to resolutions of $32 \times 32$ and $64 \times 64$ using the box interpolation method from the Pillow library, and randomly select 10% of the training samples as labeled data. Under such limited supervision and class imbalance, learning discriminative representations and a balanced classifier is particularly challenging. As reported in Table. 11, DyTrim achieves the highest balanced accuracy (bACC) at both resolutions, outperforming the strongest baseline CD-MAD by 3.0% at $32 \times 32$ and 1.2% at $64 \times 64$. These improvements demonstrate the robustness of our method, especially under low-resolution and low-resource conditions. The performance gain at lower resolutions suggests that DyTrim effectively handles the compounded difficulty of reduced vi-

Table 11: Comparison of bACC on Small-ImageNet-127.

| Algorithm | Small-ImageNet-127 | |
|---|---|---|
| | $32 \times 32$ | $64 \times 64$ |
| FixMatch | 29.7 | 42.3 |
| w/+DARP | 30.5 | 42.5 |
| w/+DARP+cRT | 39.7 | 51.0 |
| w/+CReST | 32.5 | 44.7 |
| w/+CReST+LA | 40.9 | 55.9 |
| w/+ABC | 46.9 | 56.1 |
| w/+CoSSL | 43.7 | 53.8 |
| w/+CPE | 47.8 | 58.2 |
| w/+CDMAD | 48.4 | 59.3 |
| w/+DyTrim | **50.6** | **60.0** |

sual fidelity and severe label scarcity. This makes it a promising solution for real-world applications where high-resolution data and abundant labels are often unavailable.

## H.4  MORE RESULTS ON IMAGENET-LT

ImageNet-LT (Liu et al., 2019) is a long-tailed variant of ImageNet, constructed to exhibit a heavy class-imbalance that better reflects real-world data distributions. To assess the scalability of our method on large-resolution inputs ($224 \times 224$), we conducted experiments on ImageNet-LT. Due to hardware constraints, we set the batch size to 2.

As shown in Table 3, CDMAD yields a substantial improvement over the FixMatch baseline, increasing bACC from 20.0% to 35.4%, which highlights the effectiveness of incorporating class-distribution modeling under long-tailed imbalance. Building upon the same baseline, our method

further pushes performance to 37.2%, achieving the best result among all compared approaches. Notably, the improvement over CDMAD remains consistent despite their strong performance, suggesting that our approach introduces complementary benefits rather than merely overlapping with prior re-balancing techniques.

## H.5 RESULTS ON DYNAMIC DATA PRUNING EXPERIMENT

Recently, Infobatch (Qin et al., 2024) provides a no-bias dynamic data pruning method. In this section, we compare it with DyTrim in the framework of CISSL. The experiment is conducted on the CIFAR-10-LT dataset, comparing the settings of $\gamma_l = \gamma_u$ and $\gamma_l \neq \gamma_u$. Specifically, we directly apply the pruning policy of InfoBatch to labeled samples and unlabeled samples without distinction, and the results are shown in the Table. 12 and Table. 13. It can be seen that compared with the proposed method, the pruning policy directly combined with InfoBatch is not consistently effective in all settings. In particular, when $\gamma_l \neq \gamma_u$, it will cause a decrease in accuracy, which is caused by the mismatch in the distribution of labeled samples and unlabeled samples.

Table 12: Comparison of bACC/GM on CIFAR-10-LT.

| Algorithm | CIFAR-10-LT ($\gamma = \gamma_l = \gamma_u$, $\gamma_u$ is assumed to be known) | | |
| --- | --- | --- | --- |
| | $\gamma_l = 50, \gamma_u = 50$ | $\gamma_l = 100, \gamma_u = 100$ | $\gamma_l = 150, \gamma_u = 150$ |
| FixMatch | 79.2 ±0.33 / 77.8 ±0.36 | 71.5 ±0.72 / 66.8 ±1.51 | 68.4 ±0.15 / 59.9 ±0.43 |
| w/+CDMAD | 87.3 ±0.12 / 87.0 ±0.15 | 83.6 ±0.46 / 83.1 ±0.57 | 80.8 ±0.86 / 79.9 ±1.07 |
| w/+InfoBatch* | 87.2 ±0.18 / 86.9 ±0.19 | 84.1 ±0.61 / 83.7 ±0.69 | 81.6 ±0.45 / 80.9 ±0.59 |
| w/+DyTrim | **88.0** ±0.31 / **87.8** ±0.32 | **84.8** ±0.48 / **84.4** ±0.51 | **82.0** ±0.09 / **81.3** ±0.03 |

Table 13: Comparison of bACC/GM on CIFAR-10-LT ($\gamma_l \neq \gamma_u$, $\gamma_u$ is assumed to be unknown).

| Algorithm | CIFAR-10-LT ($\gamma_l = 100$, $\gamma_u$ = Unknown) | | |
| --- | --- | --- | --- |
| | $\gamma_u = 1$ | $\gamma_u = 50$ | $\gamma_u = 150$ |
| FixMatch | 68.9 ±1.95 / 42.8 ±8.11 | 73.9 ±0.25 / 70.5 ±0.52 | 69.6 ±0.60 / 62.6 ±1.11 |
| w/+CDMAD | 87.5 ±0.46 / 87.1 ±0.50 | 85.7 ±0.36 / 85.3 ±0.38 | 82.3 ±0.23 / 81.8 ±0.29 |
| w/+InfoBatch* | 86.4 ±0.63 / 85.9 ±0.73 | 85.5 ±0.33 / 85.1 ±0.37 | 83.3 ±0.08 / 82.8 ±0.11 |
| w/+DyTrim | **88.9** ±0.88 / **88.6** ±1.03 | **86.4** ±0.43 / **86.0** ±0.43 | **83.8** ±0.34 / **83.4** ±0.33 |

## H.6 ABLATION STUDY

**Effectiveness of each component.** We conducted ablation studies on CIFAR-10-LT to assess the contribution of each component in DyTrim, varying the hyperparameter $\gamma = \gamma_l = \gamma_u$ across 50, 100, and 150. As shown in Table. 14, the best performance was achieved when both labeled and unlabeled pruning were combined with rescaling. Removing rescaling led to a bACC drop of 0.8–2.1 points across $\gamma$ values. Excluding either pruning component also reduced performance (*e.g.*, -0.5 and -0.3 at $\gamma = 50$ without unlabeled or labeled pruning, respectively). Removing both pruning strategies resulted in the most significant degradation. These results highlighted the complementary benefits of pruning and rescaling.

## H.7 QUALITATIVE ANALYSES

Since the baseline image could implicitly reflect the bias of the classifier, we argued that by customizing dynamic data pruning methods for labeled and unlabeled data, DyTrim significantly reduced classifier bias while improving performance. To verify this claim, in Figure. 6 (a) and (b), we analyzed the class probabilities predicted on the baseline image using FixMatch+DyTrim, trained on CIFAR-10-LT under various settings. We observed that classifiers trained with DyTrim consistently produced more balanced predictions than CDMAD across all settings, with improved accuracy on tail classes. We defined $r$ as the probability of pruning an unlabeled sample $u_b^m$ when $\mathcal{H}_t^u(u_b^m) < \bar{\mathcal{H}}_t^m$ and $\max(P_\theta(y|\alpha(u_b^m))) \geq \tau$. In Figure. 7, we evaluated different pruning ratios

Table 14: Ablation study for the proposed algorithm on CIFAR-10-LT.

| Labeled Pruning | Unlabeled Pruning | Rescaling | $\gamma_l = \gamma_u = 50$ | | $\gamma_l = \gamma_u = 100$ | | $\gamma_l = \gamma_u = 150$ | |
|---|---|---|---|---|---|---|---|---|
| | | | bACC | GM | bACC | GM | bACC | GM |
| | | | 87.3 | 87.0 | 83.6 | 83.1 | 80.8 | 79.9 |
| ✓ | | | 87.5 | 87.2 | 84.4 | 84.0 | 81.3 | 80.6 |
| | ✓ | ✓ | 87.7 | 87.4 | 84.0 | 83.6 | 81.4 | 80.6 |
| ✓ | ✓ | | 87.2 | 86.9 | 83.6 | 83.1 | 79.9 | 79.0 |
| ✓ | ✓ | ✓ | **88.0** | **87.8** | **84.8** | **84.4** | **82.0** | **81.3** |

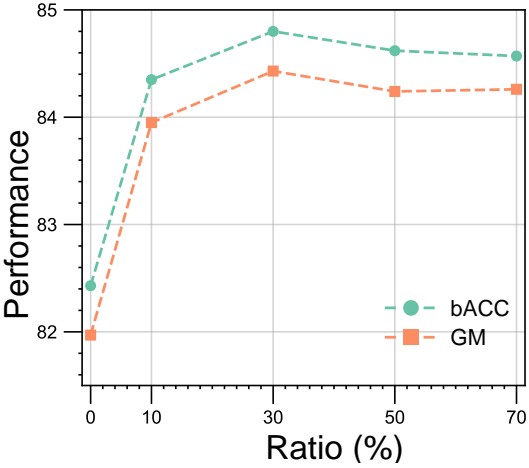

(a) FixMatch+CDMAD     (b) FixMatch+DyTrim     (c) FixMatch+CDMAD     (d) FixMatch+ DyTrim

Figure 6: (a) and (b) present the $\pi_\theta(\mathcal{I})$ using the CDMAD and DyTrim. (c) and (d) present the confusion matrices of the class predictions on test samples on CIFAR-10-LT ($\gamma_l = \gamma_u = 100$).

for unlabeled samples on CIFAR-10-LT. Results showed that setting $r \geq 0.1$ yields higher performance across both architectures, indicating that DyTrim was relatively robust with respect to the hyperparameter $r$, with the best performance achieved when $r = 0.3$.

Figure 7: Evaluation curves of hyper-parameter $r$ on CIFAR-10-LT under bACC and GM.

## H.8 COMPARISON OF CLASS DISTRIBUTIONS BEFORE AND AFTER PRUNING

Figure 8 compares the class distributions before and after applying DyTrim on the labeled, unlabeled and full training sets. Across all three subsets, pruning consistently reduces the proportion of head classes while preserving or slightly increasing the relative proportion of tail classes. This produces a noticeably flatter long-tailed distribution. Unlike traditional pruning methods, which typically remove samples that contribute least to training progress, the behavior of DyTrim is different because the pruning decision is guided by baseline logits and the reliability of pseudo-labels. This tends to eliminate redundant head-class samples and low-quality unlabeled samples while rarely discarding the already scarce tail-class data. Consequently, the resulting effective training subset becomes more balanced without sacrificing essential information from tail classes.

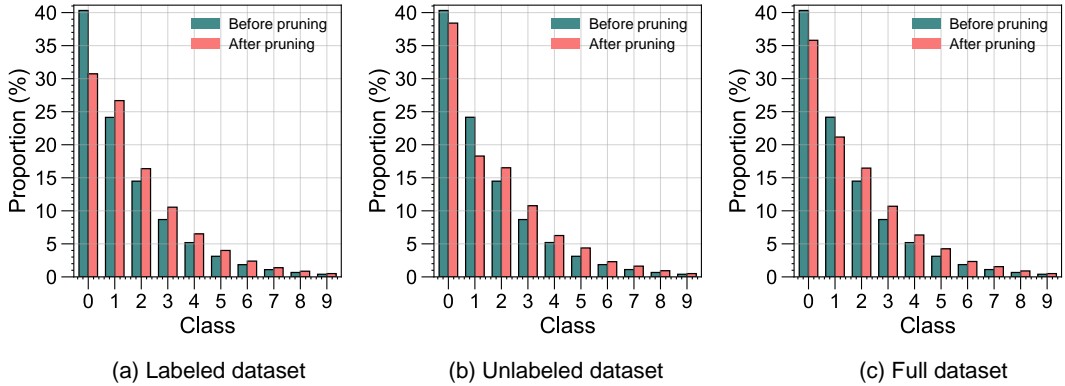

Figure 8: Comparison of class distribution before and after pruning across three datasets: (a) Labeled dataset, (b) Unlabeled dataset, (c) Full dataset.

## H.9 ANALYSIS OF SAMPLE SELECTION FREQUENCY

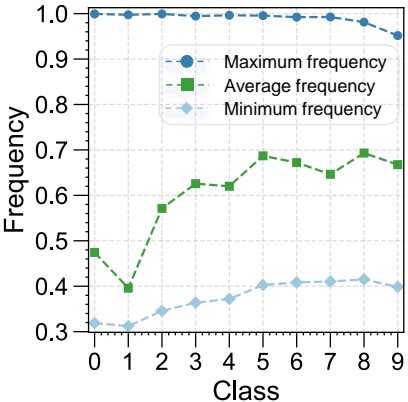

Figure 9: Illustration of per-class maximum, average, and minimum sample selection frequencies during training.

Figure 10: Comparison of class-probability distributions with and without scaling.

Figure 9 reports the maximum, average and minimum sample selection frequencies for each class. Three observations emerge clearly. First, the maximum frequency remains close to 1 for all classes, which indicates that each class contains at least a subset of highly informative samples that are almost always preserved during pruning. Second, the average frequency increases from head to tail classes, showing that

Table 15: Comparison of bACC and GM on CIFAR-10-LT on fixed and dynamic scaling factors.

| | CIFAR-10-LT | |
|---|---|---|
| Algorithm | $\gamma_l = 100, \gamma_u = 100$ | $\gamma_l = 100, \gamma_u = 1/100$ |
| Fixed Scaling | 84.8 / 84.4 | 78.2 / 76.7 |
| Dynamic Scaling | **84.9 / 84.4** | **78.9 / 78.1** |

DyTrim removes a larger fraction of redundant samples from majority classes while retaining more samples in minority classes. This behavior matches the intended effect of mitigating class dominance through selective pruning. Third, the minimum frequency stays within a narrow and relatively high range across all classes, suggesting that even the least frequently selected samples are not entirely discarded. This prevents the severe under-sampling of tail classes that often occurs in traditional pruning strategies.

## H.10 EFFECT OF SCALING STRATEGIES ON CLASS-BIAS

Figure 10 compares the class probability distributions obtained with and without the proposed scaling strategy. Although the two curves differ for several head and mid-frequency classes, the overall

decay pattern remains consistent, and the probabilities of head classes do not increase when scaling is applied. This shows that the scaling mechanism does not intensify the influence of high confidence samples and preserves the long-tailed structure shaped by DyTrim.

Additionally, to provide each class with an adaptive scaling factor that assigns smaller scaling to head classes and larger scaling to tail classes, we further compare fixed and dynamic scaling in Table 15. Dynamic scaling leads to higher bACC and GM under both matched and mismatched imbalance conditions, indicating that adapting the scaling factor to the current pruning state yields a more reliable correction for changes in the effective batch size. The dynamic scaling factor is computed as $1 - \pi_\theta(\mathcal{I})_{\hat{q}_b} + 1/(1 - r)$, which stabilizes the loss magnitude during training and prevents undesirable shifts toward majority class predictions.

### H.11 Dynamics of Sample Score Across Head and Tail Classes

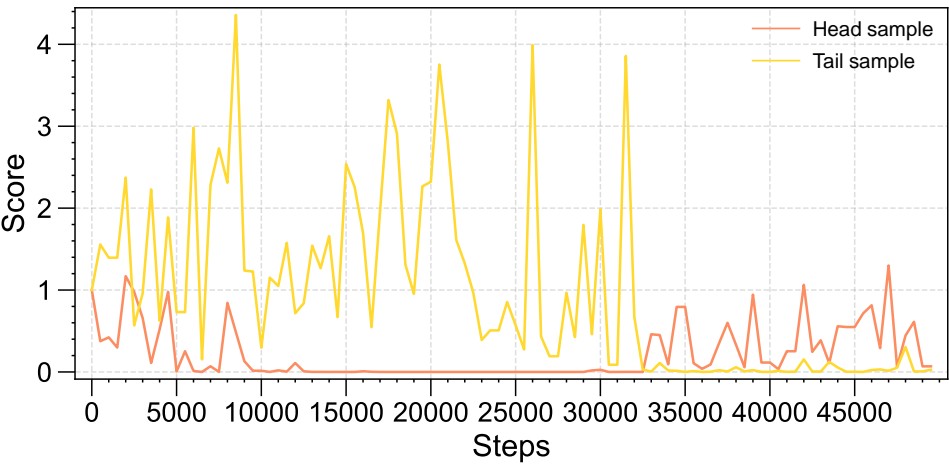

Figure 11: Scores of a representative head class sample and a representative tail class sample over the first 50,000 training steps, recorded every 500 steps.

Figure 11 shows the dynamics of scores for one head class sample and one tail class sample over the first 50,000 training steps. The two trajectories exhibit a clear contrast. The tail class sample maintains consistently higher and more volatile scores throughout training, reflecting its larger contribution to reducing class bias and its higher utility for updating the classifier. In comparison, the head class sample quickly drops to very low scores and remains close to zero for most of training. This indicates that the head sample becomes saturated early and provides little additional information, which aligns with the design of DyTrim that aims to remove redundant head class samples.

### H.12 Pruning Dynamics Across Labeled and Unlabeled Datasets

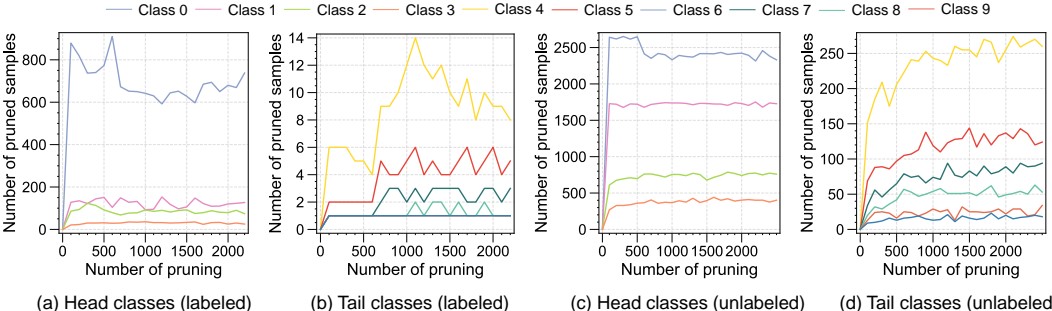

Figure 12: Number of pruned samples for each class across training process on CIFAR-10-LT. (a) and (b) show the evolution for head and tail classes in the labeled set, and (c) and (d) show the corresponding results for the unlabeled set. Each curve indicates how many samples of a given class have been removed up to each pruning step, recorded every 100 iterations.

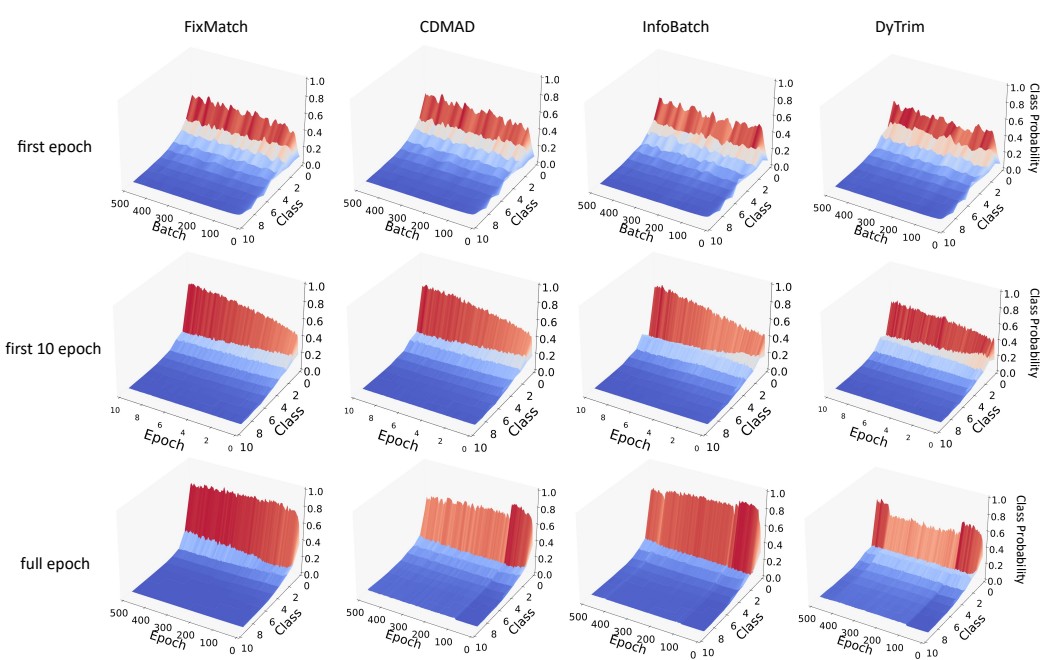

Figure 13: Comparison of the change of logits's probability distribution $\pi_\theta(\mathcal{I})$ for the baseline image on CIFAR-10-LT with $\gamma_l = \gamma_u = 100$ across different CISSL methods.

Figure 12 reports the number of pruned samples per class over the course of training. The results from both the labeled and unlabeled subsets exhibit a consistent pattern. Head classes experience a rapid increase in pruned samples at the beginning of training and maintain high pruning counts throughout the process, which reflects the large amount of redundant information contained in these majority classes. In contrast, tail classes show much slower growth curves with considerably lower pruning volumes, indicating that DyTrim preserves most of the scarce minority samples and avoids aggravating the long-tailed imbalance. The same trend appears in the unlabeled subset, where head classes accumulate substantially more pruned samples due to the prevalence of high confidence but less informative pseudo-labeled instances. These observations confirm that DyTrim adaptively modulates pruning according to class frequency and sample utility, removing redundant head-class samples while retaining informative tail-class data.

# I    VISUALIZATION

## I.1    DETAILS OF THE CHANGE OF LOGITS'S PROBABILITY DISTRIBUTION

In this section, we conduct some visualization experiments to demonstrate the advantages of the DyTrim in debiasing and improving classifier performance. We first analyze the change of logits's probability distribution $\texttt{Softmax}(g_\theta(\mathcal{I}))$ for the baseline image on CIFAR-10-LT with $\gamma_l = \gamma_u = 100$ for fixmatch, CDMAD, and the DyTrim as shown in Figure. 13. It can be seen intuitively that in the first epoch, the classifier has bias due to the imbalance of categories in the data. This situation increases significantly with the number of network training times, as shown in the second column of the figure. However, we can see that DyTrim can effectively slow down the increase of this bias. Furthermore, after the model is fully trained for 500 epochs, it can be seen that after the 100th epoch, CDMAD starts to use the baseline image for post-hoc debiasing, which significantly reduces the representation of the model. However, by dynamically pruning the data set, DyTrim obtains a more distinct debias effect as shown in Figure. 14.

## I.2    DETAILS OF THE CHANGE OF LOGITS'S PROBABILITY DISTRIBUTION

Figure. 15 and Figure. 16 compare the confusion matrices of the class predictions on the test set of CIFAR-10 using (a) FixMatch, (b) FixMatch+Infobatch, (c) FixMatch+CDMAD, and (d) Fix-

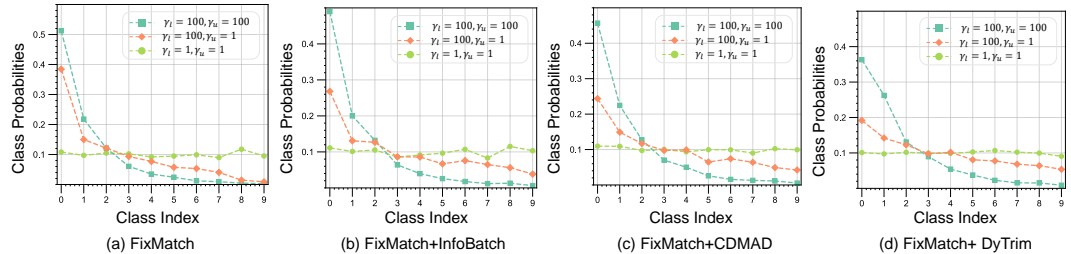

Figure 14: Class probabilities predicted on a baseline image using (a) FixMatch, (b) FixMatch+InfoBatch, (c) FixMatch+CDMAD, (d) FixMatch+DyTrim.

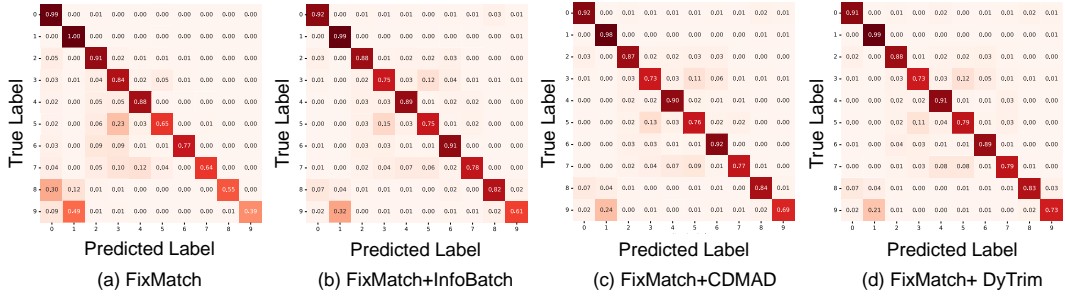

Figure 15: Confusion matrices of the class predictions on the test set of CIFAR-10 using (a) FixMatch, (b) FixMatch+InfoBatch, (c) FixMatch+CDMAD, and (d) FixMatch+DyTrim trained on CIFAR-10-LT under $\gamma_l = 100$ and $\gamma_u = 100$.

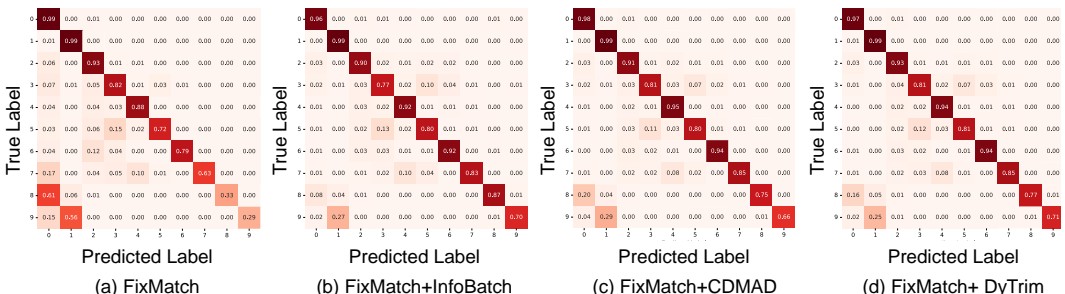

Figure 16: Confusion matrices of the class predictions on the test set of CIFAR-10 using (a) FixMatch, (b) FixMatch+InfoBatch, (c) FixMatch+CDMAD, and (d) FixMatch+DyTrim trained on CIFAR-10-LT under $\gamma_l = 100$ and $\gamma_u = 1$.

Match+DyTrim trained on CIFAR-10-LT under $\gamma_l = 100, \gamma_u = 1, 100$. FixMatch+DyTrim made more balanced predictions across classes. Furthermore, we also conducted experiments under a balanced setting ($\gamma = \gamma_1 = \gamma_u = 1$), as shown in Figure. 17. The results show that even under a balanced data distribution, DyTrimcan still achieve better results on the pruned dataset than methods such as CDMAD trained on the full dataset.

Similar to confusion matrices, we also compare t-distributed stochastic neighbor embedding (t-SNE) of representations obtained for the test set of CIFAR-10 using FixMatch, FixMatch+CDMAD, FixMatch+InfoBatch, and FixMatch+DyTrim trained on CIFAR-10 with $\gamma_l = 100$ and $\gamma_u = 1, 100$(**unknown** $\gamma_u$), where different colors indicate different classes in CIFAR-10 Figure. 18, Figure. 19. We can observe that the representations obtained using FixMatch+DyTrim are separated into classes with clearer boundaries compared the those from FixMatch and CDMAD. This is probably because CDMAD appropriately refined the biased pseudo-labels and used them for training, whereas FixMatch failed to learn the representations properly because they used the biased pseudo-labels for training. These results demonstrate that the quality of representations can be improved by using well-refined pseudo-labels for training.

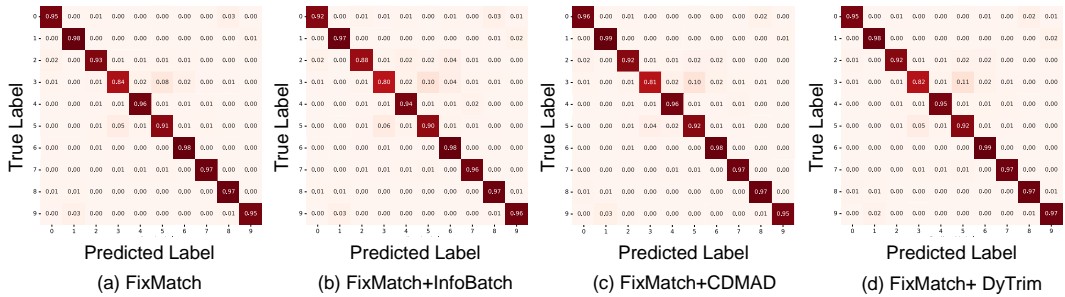

Figure 17: Confusion matrices of the class predictions on the test set of CIFAR-10 using (a) Fix-Match, (b) FixMatch+InfoBatch, (c) FixMatch+CDMAD, and (d) FixMatch+DyTrim trained on CIFAR-10-LT under $\gamma_l = 1$ and $\gamma_u = 1$.

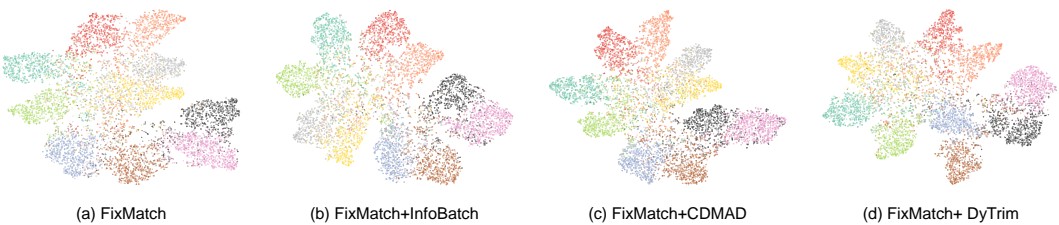

Figure 18: t-SNE of representations obtained for the test set of CIFAR-10 using (a) FixMatch, (b) FixMatch+InfoBatch, (c) FixMatch+CDMAD, and (d) FixMatch+DyTrim trained on CIFAR-10-LT under $\gamma_l = 100$ and $\gamma_u = 100$.

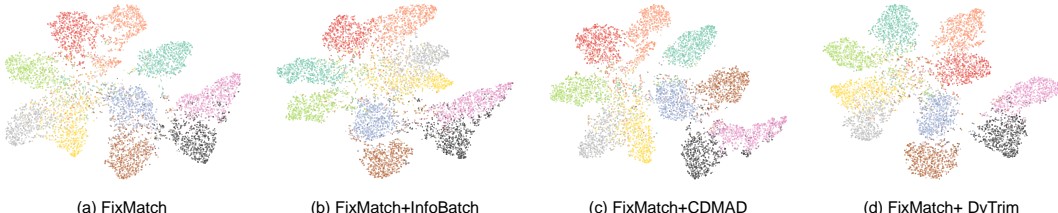

Figure 19: t-SNE of representations obtained for the test set of CIFAR-10 using (a) FixMatch, (b) FixMatch+InfoBatch, (c) FixMatch+CDMAD, and (d) FixMatch+DyTrim trained on CIFAR-10-LT under $\gamma_l = 100$ and $\gamma_u = 1$.

## J    LIMITATION

A key limitation of our method is its reliance on a task-irrelevant baseline image as a bias indicator. If this baseline image is used as a training sample, it may no longer reflect the accumulated bias, reducing the effectiveness of our debiasing mechanism. Additionally, our framework does not account for architectures with auxiliary classification heads or semi-supervised methods based on mixup-style (Zhang et al., 2017) interpolations, limiting DyTrim's applicability to these models. Extending our approach to these settings is an interesting avenue for future work.

## K    USE OF LLMS

Large language models (LLMs) were used solely to assist with minor language polishing during manuscript preparation. All scientific components of this work, including the design of experiments, data processing, analysis, and interpretation, were carried out entirely by the authors using established computational methods and human expertise, without reliance on automated reasoning or model-generated content.

