# Learning Dynamics of Logits Debiasing for Long-Tailed Semi-Supervised Learning

## Abstract

Long-tailed distributions are prevalent in real-world semi-supervised learning (SSL), where pseudo-labels tend to favor majority classes, leading to degraded generalization. Although numerous long-tailed SSL (LTSSL) methods have been proposed, the underlying mechanisms of class bias remain underexplored. In this work, we investigate LTSSL through the lens of learning dynamics and introduce the notion of baseline images to characterize accumulated bias during training. We provide a step-wise decomposition showing that baseline predictions are determined solely by shallow bias terms, making them reliable indicators of class priors. Building on this insight, we propose a novel framework, DyTrim, which leverages baseline images to guide data pruning. Specifically, we perform class-aware pruning on labeled data to balance class distribution and label-agnostic soft pruning with confidence filtering on unlabeled data to mitigate error accumulation. Theoretically, we show that our method implicitly realizes risk reweighting, effectively suppressing class bias. Extensive experiments on public benchmarks show that DyTrim consistently enhances the performance of existing LTSSL methods by improving representation quality and prediction accuracy.

## 1 Introduction

Semi-supervised learning (SSL), exemplified by FixMatch (Sohn et al., 2020), has been proven to demonstrate significant generalization advantages over supervised learning, particularly in deep neural networks (Li et al., 2025). However, many existing SSL variants (*e.g.*, FlexMatch; Zhang et al., 2021) implicitly assume that both labeled and unlabeled data are drawn from a balanced class distribution. In practice, datasets commonly exhibit a long-tailed label distribution, leading to *biased pseudo-label* toward majority classes. This discrepancy poses significant challenges to the effectiveness of SSL algorithms on real-world datasets.

Recent studies on long-tailed semi-supervised learning (LTSSL) have emerged to mitigate pseudo-label bias caused by class imbalance in both labeled and unlabeled data. These methods range from distribution alignment (Wei et al., 2021; Kim et al., 2020), data rebalancing (Fan et al., 2022; Lee et al., 2021), logit adjustment variants (Wei & Gan, 2023; Zhou et al., 2024), to foundation model-based methods (*e.g.*, LADaS; Zheng et al., 2025). In particular, the approach employ baseline image was introduced as a simple yet efficient tool to quantify classifier bias by CDMAD (Lee & Kim, 2024), which has attracted considerable attention in the community (Xing et al., 2025). However, the underlying mechanisms of how class bias emerges and why existing approaches can mitigate it remain largely unexplored and poorly understood. That also prevents us from exploring a principle-based method to improve performance.

In this paper, we analyze the underlying mechanisms of class debiasing through an innovative lens of learning dynamics, investigating how an input affects the generation of biased pseudo-labels. We first point out that in the training processes of LTSSL, the logits of the baseline image serve as an indicator of the accumulated influence of the network's bias term. We further propose a framework that formalizes the learning dynamics of semi-supervised learning by decomposing the change of the model's prediction on the baseline image into three terms. Under this framework, many existing debiasing methods for class imbalance can be unified.

Furthermore, our analysis of bias accumulation dynamics motivates a pruning-based class debiasing framework. For labeled data, we compute class-wise pruning ratios to rebalance samples. For un-

labeled data, we apply a label-agnostic criterion that prunes low-confidence, inconsistent samples. Beyond empirical gains or ad-hoc analysis, DyTrim provide a principle-based theoretical guarantees that clarify how the proposed method can alleviate class biasing and why pruning enhances generalization. Extensive experiments confirm that DyTrim consistently enhances LTSSL performance across standard benchmarks.

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

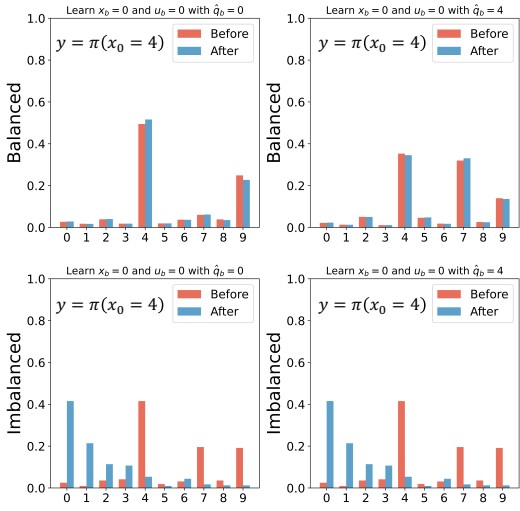

Figure 1: The per-step semi-supervised learning dynamics and the accumulated influence in an MNIST experiment.

loop. The bottom row shows that under class imbalance, such accumulated influence can drive the classifier to consistently predict the majority class (here 0), regardless of the true label. This confirms the implication of our dynamics analysis: in SSL, the effect of labeled data is mediated through pseudo-labels, so local errors can be amplified rather than averaged out, leading to catastrophic bias.

## 3.2 Learning Dynamics Analysis of Accumulated Bias Under Class Imbalance

The aforementioned phenomenon, together with the learning dynamics of the semi-supervised framework, illustrates how class imbalance accumulates into systematic bias. While per-update dynamics capture the influence of individual samples on predictions, they fall short of reflecting the global effect of imbalance. This motivates the search for an indicator that bridges class-imbalance bias with the underlying learning dynamics. We propose to use a *baseline image* $\mathcal{I}$ as such an indicator. To justify this choice, we analyze its theoretical properties in both linear and deep settings, and then incorporate it into the per-step influence decomposition.

**Baseline image and its invariance property.** For simplicity, we first consider a two-layer MLP with no bias in the first layer and a bias vector $\boldsymbol{b} \in \mathbb{R}^C$ in the output layer $h(x) = h^{(2)} \circ h^{(1)}(x)$, where $h^{(1)}(x) = \sigma(\boldsymbol{W}_1 x)$ and $h^{(2)} = \boldsymbol{W}_2 x + \boldsymbol{b}$. This setting allows us to isolate and examine the predicted class probability $\pi_\theta(\mathcal{I})$ of a baseline image. For a baseline image $\mathcal{I} \in \mathbb{R}^d$, we have

$$h(\mathcal{I}) = \boldsymbol{W}_2 h^{(1)}(\mathcal{I}) + \boldsymbol{b}. \tag{7}$$

In modern neural networks, the explicit bias term $\boldsymbol{b}$ is typically absorbed into the normalization layer, e.g., BatchNorm, LayerNorm, while other layers are usually set without bias. Without loss of generality, we take BatchNorm as an example for analysis. Since the BatchNorm transformation can be equivalently viewed as an affine linear layer with learnable parameters, we may replace $h^{(2)}$ with a `BatchNorm`$(\cdot)$ layer, *i.e.*,

$$h(\mathcal{I}) = \mathtt{BatchNorm}\big(h^{(1)}(\mathcal{I})\big) = \frac{h^{(1)}(\mathcal{I}) - \mathbb{E}[h^{(1)}(\mathcal{I})]}{\sqrt{\mathrm{Var}[h^{(1)}(\mathcal{I})]}} \cdot \boldsymbol{W}_2 + \boldsymbol{b}. \tag{8}$$

This replacement highlights that the baseline image prediction $\pi_\theta(\mathcal{I})$ is directly governed by the BN bias $\boldsymbol{b}$, thus allowing us to focus on its role in encoding and accumulating class-imbalance bias. We now state the main results regarding the $\pi_\theta(\mathcal{I})$ below:

**Proposition 2** (Invariance of baseline image under affine normalization). *Let $\mathcal{I} = k \cdot \mathbf{1}_d$ be a baseline image, where $k \in \{0, 1, \ldots, 255\}$ and $\mathbf{1}_d \in \mathbb{R}^d$ is an all-one vector. Suppose the output of the first hidden transformation is normalized by a normalization layer (e.g., BatchNorm, LayerNorm, InstanceNorm, or GroupNorm) with affine parameters $(\boldsymbol{W}_2, \boldsymbol{b})$. Then the logits $h(\mathcal{I})$ are independent of $k$ and reduce to*

$$h(\mathcal{I}) = \boldsymbol{b}, \quad \pi_\theta(\mathcal{I}) = \mathtt{Softmax}(\boldsymbol{b}). \tag{9}$$

One can immediately notice that $\pi_\theta(\mathcal{I})$ in Eq. (9) does not contain any term related to the pixel value $k$ of $\mathcal{I}$. This observation implies that the representation $\pi_\theta(\mathcal{I})$ of a baseline image is entirely determined by the BatchNorm bias term $\boldsymbol{b}$, and is invariant to the actual pixel value $k$.

Building upon this invariance, we now establish a connection between the baseline image and the underlying class distribution. Specifically, for the classifier formulation in Eq. (8), we show that the logits of the baseline image encode the class-imbalance ratio of the training data, thereby providing a direct bridge between $\pi_\theta(\mathcal{I})$ and the long-tailed class prior.

**Theorem 1** (Bias as the conditional distribution prior). *Assume the model $h(x)$ which characterized in Eq. (8), is trained by cross-entropy,*

$$\mathcal{L} = \mathbb{E}_{(x,y)}\big[-y^\top \log \mathtt{Softmax}(h(x))\big]. \tag{10}$$

*At a population risk minimizer $(\boldsymbol{W}_2^\star, \boldsymbol{b}^\star)$ we have*

$$\hat{p}^\star(x) = P(y \mid x), \qquad \hat{p}^\star(\mathcal{I}) = \mathtt{Softmax}\big(\boldsymbol{b}^\star\big) = P\big(y \,\big|\, \tfrac{h^{(1)}(\mathcal{I}) - \mathbb{E}[h^{(1)}(\mathcal{I})]}{\sqrt{\mathrm{Var}[h^{(1)}(\mathcal{I})]+\epsilon}} = \boldsymbol{0}\big). \tag{11}$$

*In particular, for solid-color $\mathcal{I}$ satisfying Proposition 2, the baseline prediction equals the conditional class distribution at the "normalized-zero" feature state, thereby encoding the class prior induced during training.*

Therefore, $\pi_\theta(\mathcal{I})$ can naturally serve as a proxy for the *accumulated bias* of the model, providing a bridge between class imbalance and learning dynamics.

**Per-step influence decomposition of the baseline image.** Let the estimate of the underlying class prior $P_\theta(y|\cdot)$ be denoted by $\pi$. Then we can track the change in the model's confidence by observing $\log \pi_\theta(y|\cdot)$. Then learning dynamics become,

$$\Delta \log \pi^t(y|\mathcal{I}) \triangleq \log \pi_{\theta^{t+1}}(y|\mathcal{I}) - \log \pi_{\theta^t}(y|\mathcal{I}). \tag{12}$$

**Proposition 3.** *Let $\pi = \texttt{Softmax}(z)$ and $z = g_\theta(x)$. The one-step dynamics decompose as*

$$\Delta \log \pi^t(y \mid \mathcal{I}) = -\eta \mathcal{T}^t(\mathcal{I}) \mathcal{K}^t(\mathcal{I}, x) \mathcal{G}^t(x, y) + \mathcal{O}(\eta^2 \|\nabla_\theta z(x)\|_{\text{op}}^2), \tag{13}$$

*where $\mathcal{T}^t(\mathcal{I}) = \nabla_z \log \pi^t(\mathcal{I}) = I - \mathbf{1}\pi_{\theta^t}^T(\mathcal{I})$, $\mathcal{K}^t(\mathcal{I}, x) = \left(\nabla_\theta z(\mathcal{I})\big|_{\theta^t}\right)\left(\nabla_\theta z(x)\big|_{\theta^t}\right)^T$ is the empirical Neural Tangent Kernel (NTK) of the logit network $z$, and $\mathcal{G}^t(x, y) = \nabla_z \mathcal{L}(x, y)\big|_{z^t}$ (see Appendix F for details).*

### 3.3 Effect of the baseline image for guiding data pruning

The training objective can be interpreted as the minimization of the empirical risk $\mathcal{L}$. Assuming that all labeled samples $x_b^n$ from $\mathcal{X}$ and unlabeled samples $u_b^m$ from $\mathcal{U}$ are drawn from continuous distributions $\rho^l(x_b^n)$ and $\rho^u(u_b^m)$, respectively, the training objective can be formulated as:

$$\arg \min_{\theta \in \Theta} \mathbb{E}_{x_b^n \in \mathcal{X}, u_b^m \in \mathcal{U}} [\mathcal{L}(x_b^n, u_b^m; \theta)] = \int_{x_b^n} \mathcal{L}_{sup}(x_b^n, \theta)\rho^l(x_b^n)dx_b^n + \int_{u_b^m} \mathcal{L}_{con}(u_b^m, \theta)\rho^l(u_b^m)du_b^m. \tag{14}$$

After applying a data pruning policy, we sample $x_b^n$ and $u_b^m$ to obtain the labeled pruned subset $\mathcal{S}_t^l$ and the unlabeled pruned subset $\mathcal{S}_t^u$, according to the labeled pruning probabilities $\mathcal{P}_t^l(x_b^n)$ and unlabeled pruning probabilities $\mathcal{P}_t^u(u_b^m)$, respectively. For the labeled samples, we directly optimize over the pruned subset $\mathcal{S}_t^l$ without reweighting the loss terms. Notably, the class-aware pruning probability $r_c = \pi_\theta(\mathcal{I})_c$ inherently adjusts $\mathcal{S}_t^l$ toward an asymptotically balanced class distribution. By retaining more samples from minority classes (lower $r_c$) and pruning more samples from majority classes (higher $r_c$), the pruned subset $\mathcal{S}_t^l$ naturally mitigates class imbalance. As a result, even without explicit rescaling, the empirical risk over $\mathcal{S}_t^l$ approximates:

$$\arg \min_{\theta \in \Theta} \mathbb{E}_{x_b^n \in \mathcal{S}_t^l} [\mathcal{L}_{sup}(x_b^n, \theta)] \propto \frac{1 - \mathcal{P}_t^l(x_b^n)}{c_t^l} \int_z \mathcal{L}_{sup}(x_b^n, \theta)\rho_l(x_b^n)dx_b^n, \tag{15}$$

where $c_t^l = \mathbb{E}_{x_b^n \sim \rho_l}[1 - \mathcal{P}_t^l(x_b^n)]$. The term $\frac{1 - \mathcal{P}_t^l(z)}{c_t^l}$ acts as an *implicit reweighting* due to the class-aware pruning policy. For unlabeled samples, pruning with uniform probability $r$ and rescaling losses by $\gamma_t(u) = \frac{1}{1 - \mathcal{P}_t^u(u)}$ yields

$$\arg \min_{\theta \in \Theta} \mathbb{E}_{u_b^m \in \mathcal{S}_t^u} [\gamma_t(u_b^m)\mathcal{L}_{con}(u_b^m, \theta)] \propto \frac{1}{c_t^u} \int_z \mathcal{L}_{con}(u_b^m, \theta)\rho^l(u_b^m)du_b^m, \tag{16}$$

where $c_t^u = \mathbb{E}_{u_b^m \sim \rho_u}[1 - \mathcal{P}_t^u(u_b^m)]$. Crucially, even with uniform pruning rates, the interplay of consistency regularization and confidence thresholding ensures $\mathcal{S}_t^u$ to be implicitly balanced, thus training on $\mathcal{S}_t^u$ with rescaled factor $\gamma_t(u_b^m)$ could achieve a better result as training on the $\mathcal{U}$.

## 4 DyTrim: A Baseline Image Guided Data Pruning Framework for CISSL

The theoretical results in Section 3 suggest that the distribution of the baseline image's logits is affected by imbalanced data and directly acts on the bias term in the shallow layers. This imbalance directly causes the model to produce bias. Fortunately, we indicated that this bias can be effectively reduced if the data is pruned to be more balanced. Based on these insights, we propose the algorithm DyTrim, which extends the data pruning by incorporating guidance from the baseline image's logits to select a balanced subset for the training of CISSL, as illustrated in Figure 2.

**Dynamic data pruning for CISSL.** We use $\mathcal{X} = \{(x^n, y^n)\}_{n=1}^N$ to denote the labeled set and $\mathcal{U} = \{u^m\}_{m=1}^M$ for unlabeled set. Critically, the distribution mismatch between $\mathcal{X}$ and $\mathcal{U}$ necessitates separate scoring mechanisms for labeled and unlabeled samples—unlike conventional supervised dynamic pruning methods that assume identical data distributions. To this end, we define step-dependent scoring functions $\mathcal{H}_t^l$ for labeled samples and $\mathcal{H}_t^u$ for unlabeled samples, which dynamically quantify sample utility at training step $t$. For the dynamic pruning process, samples are discarded by the step-dependent pruning probabilities $\mathcal{P}_t^l$ and $\mathcal{P}_t^u$:

$$\mathcal{P}_t^l(x; \mathcal{H}_t^l) = \mathbb{1}(\mathcal{H}_t^l(x), \prec_{r_c} \mathcal{H}_{c,t}^l); \quad \text{and} \quad \mathcal{P}_t^u(u; \mathcal{H}_t^u) = \mathbb{1}(\mathcal{H}_t^u(u), \bar{\mathcal{H}}_t^u), \tag{17}$$

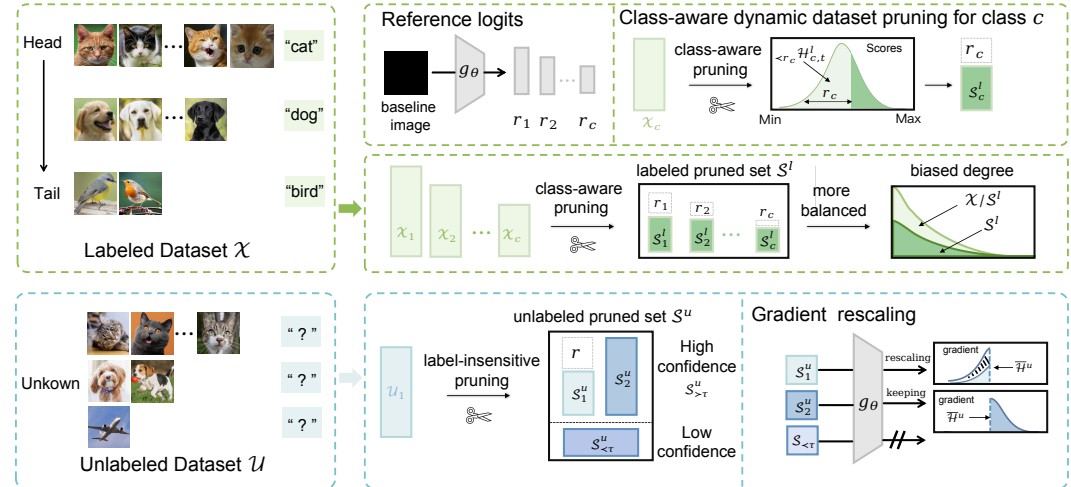

Figure 2: Illustration of the proposed DyTrim framework. DyTrim mainly consists of two operations, named labeled pruning and unlabeled pruning. $\prec_{r_c}\mathcal{H}^l_{c,t}$ and $\bar{\mathcal{H}}^u_t$ denote the adaptive thresholds of scores of labeled samples and unlabeled samples, with slight abuse of symbols. $\mathcal{S}^u_{\prec\tau}$ denote the low confidence unlabeled sample which $p^*(u^m_b) \geq \tau$. Labeled pruning provides a class-aware pruning policy for each sample from class $c$. Unlabeled pruning provides a random pruning policy from the original data $\mathcal{U}_1$ and uses a gradient rescaling strategy ($\times 1/(1-r)$ for which sample from $s^u_1$ is selected to prune) to keep the approximately same gradient expectation.

where $\prec_{r_c}\mathcal{H}^l_{c,t}$ and $\bar{\mathcal{H}}^u_t$ are adaptive thresholds, $\mathbb{1}(\cdot, \cdot)$ is the indicator function. Thus, two dynamically pruned datasets $\mathcal{S}^l_t$ and $\mathcal{S}^u_t$ are formed for labeled and unlabeled datasets, respectively.

**Dynamic pruning for labeled data.** Since the labeled data follow a long-tailed class distribution, we design a class-aware pruning policy $\mathcal{P}^l_t$ guided by $\pi_\theta(\mathcal{I})$. Critically, the classifier's pseudo-labels are primarily influenced by the labeled samples, which introduce bias toward majority classes. Since Proposition 2 shows that the baseline image has invariance to solid-color intensity, from first principles, we leverage the logits from a **black image** $\mathcal{I}$ to calibrate pruning probabilities. Given the labeled dataset $\mathcal{X}$ in the $t$-th epoch, a class-aware pruning probability is assigned to each sample based on its score, which is formulated as:

$$\mathcal{P}^l_t(x^n_b) = \begin{cases} 1 & \mathcal{H}^l_t(x^n_b) \in \prec_{r_c}\mathcal{H}^l_{c,t}, \\ 0 & \mathcal{H}^l_t(x^n_b) \notin \prec_{r_c}\mathcal{H}^l_{c,t}, \end{cases} \tag{18}$$

where $\prec_{r_c}\mathcal{H}^l_{c,t}$ denotes the $r_c \times N_c$ smallest scoring values of the class $c$ and $r_c = \pi_\theta(\mathcal{I})_c$ is the class-aware pruning probability. The labeled scoring function $\mathcal{H}^l_{c,t}(x^n_b)$ is defined using the supervised loss $\mathcal{L}_{sup}(x^n_b, y^n_b)$ to quantify sample utility. Specifically, we exploit the pruning policy to prune samples based on their scores. Then, for the pruned labeled samples, their scores remain unmodified as previously. For the remaining samples, their scores are updated by the losses in the current epoch. To ensure dynamic adaptation:

$$\mathcal{H}^l_{c,t+1}(x^n_b) = \begin{cases} \mathcal{H}^l_{c,t}(x^n_b) & x^n_b \in \mathcal{X} n \mathcal{S}^l, \\ \mathcal{L}_{sup}(x^n_b) & x^n_b \in \mathcal{S}^l. \end{cases} \tag{19}$$

**Dynamic pruning for unlabeled data.** While the distribution of the label of the unlabeled data and its imbalance ratio $\gamma_u$ are unknown. To address the uncertainty and bias of pseudo-labels, we design a label-insensitive soft pruning policy $\mathcal{P}^u_t$ inspired by (Qin et al., 2024), which introduces randomness and gradient scaling into the pruning process. Specifically, for an unlabeled dataset $\mathcal{U}$ at the $t$-th epoch, a pruning probability is assigned to each sample based on its score, which is formulated as:

$$\mathcal{P}^u_t(u^m_b) = \begin{cases} r & \mathcal{H}^u_t(u^m_b) < \bar{\mathcal{H}}^m_t \text{ and } p^*(u^m_b) \geq \tau, \\ 0 & \mathcal{H}^u_t(u^m_b) \geq \bar{\mathcal{H}}^u_t \text{ or } p^*(u^m_b) < \tau, \end{cases} \tag{20}$$

where $\bar{\mathcal{H}}^u_t$ is the adaptive threshold and $r$ is a randomized pruning rate, $\tau$ is the confidence threshold $\tau$ and $p^*(u^m_b) = \max(\text{softmax}(g^*_\theta(\alpha(u^m_b))))$ denote the debiased pseudo-label confidence. For

Table 1: Comparison of bACC/GM on CIFAR-10-LT.

| Algorithm | CIFAR-10-LT ($\gamma = \gamma_l = \gamma_u$, $\gamma_u$ is assumed to be known) | | |
|---|---|---|---|
| | $\gamma = 50$ | $\gamma = 100$ | $\gamma = 150$ |
| Vanilla | 65.2 ±0.05 / 61.1 ±0.09 | 58.8 ±0.13 / 58.2 ±0.11 | 55.6 ±0.43 / 44.0 ±0.98 |
| Re-sampling | 64.3 ±0.48 / 60.6 ±0.67 | 55.8 ±0.47 / 45.1 ±0.30 | 52.2 ±0.05 / 38.2 ±1.49 |
| LDAM-DRW | 68.9 ±0.07 / 67.0 ±0.08 | 62.8 ±0.17 / 58.9 ±0.60 | 57.9 ±0.20 / 50.4 ±0.30 |
| cRT | 67.8 ±0.13 / 66.3 ±0.15 | 63.2 ±0.45 / 59.9 ±0.40 | 59.3 ±0.10 / 54.6 ±0.72 |
| FixMatch | 79.2 ±0.33 / 77.8 ±0.36 | 71.5 ±0.72 / 66.8 ±1.51 | 68.4 ±0.15 / 59.9 ±0.43 |
| w/+DARP+cRT | 85.8 ±0.43 / 85.6 ±0.56 | 82.4 ±0.26 / 81.8 ±0.17 | 79.6 ±0.42 / 78.9 ±0.35 |
| w/+CReST+LA | 85.6 ±0.36 / 81.9 ±0.45 | 81.2 ±0.70 / 74.5 ±0.99 | 71.9 ±2.24 / 64.4 ±1.75 |
| w/+ABC | 85.6 ±0.26 / 85.2 ±0.29 | 81.1 ±1.14 / 80.3 ±1.29 | 77.3 ±1.25 / 75.6 ±1.65 |
| w/+CoSSL | 86.8 ±0.30 / 86.6 ±0.25 | 83.2 ±0.49 / 82.7 ±0.60 | 80.3 ±0.55 / 79.6 ±0.57 |
| w/+SAW+LA | 86.2 ±0.15 / 83.9 ±0.35 | 80.7 ±0.15 / 77.5 ±0.21 | 73.7 ±0.06 / 71.2 ±0.17 |
| w/+Adsh | 83.4 ±0.06 / 82.9 ±0.13 | 76.5 ±0.35/ 74.8 ±0.34 | 71.5 ±0.30 / 68.8 ±0.35 |
| w/+DebiasPL | 85.6 ±0.20 / 85.2 ±0.23 | 80.6 ±0.50 / 79.9 ±0.57 | 76.6 ±0.12 / 75.8 ±0.71 |
| w/+UDAL | 86.5 ±0.29 / 86.2 ±0.26 | 81.4 ±0.39 / 80.6 ±0.38 | 77.9 ±0.33 / 75.8 ±0.71 |
| w/+L2AC | 86.6 ±0.31 / 86.7 ±0.30 | 82.1 ±0.57 / 81.5 ±0.64 | 77.6 ±0.53 / 75.8 ±0.71 |
| w/+CDMAD | 87.3 ±0.12 / 87.0 ±0.15 | 83.6 ±0.46 / 83.1 ±0.57 | 80.8 ±0.86 / 79.9 ±1.07 |
| w/+DYTRIM | **88.0** ±0.31 / **87.8** ±0.32 | **84.8** ±0.48 / **84.4** ±0.51 | **82.0** ±0.09 / **81.3** ±0.03 |
| FlexMatch | 72.6 ±0.72 / 70.2 ±0.88 | 67.7 ±0.73 / 63.6 ±1.27 | 62.6 ±0.63 / 56.1 ±1.13 |
| w/+CDMAD | 74.4 ±0.82 / 73.0 ±1.12 | 68.4 ±0.46 / 66.8 ±0.53 | 67.0 ±0.52 / 63.2 ±0.44 |
| w/+DYTRIM | **77.2** ±0.42 / **76.2** ±0.44 | **70.7** ±0.49 / **67.8** ±0.70 | **68.6** ±0.22 / **66.3** ±0.07 |
| FreeMatch | 71.9 ±0.24 / 69.4 ±0.61 | 65.7 ±0.18 / 60.9 ±0.69 | 62.5 ±0.12 / 57.3 ±0.53 |
| w/+CDMAD | 74.7 ±0.64 / 73.6 ±1.23 | 69.9 ±0.65 / 68.2 ±0.74 | 66.2 ±0.27 / 63.2 ±0.44 |
| w/+DYTRIM | **76.9** ±0.45 / **75.9** ±0.52 | **72.3** ±0.12 / **71.4** ±0.57 | **69.4** ±0.35 / **67.5** ±0.63 |

a remaining sample with score $\mathcal{H}_t^u(u_b^m) < \bar{\mathcal{H}}_t^m$, whose corresponding pruning probability is $r$, its gradient is scaled to $1/(1-r)$ times of the original, otherwise the gradient remains unchanged. The score $\mathcal{H}_{t+1}^u(u_b^m)$ is derived from the consistency regularization loss values $\mathcal{L}_{con}(\alpha(u_b^m), \mathcal{A}(u_b^m))$ for unlabeled data points. To enhance pseudo-label reliability, we further apply a confidence threshold $\tau$, where only samples with $p^*(u_b^m) > \tau$ contribute to $\mathcal{L}_{con}$, where $\mathcal{L}_{con} = \frac{1}{B} \sum_{b=1}^{B} \mathbb{I}(p^*(u_b^m) > \tau) \mathbf{H}(P_\theta(y|\mathcal{A}(u_b^m), \hat{q}_b)$. Thus, we formulate the update of $\mathcal{H}_{t+1}^u(u_b^m)$ as:

$$\mathcal{H}_{t+1}^u(u_b^m) = \begin{cases} \mathcal{H}_t^u(u_b^m) & u_b^m \in \mathcal{U}n\mathcal{S}^u, \\ \mathcal{L}_{con}(u_b^m) & u_b^m \in \mathcal{S}^u. \end{cases} \quad (21)$$

**Initialization:** at $t = 0$, scores $\mathcal{H}_t^u$ and $\mathcal{H}_t^l$ are all set to $\{1\}$, as no prior loss is available.

## 5 EXPERIMENT

In this section, we conducted comprehensive experiments to verify the effectiveness of the proposed DyTrim on CIFAR10-LT, CIFAR100-LT (Cui et al., 2019), STL10-LT (Kim et al., 2020), and ImageNet-127 (Deng et al., 2009; Huh et al., 2016) datasets. Due to limited space, we defer the detailed experimental settings to the Appendix D.

### 5.1 BASELINES

The classification performance of the DyTrim was compared with those of the following algorithms: 1. vanilla algorithm - Deep CNN trained with cross-entropy loss, 2. CIL algorithms - Resampling (JAPKOWICZ, 2000), LDAM-DRW (Cao et al., 2019), and cRT (Kang et al., 2020), 3. SSL algorithms - FixMatch (Sohn et al., 2020), and 4. CISSL algorithms - DARP, DARP+LA, DARP+cRT (Kim et al., 2020), CReST, CReST+LA (Wei & Gan, 2023), ABC (Lee et al., 2021), CoSSL (Fan et al., 2022), DASO (Oh et al., 2022), SAW, SAW+LA and SAW+cRT (Lai et al., 2022)

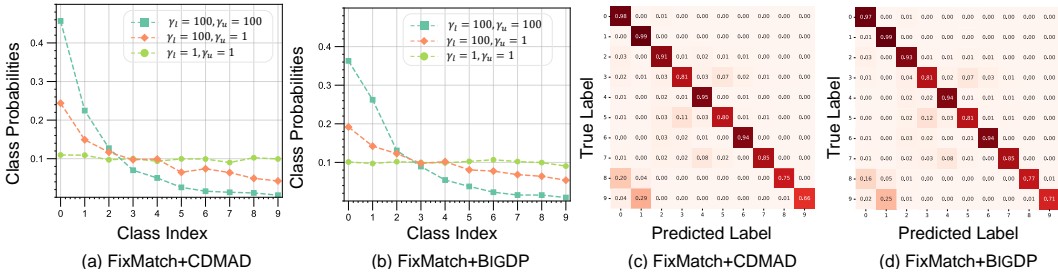

(a) FixMatch+CDMAD    (b) FixMatch+BIGDP    (c) FixMatch+CDMAD    (d) FixMatch+BIGDP

Figure 3: (a) and (b) present the $\pi_\theta(\mathcal{I})$ using the CDMAD and DyTrim. (c) and (d) present the confusion matrices of the class predictions on test samples on CIFAR-10-LT ($\gamma_l = \gamma_u = 100$). combined with FixMatch. Adsh(Guo & Li, 2022), DebiasPL (Wang et al., 2022), UDAL(Lazarow et al., 2023) and L2AC (Wang et al., 2023a) combined with FixMatch. We report the performance of the baseline algorithms reported in Tables of Lai et al. (2022) and Fan et al. (Fan et al., 2022) when it is reproducible; the performance measured using the uploaded code was reported otherwise.

## 5.2 RESULTS ON CIFAR10/100-LT AND STL-LT

Under the consistent condition where $\gamma_u$ is known and matched to $\gamma_l$, the results in Table 1 show that CISSL algorithms consistently outperform their vanilla SSL counterparts by mitigating class imbalance while effectively exploiting unlabeled data. Among them, the proposed DyTrim achieves the best performance across all imbalance ratios. Compared with the state-of-the-art CDMAD, DyTrim improves bACC by 1.2% and GM by 1.4% on average, without incurring additional computational overhead. Furthermore, when integrated into FlexMatch and FreeMatch, DyTrim yields substantial improvements, boosting bACC/GM by 2–3% on average.

Table 2 further evaluates the methods on CIFAR-100-LT, which involves more classes and stronger imbal-

Table 2: Comparison of bACC on CIFAR-100-LT.

| Algorithm | CIFAR-100-LT ($\gamma = \gamma_l = \gamma_u$, $\gamma_u$ is assumed to be known) | | |
|---|---|---|---|
| | $\gamma = 20$ | $\gamma = 50$ | $\gamma = 100$ |
| FixMatch | 49.6 ±0.78 | 42.1 ±0.33 | 37.6 ±0.48 |
| w/+DARP | 50.8 ±0.77 | 43.1 ±0.54 | 38.3 ±0.47 |
| w/+DARP+cRT | 51.4 ±0.68 | 44.9 ±0.54 | 40.4 ±0.78 |
| w/+CReST | 51.8 ±0.12 | 44.9 ±0.50 | 40.1 ±0.65 |
| w/+CReST+LA | 52.9 ±0.07 | 47.3 ±0.17 | 42.7 ±0.70 |
| w/+ABC | 53.3 ±0.79 | 46.7 ±0.26 | 41.2 ±0.06 |
| w/+CoSSL | 53.9 ±0.78 | 47.6 ±0.26 | 43.0 ±0.61 |
| w/+UDAL | 54.1 ±0.23 | 48.0 ±0.56 | 43.7 ±0.41 |
| w/+CPE | 52.4 ±0.17 | 45.6 ±0.68 | 39.9 ±0.40 |
| w/+CDMAD | 54.3 ±0.44 | 48.8 ±0.75 | 44.1 ±0.29 |
| w/+DYTRIM | **55.5** ±0.53 | **50.8** ±0.80 | **44.8** ±0.27 |
| FlexMatch | 36.5 ±0.51 | 29.6 ±0.35 | 25.8 ±0.79 |
| w/+CDMAD | 39.2 ±0.47 | 31.9 ±0.46 | 27.0 ±0.66 |
| w/+DYTRIM | **40.9** ±0.09 | **33.5** ±0.21 | **29.8** ±0.67 |
| FreeMatch | 35.9 ±0.69 | 31.3 ±0.65 | 24.5 ±0.66 |
| w/+CDMAD | 36.9 ±0.96 | 32.8 ±0.93 | 28.0 ±0.68 |
| w/+DYTRIM | **39.0** ±0.61 | **33.4** ±0.70 | **29.8** ±0.09 |

ance. The results demonstrate that DyTrim consistently outperforms all competing approaches under this more challenging setting. In particular, DyTrim delivers clear gains over CDMAD across all imbalance levels, confirming its scalability to large-scale, fine-grained datasets. Similar to the observations on CIFAR-10-LT, the integration of DyTrim continues to provide consistent improvements across different SSL frameworks.

Under the inconsistent condition where $\gamma_u$ was unknown and mismatched to $\gamma_l$, the results in Table 3 show that DyTrim remains the most effective method overall. When the labeled and unlabeled data distributions deviate, DyTrim consistently outperforms CDMAD on both CIFAR-10-LT and STL-10-LT. The benefits are particularly notable when combined with FlexMatch and FreeMatch: on STL-10-LT, FlexMatch + DyTrim improves bACC by more than 2%, while FreeMatch + DyTrim achieves nearly 2% gains.

In addition, Table 4 highlights the performance of various algorithms under both consistent and inconsistent imbalance settings with ViT backbones. On CIFAR-10-LT, DyTrim yields the best results, improving bACC 0.6% over CDMAD and nearly 4% over FixMatch when $\gamma_l = \gamma_u = 100$. Under the inconsistent condition, DyTrim maintains a clear margin, surpassing CDMAD almost 2%. On CIFAR-100-LT, although the absolute accuracies are lower due to the increased difficulty, DyTrim still matches or slightly improves upon CDMAD, while consistently outperforming FixMatch. Additional experimental results are provided in Appendix E.

## 5.3 SCALABILITY EVALUATION OF DYTRIM

DyTrim exhibited robust extensibility as a universal plug-in component, consistently boosting performance across diverse SSL frameworks (CDMAD/CCL), datasets (CIFAR/STL10-LT), and im-

Table 3: Comparison of bACC/GM on CIFAR-10-LT and STL-10-LT ($\gamma_l \neq \gamma_u$, $\gamma_u$ is assumed to be unknown).

| Algorithm | CIFAR-10-LT ($\gamma_l = 100$, $\gamma_u$ = Unknown) | | STL-10-LT ($\gamma_u$ =Unknown) | |
|---|---|---|---|---|
| | $\gamma_u = 50$ | $\gamma_u = 150$ | $\gamma_l = 10$ | $\gamma_l = 20$ |
| FixMatch | 73.9 ±0.25 / 70.5 ±0.52 | 69.6 ±0.60 / 62.6 ±1.11 | 72.9 ±0.09 / 69.6 ±0.01 | 63.4 ±0.21 / 52.6 ±0.09 |
| w/+DARP | 77.3 ±0.17 / 75.5 ±0.21 | 72.9 ±0.24 / 69.5 ±0.18 | 77.8 ±0.33 / 76.5 ±0.40 | 69.9 ±1.77 / 65.4 ±3.07 |
| w/+DARP+LA | 82.3 ±0.32 / 81.5 ±0.29 | 78.9 ±0.23 / 77.7 ±0.06 | 78.6 ±0.30 / 77.4 ±0.40 | 71.9 ±0.49 / 68.7 ±0.51 |
| w/+DARP+cRT | 82.7 ±0.21 / 82.3 ±0.25 | 80.7 ±0.44 / 80.2 ±0.61 | 79.3 ±0.23 / 78.7 ±0.21 | 74.1 ±0.61 / 73.1 ±1.21 |
| w/+ABC | 82.7 ±0.64 / 82.0 ±0.76 | 78.4 ±0.87 / 77.2 ±1.07 | 79.1 ±0.46 / 78.1 ±0.57 | 73.8 ±0.15 / 72.1 ±0.15 |
| w/+SAW | 79.8 ±0.25 / 79.1 ±0.32 | 74.5 ±0.97 / 72.5 ±1.37 | 78.3 ±0.25 / 77.0 ±0.19 | 71.9 ±0.81 / 69.0 ±0.81 |
| w/+SAW+LA | 82.9 ±0.38 / 82.6 ±0.38 | 79.1 ±0.81 / 78.6 ±0.91 | 79.4 ±0.26 / 78.4 ±0.17 | 73.9 ±0.91 / 71.8 ±0.99 |
| w/+SAW+cRT | 81.6 ±0.38 / 81.3 ±0.32 | 77.6 ±0.40 / 77.1 ±0.41 | 78.9 ±0.22 / 77.8 ±0.14 | 72.3 ±0.86 / 69.5 ±0.83 |
| w/+CPE | 86.2 ±0.26 / 85.9 ±0.33 | 82.4 ±0.49 / 82.1 ±0.53 | 79.0 ±0.05 / 78.7 ±0.54 | 77.0 ±0.73 / 76.1 ±0.68 |
| w/+CDMAD | 85.7 ±0.36 / 85.3 ±0.38 | 82.3 ±0.23 / 81.8 ±0.29 | 79.9 ±0.23 / 78.9 ±0.38 | 75.2 ±0.40 / 73.5 ±0.31 |
| w/+DyTrim | **86.4** ±0.43 / **86.0** ±0.43 | **83.8** ±0.34 / **83.4** ±0.33 | **80.7** ±0.64 / **79.8** ±0.70 | **77.9** ±1.04 / **76.7** ±1.26 |
| FlexMatch | 67.7 ±0.67 / 62.8 ±0.65 | 63.0 ±0.77 / 56.3 ±1.70 | 62.1 ±0.29 / 60.8 ±0.43 | 56.9 ±0.90 / 51.4 ±0.81 |
| w/+CDMAD | 69.2 ±0.22 / 67.0 ±0.11 | 67.0 ±1.69 / 63.4 ±0.91 | 65.5 ±1.05 / 63.7 ±1.02 | 62.4 ±1.05 / 60.5 ±0.99 |
| w/+DyTrim | **72.5** ±0.39 / **70.7** ±0.45 | **70.3** ±1.01 / **67.4** ±0.21 | **68.0** ±0.94 / **66.4** ±0.85 | **63.9** ±0.16 / **61.7** ±0.28 |
| FreeMatch | 69.3 ±0.99 / 65.4 ±1.45 | 63.5 ±0.76 / 55.7 ±0.77 | 63.9 ±0.77 / 62.0 ±0.90 | 59.0 ±1.43 / 57.6 ±0.67 |
| w/+CDMAD | 71.0 ±0.98 / 69.0 ±1.05 | 67.1 ±0.96 / 64.3 ±0.99 | 66.1 ±0.32 / 63.8 ±0.97 | 61.5 ±0.47 / 59.5 ±0.63 |
| w/+DyTrim | **72.3** ±0.69 / **71.1** ±1.23 | **69.9** ±0.15 / **67.4** ±0.37 | **68.0** ±0.64 / **66.5** ±1.20 | **64.6** ±0.77 / **62.7** ±1.16 |

Table 4: Comparison of bACC/GM on CIFAR-10-LT and CIFAR-100-LT with ViT.

| Algorithm | CIFAR-10-LT | | CIFAR-100-LT |
|---|---|---|---|
| | $\gamma_l = \gamma_u = 100$ | $\gamma_l = 100, \gamma_u = 150$ | $\gamma_l = \gamma_u = 100$ |
| FixMatch | 45.5 ±0.14 / 30.0 ±0.41 | 45.3 ±0.12 / 28.9 ±0.96 | 23.2 ±0.13 / 5.7 ±0.33 |
| w/+CDMAD | 48.7 ±0.49 / **40.5** ±0.26 | 45.4 ±0.13 / **39.9** ±0.10 | 24.0 ±0.15 / **9.0** ±0.77 |
| w/+DyTrim | **49.3** ±0.47 / 40.3 ±0.36 | **47.3** ±0.12 / 39.7 ±0.57 | **24.1** ±0.22 / 8.9 ±0.15 |

Table 5: Comparison of bACC with two state-of-the-art CISSL algorithms with and without DyTrim on CIFAR-10, CIFAR-100, and STL-10.

| **Dataset** | | FixMatch+ | | | FixMatch+ | | |
|---|---|---|---|---|---|---|---|
| | | CDMAD | CDMAD+DyTrim | Gain | CCL | CCL+DyTrim | Gain |
| CIFAR10-LT | $\gamma_l = \gamma_u = 100$ | 83.6 ±0.46 | **84.8** ±0.48 | ↑1.2 | 86.2 ±0.35 | **86.7** ±0.39 | ↑0.5 |
| | $\gamma_l = \gamma_u = 150$ | 80.8 ±0.86 | **82.0** ±0.09 | ↑1.2 | 84.0 ±0.21 | **84.0** ±0.26 | ↑0.0 |
| | $\gamma_l = 100, \gamma_u = 1$ | 87.5 ±0.46 | **88.9** ±0.88 | ↑1.4 | 93.9 ±0.12 | **94.1** ±0.17 | ↑0.2 |
| CIFAR100-LT | $\gamma_l = \gamma_u = 20$ | 54.3 ±0.44 | **55.5** ±0.53 | ↑1.2 | 57.5 ±0.16 | **58.1** ±0.49 | ↑0.6 |
| STL10-LT | $\gamma_l = 10$ | 79.9 ±0.23 | **80.7** ±0.64 | ↑0.8 | 84.8 ±0.15 | **85.1** ±0.33 | ↑0.3 |
| | $\gamma_l = 20$ | 75.2 ±0.40 | **77.9** ±1.04 | ↑2.7 | 83.1 ±0.18 | **83.3** ±0.40 | ↑0.2 |

balance ratios ($\gamma = 1 \sim 150$). Notably, it achieved up to +1.4% (CDMAD on CIFAR10-LT) and +2.7% (STL10-LT, $\gamma_l$=20) gains without architecture-specific tuning, validating its versatility in semi-supervised long-tailed scenarios.

# 6 CONCLUSION

In this work, we introduce DyTrim, a simple yet effective method to address long-tailed semi-supervised learning (LTSSL) by leveraging baseline image-guided dynamic data pruning for de-biasing. We provide a theoretical framework that explains the role of baseline images in reducing classifier bias under class imbalance, showing how the class imbalance datasets influence the model prediction of the baseline image. Our method, DyTrim, offers an elegant solution to the problem of class imbalance in semi-supervised learning without requiring complex network branches or de-biasing mechanisms. Through extensive experiments, we demonstrate that DyTrim significantly improves the performance of LTSSL models on public benchmarks. We believe that our approach can serve as a powerful tool for improving LTSSL in real-world datasets where class imbalance remains a major challenge.

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

## F   PROOF FOR SECTION 3

In this section, we present the technical details of Section 3. In particular, Section F.1 first discuss the relationship between classifier $f_\theta$, dataset $\{\mathcal{X}; \mathcal{U}\}$ and the baseline image $\mathcal{I}$. Then, Section **??** proves Theorem **??** for revealing the baseline image's intrinsic debiasing effect, and Section F.2 presents the details of the bias term and running statistics.

### F.1   PROOF OF PROPOSITION 3

**Proposition 1.** *Let $\pi = \texttt{Softmax}(z)$ and $z = g_\theta(x)$. The one-step dynamics decompose as*

$$\Delta \log \pi^t(y \mid \mathcal{I}) = -\eta \mathcal{T}^t(\mathcal{I}) \mathcal{K}^t(\mathcal{I}, x) \mathcal{G}^t(x, y) + \mathcal{O}(\eta^2 \|\nabla_\theta z(x)\|^2_{\text{op}}), \tag{32}$$

*where $\mathcal{T}^t(\mathcal{I}) = \nabla_z \log_{\pi^t}(\mathcal{I}) = I - \mathbf{1}\pi^T_{\theta^t}(\mathcal{I})$, $\mathcal{K}^t(\mathcal{I}, x) = (\nabla_\theta z(\mathcal{I})|_{\theta^t})(\nabla_\theta z(x)|_{\theta^t})^T$ is the empirical neural tangent kernel of the logit network $z$, and $\mathcal{G}^t(x, y) = \nabla_z \mathcal{L}(x, y)|_{z^t}$.*

*Proof.* Inspired by the analysis of the learning dynamic of (Ren et al., 2022; Ren & Sutherland, 2024). In this work, we want to observe the classifier's prediction on the baseline image $\mathcal{I}$. Starting from Eq (12), we first approximate $\log \pi^{t+1}(y \mid \mathcal{I})$ using first-order Talyor expansion, with slightly abused symbols, we use $\pi^t$ to represent $\pi_{\theta^{t+1}}$, $x$ to represent labeled sample $x_b^n$ and $u$ to represent unlabeled sample $u_b^m$:

$$\log \pi^{t+1}(y|\mathcal{I}) = \log \pi^t(y|\mathcal{I}) + <\nabla \log \pi^t(y|\mathcal{I}), \theta^{t+1} - \theta^t> + \mathcal{O}(\|\theta^{t+1} - \theta^t\|^2)$$

Then, assuming the model updates its parameters using SGD calculated by an "updating labeled example" $(x, y)$ or an "updating unlabeled example" $u$, we can rearrange the terms in the above equation to get the following expression:

$$\Delta \log \pi^t(y|\mathcal{I}) = \log \pi^{t+1}(y|\mathcal{I}) - \log \pi^{t+1}(y|\mathcal{I}) = \nabla_\theta \log \pi^t(y|\mathcal{I})|_{\theta^t}(\theta^{t+1} - \theta^t) + \mathcal{O}(\|\theta^{t+1} - \theta^t\|^2),$$

To evaluate the leading term, we first take a labeled sample as an example plug in the definition of SGD, and repeatedly use the chain rule:

$$\begin{aligned}
\nabla_\theta \log \pi^t(y|\mathcal{I})|_{\theta^t}(\theta^{t+1} - \theta^t) &= (\nabla_z \log \pi^t(y|\mathcal{I})|_{z^t})(-\eta \nabla_\theta \mathcal{L}(x)|_{\theta^t})^T \\
&= (\nabla_z \log \pi^t(y|\mathcal{I})|_{z^t})(-\eta \nabla_\theta \mathcal{L}(x)|_{z^t} - \nabla_\theta z^t(x)|_{\theta^t})^T \\
&= -\eta \nabla_z \log \pi^t(\mathcal{I})|_{z_t} [\nabla_\theta z(\mathcal{I})|_{\theta^t}(\nabla_\theta z(x)|_{\theta^t})^T](\nabla_z \mathcal{L}(x)|_{z^t})^T \\
&= -\eta \mathcal{T}^t(\mathcal{I})\mathcal{K}^t(\mathcal{I}, x)\mathcal{G}^t(x, y)
\end{aligned} \quad (33)$$

For the higher-order term, using as above that

$$\theta^{t+1} - \theta^t = -\eta \nabla_\theta z^t(x)|_{\theta_t}^T \mathcal{G}^t(x, \hat{y})$$

and noting that, since the residual term $\mathcal{G}^t$ is usually bouned, we have that

$$\mathcal{O}(\|\theta^{t+1} - \theta^t\|^2) = \mathcal{O}(\eta^2 \|(\nabla_\theta z^t(x)|\theta^t)^T\|_{op}^2 \|\mathcal{G}^t(x, \hat{y})\|_{op}^2) = \mathcal{O}(\eta^2 \|\nabla_\theta z(x)\|_{op}^2)$$

$\square$

In the decomposition, we can write our $\mathcal{T}^{(t)}$ as $\mathcal{T}^t(\mathcal{I}) = \nabla_z \log_{\pi^t}(\mathcal{I}) = I - \mathbf{1}\pi_{\theta^t}^T(\mathcal{I})$, this term is only related to the input $\mathcal{I}$, and reflects the model's correspondence to the baseline image, we will further analysis $\log_{\pi^t}(\mathcal{I})$. The second term in this decomposition, $\mathcal{K}^t(\mathcal{I}, x)$ is the product of gradients at $\mathcal{I}$ and $x$ or $u$. As shown in (Ren & Sutherland, 2024), if their gradients have similar directions, the Frobenius norm of this matrix is large, and vice versa. This matrix is known as the empirical neural tangent kernel, and it can change through the course of training as the network's notion of "similarity" evolves. The third term in this decomposition, $\mathcal{G}^t$ is determined by the loss function $\mathcal{L}$, which provides the energy and direction for the model's adaptation. We have $\mathcal{L} = \mathcal{L}_{sup}(x_b^n, y_b^n)$ for each labeled sample and $\mathcal{L} = \mathcal{L}_{con}(\alpha(u_b^m), \mathcal{A}(u_b^m))$ for each unlabeled sample. According to the analysis of Xing et al. (2025), $\mathcal{G}^t$ using the baseline image enhances the balance of the base SSL model implicitly utilizing the integrated gradient flow $\nabla_\theta \mathcal{L}_{\text{Con}} = \sum_b \left( \sum_{i=1}^d \text{IntegratedGrads}_i(u_b^m) \right) \nabla g_b + \sum_b q_{\mathcal{A},b} \frac{\partial q_{\mathcal{A},b}}{\partial \theta}$.

*Proof.* For the baseline image $\mathcal{I}$ is a solid-balck image, *i.e.*, the $k = 0$, with $h(\mathcal{I}) = \beta$

$$g_\theta^*(x_b) = h(x_b) - h(\mathcal{I}) = \frac{\langle \mathbf{w}, x_b \rangle - \mathbb{E}[\langle \mathbf{w}, x_b \rangle]}{\sqrt{\text{Var}[\langle \mathbf{w}, x_b \rangle]}} * \gamma + \beta - \beta = \frac{\langle \mathbf{w}, x_b \rangle - \mathbb{E}[\langle \mathbf{w}, x_b \rangle]}{\sqrt{\text{Var}[\langle \mathbf{w}, x_b \rangle]}} * \gamma$$

$\square$

## F.2 DETAIL OF THE BIAS TERM AND RUNNING STATISTICS

**Effects of bias term.** When the bias term $\beta$ of the BN layer is frozen and equal to 0, $h(\mathcal{I})$ becomes $\gamma * (\langle \mathbf{w}, k \rangle - \mathbb{E}[\langle \mathbf{w}, k \rangle])/\sqrt{\text{Var}[\langle \mathbf{w}, k \rangle]}$ which is the same as the Eq.(7) except for a bias term. Ignoring the running statistics strategy, the form of $h(\mathcal{I})$ only depends on the $\beta$. As a result, $h(\mathcal{I})$ becomes $h(\mathcal{I}) \to 0$ during training and $h(\mathcal{I}) \to -\gamma * \mathbb{E}_{mom}[\langle \mathbf{w}, x_b \rangle]/\sqrt{\text{Var}_{mom}[\langle \mathbf{w}, x_b \rangle]}$ during testing. This shows that the $g_\theta^*$ operation has no effect in the training phase and only eliminates the impact of the unbalanced running means in the testing phase. This will affect the ability to benefit $h$ from $g_\theta^*$, as shown in Table. 13.

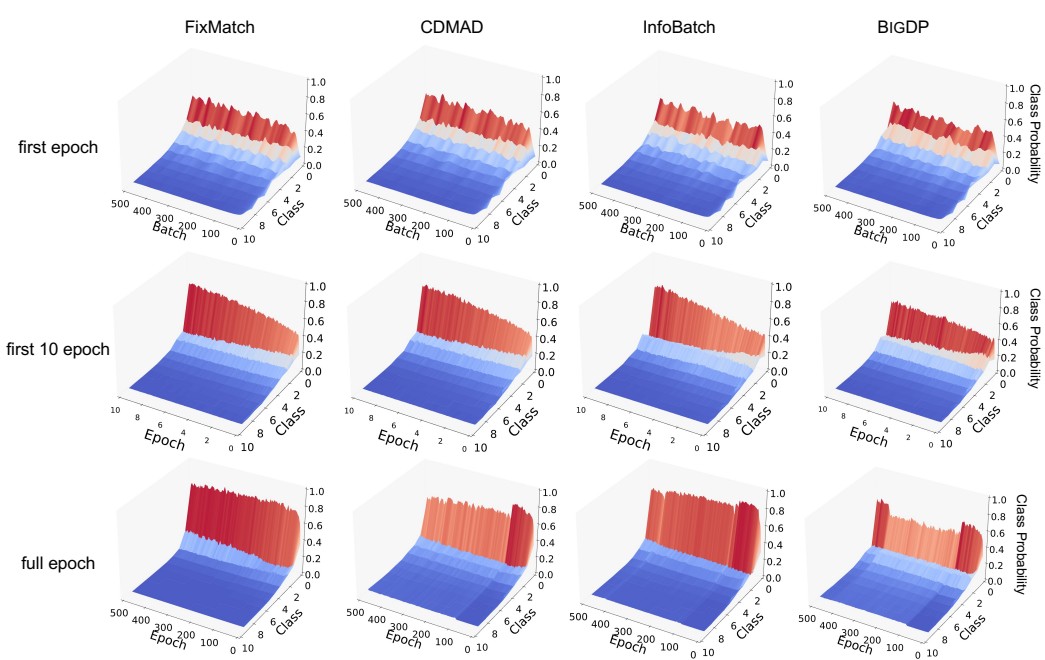

Figure 6: Comparison of the change of logits's probability distribution $\pi_\theta(\mathcal{I})$ for the baseline image on CIFAR-10-LT with $\gamma_l = \gamma_u = 100$ across different CISSL methods.

**Effects of running statistics.** When we do not keep running estimates, batch statistics are instead used during evaluation time as well. The form of $h(\mathcal{I})$ becomes $h(\mathcal{I}) \to \beta$ both training and testing. We can rewrite $g_\theta^*(x_t) = \gamma * (\langle \mathbf{w}, x_t \rangle - \mathbb{E}[\langle \mathbf{w}, x_t \rangle]) / \sqrt{\text{Var}[\langle \mathbf{w}, x_t \rangle]}$. On the other hand, as $h(\mathcal{I}) \to 0$, the benefit of $g_\theta^*$ is also vanishes, also shown in Table. 13.

We then extend our results to a non-linear neural network, thus we have the following corollary:

Table 13: Comparison of bACC/GM on CIFAR-10-LT.

| Metric | With original $g_\theta^*$ | $g_\theta^*$ without $\beta$ | $g_\theta^*$ without $\mathbf{x}_{mom}$ | $g_\theta^*$ without $\beta$ & $\mathbf{x}_{mom}$ |
|--------|---------------------------|------------------------------|------------------------------------------|---------------------------------------------------|
| bACC | $83.6 \pm 0.46$ | $80.92 \pm 0.02 \downarrow 2.68$ | $71.63 \pm 0.35 \downarrow 11.97$ | $64.01 \pm 0.14 \downarrow 19.59$ |
| GM | $83.1 \pm 0.57$ | $80.37 \pm 0.23 \downarrow 2.73$ | $67.85 \pm 0.51 \downarrow 15.25$ | $54.48 \pm 0.36 \downarrow 28.62$ |

## G VISUALIZATION

### G.1 DETAILS OF THE CHANGE OF LOGITS'S PROBABILITY DISTRIBUTION

In this section, we conduct some visualization experiments to demonstrate the advantages of the DyTrim in debiasing and improving classifier performance. We first analyze the change of logits's probability distribution $\text{Softmax}(g_\theta(\mathcal{I}))$ for the baseline image on CIFAR-10-LT with $\gamma_l = \gamma_u = 100$ for fixmatch, CDMAD, and the DyTrim as shown in Figure. 6. It can be seen intuitively that in the first epoch, the classifier has bias due to the imbalance of categories in the data. This situation increases significantly with the number of network training times, as shown in the second column of the figure. However, we can see that DyTrim can effectively slow down the increase of this bias. Furthermore, after the model is fully trained for 500 epochs, it can be seen that after the 100th epoch, CDMAD starts to use the baseline image for post-hoc debiasing, which significantly reduces the representation of the model. However, by dynamically pruning the data set, DyTrim obtains a more distinct debias effect as shown in Figure. 7.

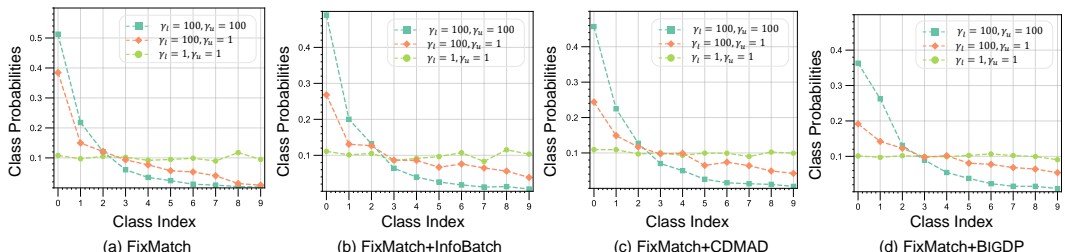

Figure 7: Class probabilities predicted on a baseline image using (a) FixMatch, (b) Fix-Match+InfoBatch, (c) FixMatch+CDMAD, (d) FixMatch+DyTrim.

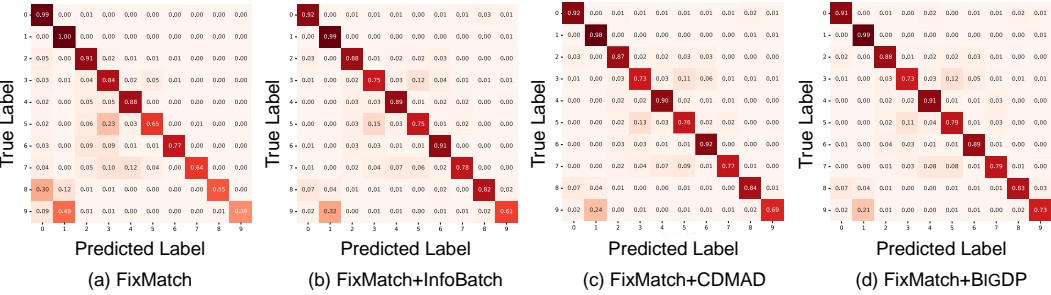

Figure 8: Confusion matrices of the class predictions on the test set of CIFAR-10 using (a) FixMatch, (b) FixMatch+InfoBatch, (c) FixMatch+CDMAD, and (d) FixMatch+DyTrim trained on CIFAR-10-LT under $\gamma_l = 100$ and $\gamma_u = 100$.

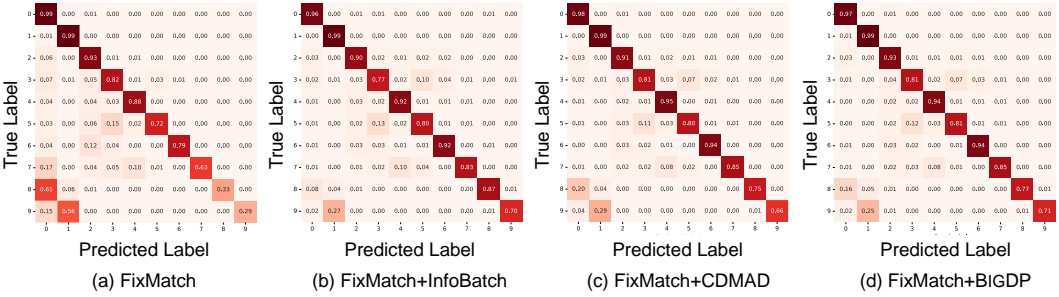

Figure 9: Confusion matrices of the class predictions on the test set of CIFAR-10 using (a) FixMatch, (b) FixMatch+InfoBatch, (c) FixMatch+CDMAD, and (d) FixMatch+DyTrim trained on CIFAR-10-LT under $\gamma_l = 100$ and $\gamma_u = 1$.

## G.2 DETAILS OF THE CHANGE OF LOGITS'S PROBABILITY DISTRIBUTION

Figure. 8 and Figure. 9 compare the confusion matrices of the class predictions on the test set of CIFAR-10 using (a) FixMatch, (b) FixMatch+Infobatch, (c) FixMatch+CDMAD, and (d) Fix-Match+DyTrim trained on CIFAR-10-LT under $\gamma_l = 100, \gamma_u = 1, 100$. FixMatch+DyTrim made more balanced predictions across classes. Furthermore, we also conducted experiments under a balanced setting ($\gamma = \gamma_1 = \gamma_u = 1$), as shown in Figure. 10. The results show that even under a balanced data distribution, BigDP can still achieve better results on the pruned dataset than methods such as CDMAD trained on the full dataset.

Similar to confusion matrices, we also compare t-distributed stochastic neighbor embedding (t-SNE) of representations obtained for the test set of CIFAR-10 using FixMatch, FixMatch+CDMAD, FixMatch+InfoBatch, and FixMatch+DyTrim trained on CIFAR-10 with $\gamma_l = 100$ and $\gamma_u = 1, 100(\textbf{unknown } \gamma_u)$, where different colors indicate different classes in CIFAR-10 Figure. 11, Figure. 12. We can observe that the representations obtained using FixMatch+DyTrim are separated into classes with clearer boundaries compared the those from FixMatch and CDMAD. This is prob-

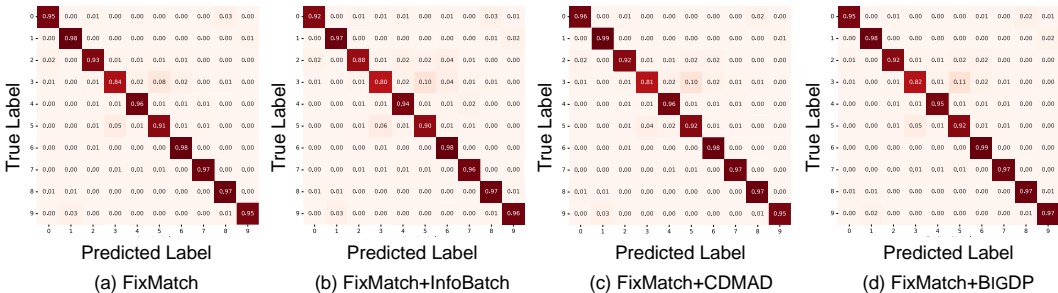

Figure 10: Confusion matrices of the class predictions on the test set of CIFAR-10 using (a) Fix-Match, (b) FixMatch+InfoBatch, (c) FixMatch+CDMAD, and (d) FixMatch+DyTrim trained on CIFAR-10-LT under $\gamma_l = 1$ and $\gamma_u = 1$.

ably because CDMAD appropriately refined the biased pseudo-labels and used them for training, whereas FixMatch failed to learn the representations properly because they used the biased pseudo-labels for training. These results demonstrate that the quality of representations can be improved by using well-refined pseudo-labels for training.

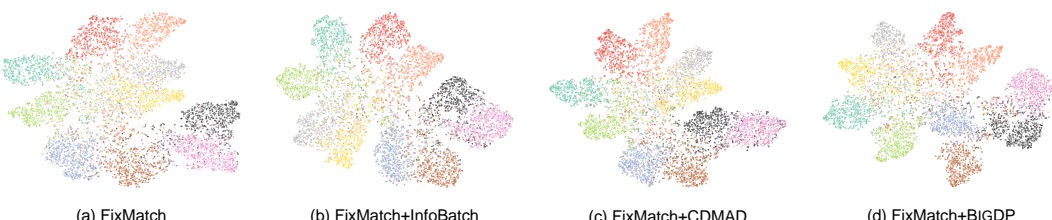

| (a) FixMatch | (b) FixMatch+InfoBatch | (c) FixMatch+CDMAD | (d) FixMatch+BIGDP |

Figure 11: t-SNE of representations obtained for the test set of CIFAR-10 using (a) FixMatch, (b) FixMatch+InfoBatch, (c) FixMatch+CDMAD, and (d) FixMatch+DyTrim trained on CIFAR-10-LT under $\gamma_l = 100$ and $\gamma_u = 100$.

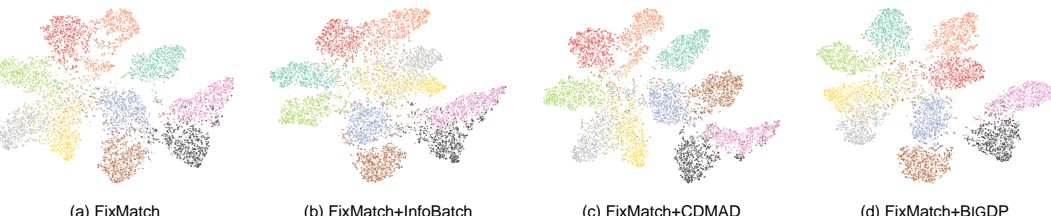

| (a) FixMatch | (b) FixMatch+InfoBatch | (c) FixMatch+CDMAD | (d) FixMatch+BIGDP |

Figure 12: t-SNE of representations obtained for the test set of CIFAR-10 using (a) FixMatch, (b) FixMatch+InfoBatch, (c) FixMatch+CDMAD, and (d) FixMatch+DyTrim trained on CIFAR-10-LT under $\gamma_l = 100$ and $\gamma_u = 1$.

## H    USE OF LLMS

This work did not involve the use of large language models (LLMs) at any stage. The design of experiments, data analysis, and manuscript preparation were conducted entirely by the authors through conventional computational methods and human expertise, without reliance on automated text generation or model-driven reasoning.