# OpenReview forum: "Learning Dynamics of Logits Debiasing for Long-Tailed Semi-Supervised Learning"
_ICLR.cc/2026/Conference — ICLR 2026 Poster_

### Official Review · Reviewer_jDsT · 2025-10-26

**Soundness:** 3
**Presentation:** 3
**Contribution:** 2
**Rating:** 4
**Confidence:** 4

**Summary:**

This paper proposes the DyTrim method, which eliminates bias by employing benchmark image-guided dynamic data pruning. The authors demonstrate that this approach implicitly implements risk reweighting, elucidate the mechanism by which benchmark images mitigate classifier bias in class-imbalanced scenarios, and reveal how class-imbalanced datasets influence a model's predictions on baseline images.

**Strengths:**

This paper demonstrates through a solid theoretical foundation that the DyTrim method can effectively suppress class bias. The DyTrim method provides an elegant solution to the class imbalance problem in semi-supervised learning without requiring complex network branches or de-biasing mechanisms. Extensive experiments on public benchmarks validate the effectiveness of the proposed method. This approach contributes to the advancement of the field.

**Weaknesses:**

[1] Conventional pruning strategies are typically employed to eliminate samples that contribute minimally to training, thereby accelerating the training process. However, this paper utilizes such strategies to obtain a relatively balanced training subset. Would this approach suffer from the undersampling problem, specifically for tail classes?

[2] Figure 1 fails to provide sufficient evidence for the conclusions put forward in the paper. Especially in the bottom part, the predicted distributions derived from pseudo-labels—whether correct or incorrect—show no significant difference; in some cases, they are even completely identical.

[3] In Equation 18, what is the distinction between H_t^l  and the subsequent H_(c,t)^l? Additionally, how are the scoring functions H_t^l and H_t^u specifically obtained?

[4] In Eq (20), for unlabeled data, high pseudo-label confidence links to a random pruning probability r , where a larger r results in a larger gradient scaling factor. Could this intensify the model’s bias toward high-confidence samples (usually head classes)? Furthermore, the gradient scaling follows Qin et al.’s setting. Ablation studies demonstrate that it contributes more to model performance than other components proposed in this work, and removing it causes a notable decline in performance.

[5] According to Equations (19) and (20), the scores corresponding to pruned samples will no longer be modified. Does this mean that these samples will not be used in subsequent training processes? If so, it would result in severe information waste.

[6] The experimental section lacks comparisons with the latest literature. Additionally, it would be beneficial if the authors could provide information on the pruning amount of training samples in each iteration.

[7] It seems that the predicted class probabilities obtained from Figure 3(a) and (b) both appear to be unsatisfactory. Additionally, some errors appear to exist in the titles and their corresponding descriptions for Figure 3, Figure 4, and Figures 6 to 10.

[8] There are still some expression errors and unclear descriptions in the manuscript. For instance, the definition of the imbalance of the unlabeled dataset and the description of Figure 2 require clarification.

**Questions:**

As described in “Weaknesses”.

---

> ### Author Response · Authors · 2025-11-22
> **Response, Part I/III**
>
> We thank the reviewer very much for the valuable comments. Below we address your concerns.
>
> ### **Weaknesses Part**
>
> **W1**. "*Conventional pruning ... classes?*"
>
> **Response**: We thank the reviewer for this insightful suggestion. While conventional pruning is typically used to accelerate training by discarding low-informative samples [1], DyTrim is designed with a different goal and mechanism: to obtain a dynamic, more balanced, and informative training subset without undersampling the tail classes.
>
> For labeled samples, DyTrim uses an **adaptive class-specific pruning ratio**, pruning head classes more aggressively and tail classes more mildly. Pruning is based on their **class prior** ($r_c$) and **loss function** ($\mathcal{L}_{sup}(x^n_b, y^n_b)$) removing only **redundant** or **easy** samples. Since tail classes have **fewer samples** and **higher difficulty scores**, most tail samples are preserved, as shown in **Figure 9**, **page 28**, preventing undersampling and retaining critical tail information.
>
> For unlabeled samples, Head class samples dominate training, receive lower difficulty scores ($\mathcal{L}_{con}(\alpha(u^m_b), \mathcal{A}(u^m_b))$, in **Eq.(43)**, **page 23**) and high confidence ($p^*(u^m_b) \geq \tau$, in **Eq.(22)**, **lines 397-398**, **page 8**) are pruned more often. In contrast, tail-class samples remain under-trained, resulting in higher difficulty scores and, therefore, higher retention rates, even without class priors.
>
> The table below confirms this behavior across all three datasets (labeled, unlabeled, and full). Head class 0 shows a clear reduction after pruning (e.g., ( 40.31% \rightarrow 35.80% ) in the full set), while tail classes see increased proportions. Notably, even in the unlabeled dataset—where class identities are unknown—the relative shares of tail classes consistently rise (e.g., class 2: ( 14.49% \rightarrow 16.51% ); class 7: ( 1.11% \rightarrow 1.63% )). This demonstrates that DyTrim effectively preserves tail samples by relying solely on sample contribution, allowing the method to rebalance the data distribution without labels.
>
> **Comparison of class distribution before and after pruning across Labeled dataset, Unlabeled dataset and Full dataset.**
> | Class | Dataset Type | Before Pruning (%) | After Pruning (%) | Δ Change (%) |
> |-------|--------------|---------------------|--------------------|---------------|
> | **0** | Labeled      | 40.32 | 30.73 | ↓9.59 |
> |       | Unlabeled    | 40.32 | 38.41 | ↓1.91 |
> |       | Full         | 40.31 | 35.80 | ↓4.51 |
> | **1** | Labeled      | 24.14 | 26.68 | ↑2.54 |
> |       | Unlabeled    | 24.16 | 18.29 | ↓5.87 |
> |       | Full         | 24.16 | 21.16 | ↓3.00 |
> | **2** | Labeled      | 14.49 | 16.38 | ↑1.89 |
> |       | Unlabeled    | 14.49 | 16.51 | ↑2.02 |
> |       | Full         | 14.49 | 16.46 | ↑1.97 |
> | **3** | Labeled      | 8.68  | 10.55 | ↑1.87 |
> |       | Unlabeled    | 8.68  | 10.77 | ↑2.09 |
> |       | Full         | 8.68  | 10.69 | ↑2.01 |
> | **4** | Labeled      | 5.19  | 6.51  | ↑1.32 |
> |       | Unlabeled    | 5.20  | 6.26  | ↑1.06 |
> |       | Full         | 5.19  | 6.34  | ↑1.15 |
> | **5** | Labeled      | 3.12  | 4.00  | ↑0.88 |
> |       | Unlabeled    | 3.12  | 4.38  | ↑1.26 |
> |       | Full         | 3.12  | 4.25  | ↑1.13 |
> | **6** | Labeled      | 1.85  | 2.39  | ↑0.54 |
> |       | Unlabeled    | 1.86  | 2.30  | ↑0.44 |
> |       | Full         | 1.86  | 2.33  | ↑0.47 |
> | **7** | Labeled      | 1.10  | 1.39  | ↑0.29 |
> |       | Unlabeled    | 1.11  | 1.63  | ↑0.52 |
> |       | Full         | 1.11  | 1.55  | ↑0.44 |
> | **8** | Labeled      | 0.67  | 0.86  | ↑0.19 |
> |       | Unlabeled    | 0.67  | 0.93  | ↑0.26 |
> |       | Full         | 0.67  | 0.90  | ↑0.23 |
> | **9** | Labeled      | 0.40  | 0.50  | ↑0.10 |
> |       | Unlabeled    | 0.40  | 0.50  | ↑0.10 |
> |       | Full         | 0.40  | 0.50  | ↑0.10 |
>
> **W2**. "*Figure 1 ... identical.*"
>
> **Response**: We thank the reviewer for the suggestion. We have added a more detailed caption to **Figure 1** (in **lines 154-160**, **page 3**), explaining that in the balanced setting, similarity terms dominate model updates in **Figure 1** **(a)** and **(b)**. However, in the unbalanced setting, class bias is significantly amplified, forcing the model to output predictions aligned with label priors, thereby suppressing the influence of similarity. As shown in **Figure 1 (c)** and **(d)**, this directly results in the accuracy of pseudo-labels having almost little impact on model predictions.

---

> > ### Author Response · Authors · 2025-11-22
> > **Response, Part II/III**
> >
> > ### **Weaknesses Part (continuing)**
> >
> > **W3**. "*In Equation 18 ... obtained?*"
> >
> > **Response**: We thank the reviewer for this nice suggestion. We use $\mathcal{H}_t^l(\cdot)$ ($\mathcal{H}_t^u(\cdot)$) to represent step-dependent scoring functions for labeled samples $\mathcal{X}$ (unlabeled samples $\mathcal{U}$). We use $\bar{H}_t^l$ and  $\bar{H}_t^{l}$  to represent adaptive thresholds.
> >
> > In actual pruning, we choose hard pruning for labeled samples, so we rewrite $\bar{H}_t^l$ as
> >
> > $H^l_{< r_{c}, t}$,
> >
> > where it denotes the $r_c \times N_c$ smallest scoring values of the class $c$, and $r_c = \pi_{\theta} (\mathcal{I})_c$ is the class-aware pruning probability.
> >
> >
> > **W4**. "*In Eq(20) ... performance.*"
> >
> > **Response**: We thank the reviewer for this insightful question and suggestion. We agree that setting an excessively large $r$ might lead to a larger gradient scaling factor, potentially intensifying the model's bias toward high-confidence samples. In practical applications, we conducted hyperparameter analysis and selected a relatively small $r = 0.3$, as shown in **Figure 7, page 27**. However, since the initial design goal of gradient scaling is to match the loss of training on the full data, even slightly increasing $r$ will not significantly impair DyTrim’s performance. The ablation experiments (in **Table 14, page 26**) further show that even **without processing the unlabeled data**, balancing and pruning only the labeled samples yields better results than the baseline.
> >
> > To clarify, the original use of gradient scaling was due to the inability to know the class prior of the unlabeled data. However, **inspired by your suggestion**, we implemented **an alternative version of DyTrim** during the rebuttal, which estimates a class ratio $\hat{r}^u_c$ for the unlabeled class and designs a dynamic gradient scaling strategy based on this. The results, shown in **Table 15**, demonstrate that the dynamic scaling version of DyTrim provides significant improvement over the fixed version when the distributions of unlabeled and labeled data are inconsistent.
> >
> > **Table 15. Comparison of bACC and GM on CIFAR10-LT on fixed and dynamic scaling factors**
> > | Algorithm | $\gamma_l=100, \gamma_u=100$ | $\gamma_l=100, \gamma_u=1/100$|
> > |-------|--------------|---------------------|
> > |Fix Scaling| 84.8 / 84.4 | 78.2 / 76.7|
> > |Dynamic Scaling| **84.9** / **84.4** | **78.9** / **78.1** |
> >
> > **W5**. "*According to ... waste.*"
> >
> > **Response**: We thank the reviewer for the insightful suggestion. We will discuss whether pruned samples will be used in subsequent training from two perspectives:
> >
> > 1. **Relative Change of Gradient**: Since we randomly prune high-confidence samples, as training progresses and model performance improves, the overall confidence of the remaining samples increases, and the average loss decreases. This means that even if the gradient of the pruned samples remains unchanged, the pruned samples will still have their gradients updated in subsequent rounds due to relative changes in the overall training dynamics.
> >
> > 2. **Randomness of Pruning**: For unlabeled samples with difficulty scores smaller than the threshold $\bar{H}^u_t$, not all are pruned, but rather pruned according to a random probability $r$. In this case, after the previous round of pruning, there is a $1-r$ probability that the pruned samples will be selected for gradient update again. This randomness is one of the reasons we propose dynamic pruning instead of static reweighting.
> >
> > We also plot the dynamic changes in loss of a head class sample and a tail class sample during the iteration process to illustrate that they are not wasted, as shown in **Figure 11**.

---

> ### Author Response · Authors · 2025-11-22
> **Response, Part III/III**
>
> ### **Weaknesses Part (continuing)**
>
> **W6**. "*The experimental ... iteration.*"
>
> **Response**: We thank the reviewer for the insightful suggestion. To clarify, as a byproduct of the analysis, we expect DyTrim to perform logits debias on LTSSL in the simplest possible way. Therefore, we tested its capabilities on many general SSL frameworks, such as FlexMatch and FreeMatch. Furthermore, we combined it with the state-of-the-art CCL [2] in the field to achieve improved performance.
>
> We also provide the pruning amount of training samples in each iteration, as shown in **Figure 9**. We also plotted the maximum, minimum, and average probabilities of each class being pruned during training, as shown in **Figure 9**. It can be observed that in each class, some samples are always retained, with tail classes having the highest probability of retention.
>
> We also provide the specific pruning values ​​for each class in each iteration, as shown in **Figure 12**.
>
> **W7**. "*It seems ... 10.*"
>
> **Response**: We agree that the predicted class probabilities obtained are still not ideal, and the predicted class probabilities they obtain do still exhibit class imbalance. However, our method significantly reduces this class imbalance compared to the baseline image. **Figure 3(a)** shows the predicted class probabilities obtained by the model on the baseline image after direct LA correction, while **Figure 3(b)** shows the predicted class probabilities obtained by the model on the baseline image after pruning the dataset using the proposed DyTrim method. In addition, we have revised the titles and their corresponding descriptions for Figures 3, 4, and 6 to 10.
>
> **W8**. "*There are ... clarification.*"
>
> **Response**: We have addressed this issue and made the necessary corrections in the revised version.
>
> ### **Reference**
>
> [1] Ziheng Qin et al. Infobatch: Lossless training speed up by unbiased dynamic data pruning. ICLR 2024.
>
> [2 Zi-Hao Zhou et.al. Continuous contrastive learning for long-tailed semi-supervised recognition. NeurIPS 2024.

---

> ### Comment · Reviewer_jDsT · 2025-11-22
>
> [1] “As shown in Figure 1 (c) and (d), this directly results in the accuracy of pseudo-labels having almost little impact on model predictions.” This appears counterintuitive, as in the SSL domain, the accuracy of pseudo labels plays a decisive role in model predictions. When all pseudo labels are incorrect, the nearly identical distributions shown in Figures 1(c) and (d) would be highly improbable.
>
> [2] Without the “Rescaling” module, the performance of this method is significantly lower than that of the simpler CDMAD;
>
> [3] In the “Notions” section, it is not advisable to directly define γ_u using M_1 and M_C. It is preferable to use the more standard notation max{M_i} and min{M_i}.
>
> The authors have provided satisfactory responses to other questions, and this paper is theoretically well-grounded. Therefore, I have raised the score to 6.

---

> > ### Author Response · Authors · 2025-11-22
> >
> > We thank the reviewer so much for the quick reply and we really appreciate the reviewer for raising score. We will modify the definition of $\gamma_u$ using $M_1$ and $M_C$ as per your suggestion and update it accordingly in the revision.

---

> ### Author Response · Authors · 2025-11-22
> **Response**
>
> Thank you so much for your prompt reply and interesting questions. We are clearer about your concerns and below we answer your questions.
>
> 1. We agree that the accuracy of pseudo-labels is critical to model predictions in the SSL domain. While it may seem unlikely to observe nearly identical distributions in Figures 1(c) and (d) if all pseudo labels were incorrect, the figures actually show the logits distribution on the test sample with $x_0=4$ before and after training with **a labeled sample** {$x_b$, $y_b=0$} and **an unlabeled sample** {$u_b$}. In this specific scenario, due to class imbalance, the accuracy of pseudo-labels has minimal impact on model predictions. However, as shown in Figures 1(a) and (b) with the balanced dataset, the results are consistent with the phenomenon you mentioned, where the accuracy of pseudo-labels plays a larger role.
>
> 2. Regarding your concern about the gradient and the number of pruning samples during training, we have now provided additional clarification in the revised paper. You can refer to **Figure 11** and **Figure 12** (in **Appendix H.11** and **H.12**), which answer this question in detail.
>
>     - In **Appendix H.11**, **Figure 11** illustrates the behavior of the loss functions for both a head class sample and a tail class sample during training. As demonstrated in the figure, the loss function of the head class sample remains fixed after it is pruned. However, once a relative change occurs in the training process, the loss function of the head class sample reverts to its previous state, indicating that the pruning process does not permanently affect the update dynamics for these samples. This behavior highlights how the pruning strategy interacts with the loss function, allowing the model to adjust and refine its learning over time.
>
>     - In **Appendix H.12**, **Figure 12** shows that head classes quickly accumulate pruned samples, reflecting their redundancy, while tail classes exhibit slower pruning growth, preserving minority samples and preventing further imbalance. A similar pattern is observed in the unlabeled subset, where head classes accumulate more pruned samples due to high-confidence but less informative pseudo-labels. These results demonstrate that DyTrim adaptively prunes based on class frequency and sample utility, removing redundant head-class samples and retaining valuable tail-class data.

---

### Official Review · Reviewer_yN6a · 2025-10-31

**Soundness:** 2
**Presentation:** 2
**Contribution:** 2
**Rating:** 2
**Confidence:** 3

**Summary:**

In their manuscript, the authors address the problem of long-tailed semi-supervised learning. More specifically, the problem of estimating the accumulated biased is approached using techniques of learning dynamics and benchmark images. The proposed framework, DyTrim, uses these findings to guide data pruning. The approach is evaluated on three standard benchmarks.

**Strengths:**

- The manuscript makes use of very recent work such as Learning Dynamics (Ren&Sutherland, ICLR2025) and solid-color input for optimal bias estimation (Xing et al., AAAI2025)
- The writing is generally good, although the text-flow is sometimes confusing
- Figure 1 is a good illustration, although it needs better connection with the main storyline.
- The method has been compared in a comprehensive set of experiments with good results.

**Weaknesses:**

- Both soundness and contribution seem to be problematic, possibly cause by unclear presentation
    - contribution: it remains unclear in which sense the current method goes beyond combining the works by Ren&Sutherland, ICLR2025 (btw: the reference should be updated from the pre-print) and Xing et al., AAAI2025 for guiding sub-sampling
    - soundness: it remains unclear how the authors reflect about their use of definitions and propositions. The presentations of the theory leaves the reader with serious doubts that the construction based on definition 1, subsequent propositions, and theorem 1 is sound. The proofs in the appendix are not mentioned in the main text and seem to be not fully completed (e.g. missing references)
    - writing: although generally good, it is sometimes unnecessary hard to interpret the writing, also caused by some language issues (e.g. line 040: "the approach employ baseline image ")
- Some relevant references are missing, e.g. mixing-based approaches (built on [1])
- The presentation of results is a bit confusing: which results are taken from the literature and which have been reproduced; why deviate reproduced values from the literature, etc.

[1] Zhang et al. Mixup: Beyond empirical risk minimization. ICLR 2018.

**Questions:**

1. how does the proposed method goes beyond combining Ren&Sutherland, ICLR2025 and Xing et al., AAAI2025 for guiding sub-sampling?
1. what is defined in Definition 1? In its current from, it says "decompose" which implies a proposition
1. in which ways go propositions 1 and 3 beyond definition 1 (which might be a proposition)?
1. how can the use of a bias-free first layer (line 173) be justified?
1. what is the intuitive, non-trivial connection between proposition 2 and theorem 1?
1. which results in the experiments were reproduced and which were taken from the literature?
1. in case of deviations: why do the results differ?

---

> ### Author Response · Authors · 2025-11-22
> **Response, Part I/III**
>
> We thank the reviewer very much for the constructive comments and suggestions.
>
> ### **Weaknesses Part**
>
> **W1**. **About contribution, soundness and writing**:** "*Both ... sampling.*"
>
> **Response**:
>
> 1. we would like to clarify that while Ren et. al [1] used learning dynamics to analyze LLM fine-tuning methods, we are the first to extend learning dynamics specifically to analyze logits debiasing in long-tail semi-supervised learning (LTSSL). Our analytical framework offers a more general approach, which can be used to evaluate various techniques, including logits adjustments, reweighting, and resampling. In contrast, Xing's [2] work focuses solely on a specific logits adjustment method, and its results are limited to that particular approach.
>
> 2.  we first revised the **Definition 1** to be **Proposition 1** (in **line 97**, **page 2**). To clarify the connection between the theory and downstream approach, we **have made substantial revisions** in the current version of the paper to better highlight and clarify the theoretical insights. We have strengthened the writing in the **theoretical sections** to ensure that our mechanism for logits debiasing is presented more clearly and in greater detail (in **pages 5-7**) and its corresponding proof in **Appendix C.1-C.5**, **pages 17-20**.
>
> **W2**. **About relevant references**: "*Some relevant ... [1])*"
>
> **Response**: we thank you for pointing out the very useful reference. We have already discussed it in revision.
>
> **W3**. **About presentation of results**: "*The presentation ... etc.*"
>
> **Response**: In the revision, we marked the methods taken from the literature (without `*`) and reproduced (with `*`). If the original paper uses the same dataset and imbalance settings as our experiment, we adopt the results from the original paper. Otherwise, we report the reproduced values with `*`.
>
> [1] Zhang et al. Mixup: Beyond empirical risk minimization. ICLR 2018.
> [2] David Berthelot et.al. Remixmatch: Semi-supervised learning with distribution alignment and augmenta-
> tion anchoring. arXiv preprint arXiv:1911.09785.

---

> ### Author Response · Authors · 2025-11-22
> **Response, Part II/III**
>
> ### **Questions Part**
>
> We thank the reviewer for the series of interesting questions. We answer these questions point by point below.
>
> **Q1**. "*how does ... sub-sampling?*"
>
> **Response**: We would like to clarify that Ren's method [1], which applies learning dynamics as a tool to analyze LLM fine-tuning, cannot be used for guiding subsampling. Xing's work [2] guides logits debias from the perspective of encouraging gradient conflict, and also does not involve guiding subsampling. Our DyTrim is the first to propose using a baseline image to guide dynamic subsampled dataset pruning, thereby obtaining a less biased model.
>
> **Q2**. "*what is ... proposition*"
>
> **Response**: Thank you for your suggestion. We have updated the manuscript to replace "**Definition 1**" with "**Proposition 1**" in the revised version.
>
> **Q3**. "*in which ... proposition)?*"
>
> **Response**: We thank the reviewer for the suggestion. In the work introduced by Ren et al. [1], **Definition 1** analyzes the learning dynamics of supervised classification, focusing on
>
> > *how the model's prediction on a test example $x_o$ changes after a gradient descent (GD) update on a training sample $x_u$*.
>
> In our work, we extended **Proposition 1** for analyzing the learning dynamics in semi-supervised learning by incorporating both supervised and unsupervised samples. Thus, we have **Proposition 2**, focusing on
>
> > *how the model's prediction on $x_o$ changes after a GD update on either a labeled sample $x_b$ or an unlabeled sample $u_b$.*
>
> **Proposition 4** goes a step further by introducing a task-irrelevant baseline image $\mathcal{I}$ as input. This extension offers insights into **how the accumulated class bias manifests during training**, allowing for a deeper understanding of the debiasing mechanisms, focusing on
>
> > *how the model's prediction on this baseline image changes after a GD update on either $x_b$ or $u_b$*.
>
> In summary, both **Propositions** **2** and **4** go beyond **Propositions 1** by generalizing the analysis to semi-supervised learning and incorporating the task-irrelevant baseline image, thus offering a more comprehensive framework for understanding learning dynamics of logits debiasing in long-tail semi-supervised learning.
>
> **Q4**. "*how can ... justified?*"
>
> **Response**: We thank the reviewer for pointing out this question. In semi-supervised learning, many works use WRN-2 [3] as their backbone, such as CDMAD [4], ABC [5], LCGC [2], and CCL [6]. In the WRN-2 architecture, the network first defines an initial convolutional layer:
>
> ```python
> self.init_conv = conv3x3(3, 16)
> ````
>
> However, this 3x3 convolutional layer has its bias set to `False` during implementation:
>
> ```python
> def conv3x3(i_c, o_c, stride=1):
>
>     return nn.Conv2d(i_c, o_c, 3, stride, 1, bias=False)
> ````
> This setup is widely used, and therefore, our analysis is also based on the absence of a bias term in the first layer.
>
> However, in practice, our method is equally suitable if a bias term exists in the first layer of the network, and can be considered a simplified case of Eq. (7). Let's take a 3-layer network $f_\theta$ as an example,
> <div style="overflow-x: auto; white-space: nowrap; border: 1px solid #ccc; padding: 8px;">
>
> \begin{aligned}
> f_\theta(x)=W^{3} \circ \sigma ( W^{2}\circ \sigma (W^{1}x+b^1)+ b^2) + b^3
> \end{aligned}
>
> with $x=0$ and $f^1=W^1x+b$, we have
>
> \begin{aligned}
> f_\theta(x) & = h(f^1(x))\\
> & = W^{3} \circ \sigma ( W^{2}\circ \sigma (W^{1}0+b^1)+ b^2) + b^3 \\
> & = W^{3} \circ \sigma ( W^{2}\circ \sigma (b^1)+ b^2) + b^3 \\
> & = h(b^1)
> \end{aligned}
> </div>
>
> where $h$ denotes all layers except the first layer in $f_\theta$.
>
>
> **Q5**. "*what is the ... theorem 1?*"
>
> **Response**: We thank the reviewer for this insightful question. In **Proposition 3**, we propose using a task-irrelevant solid color image as the baseline to detect the cumulative bias from class imbalance in long-tail semi-supervised learning. We expect the network's output on this baseline image to reflect the class prior, as it should be insensitive to color changes and primarily output a value related to the bias term. Empirical evidence provided in **Figure 2** (in **lines 220-234**, **page 5**) demonstrates that the network's output on the solid color image is closer to the class prior than other baseline images. Additionally, **Theorem 1** offers theoretical assurance that, for the baseline image $\mathcal{I}$, the baseline prediction aligns with the conditional class distribution at the normalized-zero feature state, capturing the class prior induced by the long-tailed training distribution.

---

> > ### Author Response · Authors · 2025-11-22
> > **Response, Part III/III**
> >
> > ### **Questions Part (continuing)**
> >
> > **Q5**. "*what is the ... theorem 1?*"
> >
> > **Response**: We thank the reviewer for this insightful question. In **Proposition 3** (originally Proposition 2), we propose using a task-independent solid color image as the baseline to detect the cumulative bias from class imbalance in long-tail semi-supervised learning. We expect the network's output on this baseline image to reflect the class prior, as it should be insensitive to color changes and primarily output a value related to the bias term. Empirical evidence provided in **Figure 2** (in **lines 220-234**, **page 5**) demonstrates that the network's output on the solid color image is closer to the class prior than other baseline images. Additionally, **Theorem 1** offers theoretical assurance that, for the baseline image $\mathcal{I}$, the baseline prediction aligns with the conditional class distribution at the normalized-zero feature state, capturing the class prior induced by the long-tailed training distribution.
> >
> > **Q6**. "*which results ... literature?*"
> >
> > **Response**: In the revision, we marked the methods taken from the literature and reproduced. We used `*` to indicate reproduced methods, and methods without `*` were directly taken from the literature.
> >
> > **Q7**. "*in case ... differ?*"
> >
> > **Response**: We follow the principle that if the original paper reports results using the same dataset and settings as our experiment, we adopt the results from the original paper. Otherwise, we present our own results based on our implementation. Therefore, the differences are generally due to the use of different imbalance settings in the dataset.
> >
> > ### **References**
> >
> > [1] Yi Ren and Danica J. Sutherland. Learning dynamics of LLM finetuning. ICLR 2025.
> >
> > [2] Weiwei Xing et.al. Lcgc: Learning from consistency gradient conflicting for class-imbalanced semi-supervised debiasing. AAAI 2025.
> >
> > [3] Sergey Zagoruyko and Nikos Komodakis. Wide residual networks. arXiv preprint
> > arXiv:1605.07146, 2016.
> >
> > [4] Hyuck Lee and Heeyoung Kim. Cdmad: Class-distribution-mismatch-aware debiasing for class-
> > imbalanced semi-supervised learning. CVPR 2024.
> >
> > [5] Hyuck Lee, Seungjae Shin, and Heeyoung Kim. Abc: Auxiliary balanced classifier for class-
> > imbalanced semi-supervised learning. NeurIPS 2021.
> >
> > [6] Zi-Hao Zhou et.al. Continuous contrastive learning for long-tailed semi-supervised recognition. NeurIPS 2024.

---

> ### Author Response · Authors · 2025-11-26
>
> Dear Reviewer yN6a,
>
> Thank you again for your thoughtful comments on our submission. As the discussion period is coming to a close, we would like to ask whether our responses have addressed your concerns. If any points remain unclear or if you have additional questions, we would greatly appreciate your guidance and will be happy to clarify further. If you feel that our revisions and explanations have resolved the issues you raised, we would be grateful if you could consider updating your evaluation accordingly.
>
> We sincerely look forward to your feedback, and thank you for your time and effort throughout this process.
>
> Best regards,
>
> Authors of submission 19923

---

> > ### Comment · Reviewer_yN6a · 2025-11-27
> > **Confusing responses**
> >
> > The comments and responses on the weaknesses and questions are acknowledged, but the structure of the response is confusing. The three part answer is partly overlapping, partly complementary. Please consider editing the responses to minimize redundancy and avoid inconsistencies.
> >
> > Regarding the contribution, the explanation gives a clearer picture, but also a narrower scope. Also, it could be easier to verify the current formulation by means of the results.
> >
> > Regarding the correctness, the changes are appreciated, but are also slightly confusing comparing response I vs II.
> >
> > Overall, the manuscript has improved.

---

> > > ### Author Response · Authors · 2025-11-27
> > > **Response**
> > >
> > > We sincerely thank you for your insightful feedback and constructive comments throughout the review process, and we deeply appreciate your endorsement.
> > >
> > > 1. Thank you for your suggestion. We have edited **Responses I** and **II** to minimize redundancy and avoid inconsistencies.
> > > 2. Our main theoretical contribution is extending learning dynamics to SSL methods and providing a unified framework for mechanistic analysis of logits debias methods under class imbalance settings. Inspired by these theoretical contributions, we propose a dynamic dataset pruning method to encourage models to possess lower bias naturally.

---

### Official Review · Reviewer_CLxv · 2025-10-31

**Soundness:** 3
**Presentation:** 3
**Contribution:** 2
**Rating:** 4
**Confidence:** 2

**Summary:**

This paper tackles the problem of semi-supervised learning with heavy class imbalance. It proposes to dynamically prune the dataset according to the model confidence as well as the approximate marginal class distribution based on the logits derived from a baseline black image. The proposed approach performs well on small-scale image benchmarks (32x32).

**Strengths:**

- The proposed approach seems to intuitively make sense in solving the problem (for the most part)
- The proposed approach seems to do well on the various benchmarks explored.
- The presentation of the paper is clear and professional. Figure 2 is very helpful in understanding the proposed approach.

**Weaknesses:**

The paper load for ICLR this year has been large, and so I have not been able to spend as much time as I would like on reviewing. I encourage the authors to correct any errors/misunderstandings I may have with regards to the paper.  Moreover, I do not work in either semi-supervised or long-tailed learning, so my review confidence will be low, and I will defer to other more knowledgeable reviewers.

1. **Motivation is hard to understand**
    1. The learning dynamics theory doesn't feel particularly relevant or well-connected to the downstream approach.
    1. Figure 1 is not very easy for the reader to interpret.
1. **Baseline image design choice**
    1. I am confused by the choice to use a black image as a baseline to extract marginal class probabilities. Why not just directly extract the bias vector from the final fully connected layer? This is usually accessible for most deep learning models in my experience.
    2. Suppose we have a class-balanced dataset of images where a CNN learns to classify different solid colours (e.g. white, black, red). Wouldn't you expect the black baseline image to return a probability vector close to [0,1,0]?
1. **Experiments only on small-scale data**
    1. Although experimental results seem good, they are only on small-scale data (up to 64x64). I would be much more confident in the approach if the authors demonstrated strong performance on imagenet-scale (224x224) scale data such as https://github.com/zhmiao/OpenLongTailRecognition-OLTR.

1. **Complexity of approach**
    1. It seems like the proposed approach is quite complicated with many knobs for the practitioner to adjust. I am a little sceptical whether such an approach could be conveniently adopted.

**Questions:**

I am unsure why pruning of data is performed rather than simply reweighting the sampling frequency. Can the authors clarify this point?

I am open to raising my score if my above issues are addressed; however, I will still keep low confidence even if I do so, as I am not familiar with this field of research.

---

> ### Author Response · Authors · 2025-11-22
> **Response, Part I/II**
>
> We thank the reviewer very much for the constructive suggestions and appreciation of our work. Below we answer your questions.
>
> ### **Weaknesses Part**
>
> **W1**. **Comment:** "*Motivation ... understand.*"
>
> **Response**: We thank the reviewer for this insightful question. While many LTSSL methods focus on empirical logits debiasing, they **lack a clear understanding of the debiasing mechanism**, resulting in a gap in theoretical analysis and principled approaches. Our method **revisits LTSSL through learning dynamics**, **providing a theoretical characterisation of logits debiasing**.
>
> 1. To clarify the connection between the theory and downstream approach, we have improved the statement of **Section 3-4** (in. **pages 5-7**) and its corresponding proof in **Appendix C.1-C.5**, **pages 17-20**, where now we:
>     - First state the dynamics of semi-supervised learning are significantly affected by class bias under class imbalance.
>     - Then state the rationale for choosing a solid black image as the baseline image.
>     - Third state learning dynamics of the baseline image serve as a suitable probe for cumulative bias.
>     - Fourth, using the proposed unified analysis framework to analyse the logits adjustments, reweighting, and resampling methods. The DyTrim algorithm is presented as a byproduct of the analysis.
>
> 2. We have added a more detailed caption to **Figure 1** (**in lines 154-160**, **page 3**), explaining that in the balanced setting, similarity terms $\mathcal{K}^t$ dominate the model's logits updates. However, in the unbalanced setting, class bias is significantly amplified, forcing the model to output predictions aligned with label priors, thereby suppressing the influence of similarity.
>
> **W2**. **Comment:** "*Baseline ... choice.*"
>
> **Response**: We thank the reviewer for pointing out this nice question.
>
> 1. Our choice of a baseline image only requires that it be **task-irrelevant**. Under this requirement, a solid-color image, a Gaussian noise image, or a batch-mean image is all viable choices. Under **Proposition 3** and **Theorem 1**, we select the black image because it not only satisfies task irrelevance but also has a useful property: when a black image is fed into the model, the input to the first layer becomes zero. As a result, all weight-related terms are cancelled out, leaving only the bias term. Let's take a 3-layer network $f_\theta$ as an example,
>
>
> \begin{aligned}
> f_\theta(x)=W^{3} \circ \sigma ( W^{2}\circ \sigma (W^{1}x+b^1)+ b^2) + b^3
> \end{aligned}
>
> with $x=0$ and $f^1=W^1x+b$, we have
>
> \begin{aligned}
> f_\theta(x) & = h(f^1(x))
>     & = W^{3} \circ \sigma ( W^{2}\circ \sigma (W^{1}0+b^1)+ b^2) + b^3
>     & = W^{3} \circ \sigma ( W^{2}\circ \sigma (b^1)+ b^2) + b^3
>     & = h(b^1)
> \end{aligned}
> where $h$ denotes all layers except the first layer in $f_\theta$.
>
> This bias term is transformed by subsequent weights and biases throughout the network until it becomes the final logits. So, while the effect seems to come from the first layer, it actually reflects the **combined influence of all the following layers**.
>
> We also agree that using the bias term from the last layer could act as a proxy. However, it only captures the shift in the decision boundary for linearly separable features at the output layer. In LTSSL settings, where feature learning is often inadequate, this proxy becomes less reliable. As shown in **Figure 2** (in **lines 219-234**, **page 5**), the final-layer bias term does show an imbalance trend that aligns with the label prior, but the black-image baseline gives a much closer approximation to the true label prior.
>
> 2. We agree that if a class-balanced dataset is constructed, and a CNN is explicitly trained to classify different solid-color images, a black baseline image would indeed be expected to produce a probability vector close to [0,1,0]. However, our analysis assumes that the baseline image must be **task-irrelevant**, meaning it cannot be used as a regular training sample and classified by the network. If this condition is violated, the baseline image no longer serves as an indicator of cumulative bias. Thus, alternative choices should be used, such as a noise image or a batch-mean image in **Appendix C.2**. We have added a discussion of this limitation in the revision (in **Appendix J**, **page 31**) to clarify the conditions under which the baseline image may fail.

---

> ### Author Response · Authors · 2025-11-22
> **Response, Part II/II**
>
> ### **Weaknesses Part (continuing)**
>
> **W3**. **Comment:** "*Although experimental ... OLTR.*"
>
> **Response**: We thank the reviewer for this insightful question. To assess the scalability of our method on large-resolution inputs (224 × 224), we conducted experiments on ImageNet-LT. Due to hardware constraints, we set the batch size to 2, in **Table 11** (**page 25**). CDMAD yields a substantial improvement over the FixMatch baseline, increasing bACC from 20.0% to 35.4%, which highlights the effectiveness of incorporating class-distribution modeling under long-tailed imbalance. Building upon the same baseline, our method further pushes performance to 37.2%, achieving the best result among all compared approaches.
>
> **Table 11. Comparison of bACC on ImageNet-LT**
> | Algorithm | ImageNet-LT|
> |-------|---|
> |FixMatch*| 20.0 |
> | w/+CDMAD*| 35.4 |
> | w/+DyTrim| 37.2 |
>
> **W4**. **Comment:**"*It seems ... adopted.*"
>
> **Response**: We thank the reviewer for raising this concern and would like to clarify that our method introduces **only one tunable additional parameter** beyond what standard SSL methods like FixMatch, FlexMatch, and FreeMatch already require: the unlabeled pruning ratio $r$. As reported in **Appendix H.7**, we explore a broad range of $r$ values and observe consistent performance. Following this study, we **fix $r$ to a single default value** across all datasets and backbones. Therefore, DyTrim does not increase the tuning burden for practitioners.
>
> Additionally, **DyTrim is a plug-and-play framework** that operates purely at the data-selection level, without altering the model architecture, training objectives, or optimization pipeline. Integrating DyTrim into an existing SSL codebase requires **only six lines of code** to compute sample-level difficulty scores and apply the corresponding pruning masks during each mini-batch. For clarity, we provide a minimal illustrative snippet below to facilitate straightforward adoption.
>
>
> ```python
> # Encapsulate labeled and unlabeled dataset
> train_labeled_set = DyTrim(train_labeled_set, labelled=True)
> train_unlabeled_set = DyTrim(train_unlabeled_set, ratio=ratio, labelled=False)
> # Calculate and record model bias
> biaseddegree, _ = model(torch.zeros((1, 3, 32, 32)))
> train_labeled_set.__setbiaseddegree__(F.softmax(biaseddegree))
> # Record labeled and unlabeled sample scores
> train_labeled_set.__setscore__(idx, score)
> train_unlabeled_set.__setscore__(idx_u, score_u)
> ```
>
> ### **Questions Part**
>
> **Q1**. **Comment:** "*I am ... point?*"
>
> **Response**: We thank the reviewer for the insightful suggestion. Our per-step influence analysis that reweighting (in **Section 4.2**, **page 6**) and resampling (in **Appendix C.5.1**, **page 20**) operate only by modifying the gradient updates, while leaving a **static set of participating samples**, such as SAW [1]. Our approach is a **special variant of dynamic reweighting**, where the weight of a pruned sample can be masked as 0. More importantly, to the best of our knowledge, it is **the first attempt to use a dynamic pruning method for LTSSL logtis debiasing**. Furthermore, during model iteration, the purging mask is dynamically selected based on the loss function and class priors. DyTrim is explicitly designed as a **data selection level** debiasing mechanism, enabling the algorithm to exclude **redundant or harmful** head samples and suppress **uninformative** samples.
>
> We conducted a case study on the pruned samples in DyTrim during the rebuttal phase as shown in **Appendix H.8** and **H.9**， **pages 27-28**. The results show that DyTrim can generate a more balanced dynamic labeled and unlabeled subset, with a significant suppressive effect on the head class.
>
> [1] Zhengfeng Lai, et.al. Smoothed adaptive weighting for imbalanced semi-supervised learning: Improve reliability
> against unknown distribution data. ICML 2022.

---

> ### Author Response · Authors · 2025-11-26
>
> Dear Reviewer CLxv,
>
> Thank you again for your thoughtful comments on our submission. As the discussion period is coming to a close, we would like to ask whether our responses have addressed your concerns. If any points remain unclear or if you have additional questions, we would greatly appreciate your guidance and will be happy to clarify further. If you feel that our revisions and explanations have resolved the issues you raised, we would be grateful if you could consider updating your evaluation accordingly.
>
> We sincerely look forward to your feedback, and thank you for your time and effort throughout this process.
>
> Best regards,
>
> Authors of submission 19923

---

> > ### Comment · Reviewer_CLxv · 2025-11-26
> >
> > Apologies for the late reply, I have been busy the past week or so. I would first like to thank the authors for their update. The manuscript has been considerably revised based on reviewer feedback and I appreciate the authors' efforts.
> >
> > Most of my issues have been resolved however I have a few follow-up questions and suggestions.
> >
> > 1. Figure 2 provides clear empirical motivation for using an "unrelated" image (solid colour in this case) image over just using the bias, however, this creates a bit of tension with the theoretical motivation which is explicity about extracting said bias from my understanding. I think it is intuitive that an unrelated image will capture the marginal label distribution.
> > 1. The ImageNet 224x224 results should be placed in the main body of the paper.
> >
> >
> > P.S. Table 15 is enormous, you should shrink it down.

---

> > > ### Author Response · Authors · 2025-11-26
> > >
> > > We sincerely thank you for your insightful feedback and constructive comments, and we appreciate the time you've taken to review our work. We are clearer about your concerns and below we answer your questions.
> > >
> > > Q1. "*Figure 2 ... distribution.*"
> > >
> > > **Response**: We agree that it is intuitive that an unrelated image can capture the marginal label distribution, and Figure 2 provides experimental evidence supporting this under well-trained conditions. As shown, the **unrelated image serves as a probe to reveal the model bias** induced by class imbalance in the data. Importantly, because the unrelated image is not theoretically related to the training samples, **its learning dynamics offer a clearer reflection of how class bias accumulates** over time. This insight forms the basis of our theoretical contribution: we leverage the learning dynamics of the unrelated image to analyze the cumulative effect of class bias. Additionally, we use this analysis to reveal the mechanism of different LTSSL algorithms in logits debiasing, such as logits adjustment, reweighting, and resampling, which directly inspired the development of our algorithm.
> > >
> > > Q2. "*The ImageNet 224x224 ... the paper.*"
> > >
> > > **Response**: We agree that ImageNet 224x224 results are an important part of the evaluation and should be prominently featured in the main body of the paper. We have made the necessary revisions to include these results in the main text for better visibility.
> > >
> > > Q3. "*Table 15 ... it down.*"
> > >
> > > **Response**: We agree that Table 15 is quite large, and we have resized it in the updated revision to make it more readable and suitable.

---

### Official Review · Reviewer_hNP6 · 2025-11-03

**Soundness:** 3
**Presentation:** 3
**Contribution:** 3
**Rating:** 6
**Confidence:** 2

**Summary:**

This paper analyzes long-tailed SSL bias through learning dynamics. Key insight: predictions on baseline images (solid-color inputs) \pi_\theta(\mathcal{I}) encode class bias via BatchNorm terms (Theorem 1). DyTrim uses these as pruning ratios for class-aware labeled data pruning and label-agnostic unlabeled pruning. Shows 1-3% improvements over CDMAD on CIFAR-LT/STL-LT/ImageNet-127.

**Strengths:**

Principled framework: decomposing learning dynamics with a formal link between baseline representations and class priors.

Consistency: Works with FixMatch/FlexMatch/FreeMatch and WRN/ViT/ResNet.

Thorough validation: 15+ baselines, diverse settings (matched/mismatched \gamma).

**Weaknesses:**

Incremental Gains: Performance improves by only 1–2 % over CDMAD, which is modest given their conceptual overlap—both methods rely on baseline-image statistics.

**Questions:**

When does the method fail? What are its limitations?

---

> ### Author Response · Authors · 2025-11-22
> **Response**
>
> We thank the reviewer so much for the valuable comments and appreciation of our work and efforts. Below, we address your concerns and questions point by point.
>
> ###  **Our Improvements**
> We would like to first highlight that we have made a series of important improvements in the updated version:
>
> 1. We have improved the statement of **Theorem 1** and **Proposition 4** in **Section 3**, where now we:
>     - First state the dynamics of semi-supervised learning are significantly affected by class bias under class imbalance.
>     - Then state the rationale for choosing a solid black image as the baseline image.
>     - Finally, state learning dynamics of the baseline image serve as a suitable probe for cumulative bias.
> 2. We have added some paragraphs to help the readers to understand the theoretical insights, e.g., the overview of analysis in **Section 4** (**pages 6-7**) and its corresponding proof in **Appendix C.5**, **page 20**).
>
> ###  **Weaknesses Part**
>
> **W1**: *"Incremental ... statistics."*
>
> **Response**: We agree with the reviewer that DyTrim achieves incremental 1-2% gains, as it, like CDMAD, relies on baseline image statistics. However, our algorithm serves as a secondary contribution, with the primary contribution being theoretical: it provides a mechanistic explanation for logits debiasing in long-tail semi-supervised learning. We **have made substantial revisions** in the updated version of the paper to better **highlight with blue** and clarify the theoretical insights. Furthermore, our method is also a unified framework that can be easily integrated into other SSL algorithms，i.e., CCL, FreeMatch, and FlexMatch.
>
>
> ###  **Questions Part**
>
> **Q1**: *"When does the method fail? What are its limitations?"*
>
> **Response**: Our method has two main limitations. First, it relies on a task-irrelevant baseline image, i.e., an image with no pattern and not used for training, to serve as a bias indicator. If this baseline image is also used as a regular training sample and classified by the network, it no longer reliably reflects the accumulated bias, and the effectiveness of our debiasing mechanism can degrade. Second, our current theoretical analysis and design do not explicitly cover architectures with auxiliary classification heads or semi-supervised pipelines based on mixup-style interpolations. As a result, DyTrim provides only limited improvements when directly applied to these types of methods, as shown in **Table 5** (**lines 518-529**, **page 10**). We have discussed these **limitations** in the reversion(in **Appendix J**, **page 31**), and we regard extending the framework to these settings as a promising direction for future work.

---

### Author Response · Authors · 2025-11-22
**General Response to All Reviewers and Summary of Changes of the Revision**

We thank all reviewers a lot for the valuable comments and insightful questions and suggestions.

To answer questions and address the concerns of the reviewers, we make the corresponding changes in the rebuttal revision, which will be summarized as follows: revised and strengthened theoretical explanation, Supplementary experimental evidence and visualizations; extension of DyTrim.

For clarity, all changes in the revised manuscript are highlighted in `blue`.


### 1. Revised and strengthened theoretical explanation

As suggested by Reviewer **CLxv**, **yN6a**, **hNP6** and **jDsT**, we revised and strengthened the theoretical explanation of the paper.

(1). We have made substantial revisions in the current version of the paper to **better highlight and clarify the theoretical insights**. We have strengthened the writing in the theoretical sections to ensure that our mechanism for logits debiasing is presented more clearly and in greater detail (in **Section 3-4**, **pages 5-7**) and its corresponding proof in **Appendix C.1-C.5,** **pages 17-20**.

(2). Using our proposed **unified framework**, we add paragraphs to analyze several typical algorithms in long-tail semi-supervised learning (LTSSL), such as **logits adjustment** (in in **Section 4.1**, **page 6**), **reweighting** (in **Section 4.2**, **page 6-7**), and **resampling** (in **Appendix C.5.1**, **page 20**). Based on this, we analyze the logits debias method for pruning and introduce our DyTrim algorithm as a slide product (in **Section 4.3**, **page 7-8**).

### 2. Supplementary experimental evidence and visualizations

(1). We added experimental evidence and visualizations of the pruning process. This includes the dynamic changes in loss during iterations (in **Figure 11**); the specific pruning values for each sample in each iteration (in **Figure 12**); the maximum, minimum, and average probabilities of pruning for each class of samples during training (in **Figure 9**, **page 28**). The bias changes resulting from dynamic scaling and non-scaling gradients (in **Figure 10**, **page 28**).

(2). We conducted experiments on ImageNet-LT, in **Table 11** (**lines 1374-1382**, **page 26**). Building upon the same baseline, i.e., FixMatch, our method further pushes performance to 37.2%, achieving the best result among all compared approaches.

### 3. Extension of DyTrim

As suggested by Reviewer **jDsT**, we discuss the extension of DyTrim in **Appendix H.10** (starting from **line 1488**, **page 28** of the revision).

Our DyTrim algorithm is flexible, which can be easily incorporated into other sophisticated class-aware pruning strategies for unlabeled samples.

We assign an adaptive scaling factor to each class, i.e., a smaller scaling factor for head classes and a larger scaling factor for tail classes. The dynamic scaling factor is calculated as

$1 - \pi_{\theta} (\mathcal{I})_{\hat q_{b,c}} + 1/(1-r)$

This formula stabilizes the loss magnitude during training and prevents predictions from shifting towards the majority class. We further compare fixed scaling and dynamic scaling in **Table 15**, **page 29**.

---

### Author Response · Authors · 2025-12-01
**Summary of the discussion for the Area Chair**

Thank you to all the reviewers for your considerable time and effort in the review, and for your positive assessment. We have been informed that all review scores have been reverted to their pre-discussion status, and no further reviewer discussions or public comments are allowed. We sincerely appreciate the Area Chair’s time and effort in considering our responses and discussions, and his or her fair review. To facilitate the area chair to consider the existing discussion and improved scores, we summarize the discussions with all the reviewers:

### *Score*:
The reviewers gave positive feedback on our research. By the time the review scores were reverted, we have received the following ratings:

**Reviewer hNP6: 6**

**Reviewer CLxv:  4**

**Reviewer yN6a: 2**

**Reviewer jDsT: 6**

Most reviewers provided a positive assessment. We had extensive discussions with Reviewers **hNP6**, **jDsT**, **yN6a**, and **CLxv**, and we believe we have fully addressed their concerns. Reviewer **jDsT** has improved the rating to **6**. Prior to the **leak bug**, we were actively engaged in constructive discussions with **Reviewer CLxv** and **Reviewer yN6a**, having addressed and resolved their questions.

### *For the Reviewer hNP6 (rating 6)*:

**Reviewer hNP6 appreciated the principled learning-dynamics perspective** and raised questions about limitations, while noting that improvements over CDMAD can be modest. In our rebuttal, we clarified that the primary contribution is a mechanistic theory framework, and we added a clearer limitations discussion (e.g., the baseline image should be task-irrelevant; the analysis may not directly cover auxiliary-head or mixup-style SSL pipelines), together with supporting discussion and evidence in the revision. These responses address Reviewer hNP6’s concerns.


### *For the Reviewer CLxv (rating 4)*:

**The Reviewer CLxv has affirmed the innovation, robustness, and contribution of this paper**. In our rebuttal, we rewrote and reorganized the theory sections to make the theoretical storyline explicit and connected to DyTrim. We justified the black-image baseline theoretically and practically, and explained why using only the final-layer bias is less reliable under LTSSL. We added ImageNet-LT results and clarified that DyTrim introduces only one extra tunable parameter and is plug-and-play at the data-selection level. The Reviewer CLxv noted that most of their issues had been resolved and additionally provided two suggestions on improving the paper’s layout, indicating an **openness to raising the score based on our constructive discussion**. However, due to the **leak bug**, this exchange had to be prematurely halted.

### *For the Reviewer yN6a (rating 2)*:

**Reviewer yN6a appreciated that our revisions improved correctness and made the explanations clearer**. They mainly questioned the novelty beyond combining prior work, the clarity of the definition–proposition–theorem chain and proof references, missing citations, and the distinction between reproduced vs. borrowed experimental results. In response, we clarified the novelty as a unified learning-dynamics framework for LTSSL logits debiasing across multiple debiasing families, streamlined the theoretical presentation, strengthened the baseline-image theory–evidence link, and made experimental reporting explicit. The **Reviewer yN6a later acknowledged that the manuscript improved overall**.

### *For the Reviewer jDsT (rating 6)*:

**Reviewer jDsT valued the theoretical foundation** and raised concerns about potential tail-class undersampling from pruning, figure interpretation, gradient rescaling, notation, and pruning details. We clarified that DyTrim is class-adaptive and does not undersample tail classes, added quantitative evidence, and addressed the figure/rescaling/notation issues. **Reviewer jDsT** has approved our responses satisfactorily, considered the paper theoretically well-grounded, and raised the score to **6**.

We have incorporated the main theoretical and experimental results from our discussion into the revision. We promise that the summaries of the review scores and discussions are all true and reliable. Thank you once again to the Area Chair for your efforts to provide a fair review.

---

### Meta-Review · Area_Chair_BKQY · 2026-01-07

**Summary:**

Reviewer **hNP6** gives this paper a positive rating (6) and suggests discussing the limitations of the proposed method. The authors have added a discussion of limitations to Appendix J.

Reviewer **CLxv** states he/she does not work in the field of semi-supervised learning and long-tailed learning, and he does not spend too much time reviewing this paper. So I prefer not to consider his/her comments in making a recommendation.

Reviewer **yN6a** gave an initial rating of 2, and his/her main concerns are the novelty of the proposed method (how does the proposed method go beyond combining Ren&Sutherland, ICLR2025, and Xing et al., AAAI2025 for guiding sub-sampling?) and the writing of this paper.

Reviewer **jDsT** acknowledges the theoretical contribution of this paper. And he/she has some concerns about the undersampling issue for the tail classes and the lack of comparison with the latest literature.

Most reviewers acknowledge the contribution of this work (good theoretical foundation, simple method, good classification performance). I prefer to recommend acceptance of this paper, although its initial scores are not too high.

**Reviewer Concerns:**

I think most of the major concerns have been addressed by the authors in the rebuttal.

Concerns remain: this paper does not compare with recent SOTA methods.

**Reviewer Scores:**

I think **Reviewer yN6a** will raise the rating from 2 to 4 or 6.

I think **Reviewer jDsT** will raise the rating from 4 to 6.

---

### Decision · Program_Chairs · 2026-01-26

Accept (Poster)